# THE POWER OF REGULARIZATION IN SOLVING EXTENSIVE-FORM GAMES

**Mingyang Liu**[♮ 1]**, Asuman Ozdaglar** [2]**, Tiancheng Yu** [2]**, Kaiqing Zhang** [3]

[1]Institute for Interdisciplinary Information Sciences, Tsinghua University
[2]LIDS, EECS, Massachusetts Institute of Technology
[3]University of Maryland, College Park
[1]`liumy19@mails.tsinghua.edu.cn`
[2]`{asuman,yutc}@mit.edu`
[3]`kaiqing@umd.edu`

## ABSTRACT

In this paper, we investigate the power of *regularization*, a common technique in reinforcement learning and optimization, in solving extensive-form games (EFGs). We propose a series of new algorithms based on regularizing the payoff functions of the game, and establish a set of convergence results that strictly improve over the existing ones, with either weaker assumptions or stronger convergence guarantees. In particular, we first show that dilated optimistic mirror descent (DOMD), an efficient variant of OMD for solving EFGs, with adaptive regularization can achieve a fast $\widetilde{O}(1/T)$ last-iterate convergence in terms of duality gap and distance to the set of Nash equilibrium (NE) without uniqueness assumption of the NE. Second, we show that regularized counterfactual regret minimization (`Reg-CFR`), with a variant of optimistic mirror descent algorithm as regret-minimizer, can achieve $O(1/T^{1/4})$ best-iterate, and $O(1/T^{3/4})$ average-iterate convergence rate for finding NE in EFGs. Finally, we show that `Reg-CFR` can achieve asymptotic last-iterate convergence, and optimal $O(1/T)$ average-iterate convergence rate, for finding the NE of perturbed EFGs, which is useful for finding approximate extensive-form perfect equilibria (EFPE). To the best of our knowledge, they constitute the first last-iterate convergence results for CFR-type algorithms, while matching the state-of-the-art average-iterate convergence rate in finding NE for non-perturbed EFGs. We also provide numerical results to corroborate the advantages of our algorithms.

## 1 INTRODUCTION

Extensive-form games (EFGs) are widely used in modeling sequential decision-making of multiple agents with imperfect information. Many popular real-world multi-agent learning problems can be modeled as EFGs, including Poker (Brown and Sandholm, 2018; 2019b), Scotland Yard (Schmid et al., 2021), Bridge (Tian et al., 2020), cloud computing (Kakkad et al., 2019), and auctions (Shubik, 1971), etc. Despite the recent success of many of these applications, efficiently solving large-scale EFGs is still challenging.

Solving EFGs typically refers to as finding a Nash equilibrium (NE) of the game, especially in the two-player zero-sum setting. In the past decades, the most popular methods in solving EFGs are arguably regret-minimization based methods, such as counterfactual regret minimization (CFR) (Zinkevich et al., 2007) and its variants (Tammelin et al., 2015; Brown and Sandholm, 2019a). By controlling the regret of each player, the average of strategies constitute an approximated NE in two-player zero-sum games, which is called *average-iterate* convergence (Zinkevich et al., 2007; Tammelin et al., 2015; Farina et al., 2019a).

However, averaging the strategies can be undesirable, which not only incurs more computation (Bowling et al., 2015) (additional memory and computation for the average strategy), but also intro-

---

[♮]Alphabetical Order

duces additional representation and optimization errors when function approximation is used. For example, when using neural networks to parameterize the strategies, the averaged strategy may not be able to be represented properly and the optimization object can be highly non-convex. Therefore, it is imperative to understand if (approximated) NE can be efficiently solved *without* average, which motivates the study of *last-iterate convergence*. In fact, the popular CFR-type algorithms mentioned above only enjoy average-iterate convergence guarantees so far (Zinkevich et al., 2007; Tammelin et al., 2015; Farina et al., 2019a), and it is unclear if such a last-iterate convergence is achievable for this type of algorithms.

The recent advances of Optimistic Mirror Descent (Rakhlin and Sridharan, 2013; Mertikopoulos et al., 2019; Wei et al., 2021; Cai et al., 2022) shed lights on how to achieve last-iterate convergence for solving normal-form games (NFGs), a strict sub-class of EFGs. The last-iterate convergence in EFGs has not received attention until recently (Bowling et al., 2015; Farina et al., 2019c; Lee et al., 2021). Specifically, Bowling et al. (2015) provided some empirical evidence of last-iterate convergence for CFR-type algorithms, while Farina et al. (2019c) empirically proved that OMD enjoyed last-iterate convergence in EFGs. Lee et al. (2021) proposed an OMD variant with the first last-iterate convergence guarantees in EFGs, but the solution itself might have room for improvement: To make the update computationally efficient, the mirror map needs to be generated through a *dilated* operation (see §2 for more details); and for this case, the analysis in Lee et al. (2021) requires the NE to be unique. In particular, an important and arguably most well-studied instance of OMD for no-regret learning over simplex, i.e., the optimistic multiplicative weights update (OMWU) (Daskalakis and Panageas, 2019; Wei et al., 2021), cannot be shown to have explicit last-iterate convergence rate so far , without such a uniqueness condition, even for normal-form games. Anagnostides et al. (2022) can only guarantee an asymptotic last-iterate convergence rate without uniqueness assumption[1]. Indeed, it is left as an open question in (Wei et al., 2021) if the uniqueness condition is necessary for OMWU to converge with an explicit rate for this strict sub-class of EFGs, when constant stepsize is used.

In this paper, we remove the uniqueness condition, while establishing the last-iterate convergence for Dilated Optimistic Mirror Descent (`DOMD`) type methods. The solution relies on exploiting the power of the regularization techniques in EFGs. Our last-iterate convergence guarantee is not only for the convergence of *duality gap*, a common metric used in the literature, but also for the actual iterate, i.e., the convergence of the distance to the set of NE. This matches the *bona fide* last-iterate convergence studied in the literature, e.g., Daskalakis and Panageas (2019); Wei et al. (2021), and such a kind of last-iterate guarantee is unknown when the mirror map is either dilated or entropy-based. More importantly, the techniques we develop can also be applied to CFR, resulting in the first last-iterate convergence guarantee for CFR-type algorithms. We detail our contributions as follows.

**Contributions.** Our contributions are mainly four-fold: (i) We develop a new type of dilated OMD algorithms, an efficient variant of OMD that exploits the structure of EFGs, with adaptive regularization (`Reg-DOMD`), and prove an explicit convergence rate of the duality gap, without the uniqueness assumption of the NE. (ii) We further establish a last-iterate convergence rate for dilated optimistic multiplicative weights update to the NE of EFGs (beyond the duality gap as in Cen et al. (2021b), for the NFG setting), when constant stepsize is used. This also moves one step further towards solving the open question for the NFG setting, about whether the uniqueness assumption can be removed to prove last-iterate convergence of the authentic OMWU algorithms with constant stepsizes (Daskalakis and Panageas, 2019; Wei et al., 2021). (iii) For CFR-type algorithms, using the regularization technique, we establish the first best-iterate convergence rate of $O(1/T^{1/4})$ for finding the NE of non-perturbed EFGs, and last-iterate asymptotic convergence for finding the NE of perturbed EFGs in terms of duality gap, which is useful for finding approximate *extensive-form perfect equilibrium* (EFPE) (Selten, 1975). (iv) As a by-product of our analysis, we also provide a faster and optimal rate of $O(1/T)$ average-iterate convergence guarantee in finding NE of perturbed EFGs (see formal definition in §4.1), while also matching the state-of-the-art guarantees for CFR-type algorithms in finding NE for the non-perturbed EFGs in terms of duality gap (Farina et al., 2019a).

**Technical challenges.** We emphasize the technical challenges we address as follows. First, by adding regularization to the original problem, `Reg-DOMD` will converge to the NE of the regularized

---

[1] A recent result (Anagnostides et al., 2022, Theorem 3.4) also gave a best-iterate convergence result with rate, but only asymptotic convergence result for the last iterate.

| | Algorithm | Games | Duality Gap | Iterate | Require NE Unique |
|---|---|---|---|---|---|
| Daskalakis and Panageas (2019) | OMWU | | Asymptotic | | Yes |
| Wei et al. (2021) | OMWU | NFGs | $O(1/T)$ *(G)* | | |
| | OGDA | | Linear *(L)* | | No |
| Cen et al. (2021b) | Reg-OMWU | | $\widetilde{O}(1/T)$ | No | |
| Lee et al. (2021) | DOMWU | EFGs | $O(1/T)$ *(G)* Linear *(L)* | | Yes |
| Reg-DOMD (Ours) | Reg-DOMWU | | $\widetilde{O}(1/T)$ | | No |
| | Reg-DOGDA | | | | |

Table 1: Comparisons between our methods and previous last-iterate convergence methods. *(D)OMWU* refers to (Dilated) Optimistic Multiplicative Weights Update (Daskalakis and Panageas, 2019) and *(D)OGDA* refers to (Dilated) Optimistic Gradient Descent Ascent (Daskalakis et al., 2018; Liang and Stokes, 2019; Mokhtari et al., 2020). And Reg-DOMWU *(*Reg-DOGDA*)* refers to DOMWU (DOGDA) with regularization. The fifth column *Iterate* refers to the Euclidean distance to NE. *(G), (L)* refer to global convergence rate and local convergence rate, respectively.

| | Algorithm | Regret Minimizer | Last | Average |
|---|---|---|---|---|
| NE | CFR (Zinkevich et al., 2007) | RM (Hart and Mas-Colell, 2000) | No | $O(1/\sqrt{T})$ |
| | CFR+ (Tammelin et al., 2015) | RM+ (Tammelin et al., 2015) | | |
| | Stable Predictive CFR (Farina et al., 2019a) | Optimistic FTRL (Syrgkanis et al., 2015) | | $O(1/T^{3/4})$ |
| | Reg-CFR (Ours) | Reg-DS-OptMD | Best-Iterate | |
| NE of Perturbed EFG | CFR+ (Farina et al., 2017) | RM+ (Tammelin et al., 2015) | No | $O(1/\sqrt{T})$ |
| | Reg-CFR (Ours) | Reg-DS-OptMD | Asymptotic Last-Iterate | $O(1/T)$ **Optimal** |

Table 2: We show the performance of CFR-type algorithms in finding NE and NE of perturbed EFG (see §4.1). The fourth column *Last* and the fifth column *Average* represent last-iterate convergence and average-iterate convergence individually. Note that we here view *Best-Iterate* as some convergence guarantee similar to (but slightly weaker than) the last-iterate ones.

problem, instead of that of the original one. By controlling the regularization, one can readily obtain a convergence guarantee of the regularized algorithm in terms of duality gap, as in Cen et al. (2021b) for NFGs. However, it is highly non-trivial to connect back to the iterate convergence. We achieve so by proving a relationship between the distance to the NE set and the duality gap (See Lemma D.6 for the complete proof).

Second, it is challenging to obtain last-iterate convergence guarantees in CFR-type algorithms, since the regret minimizer in each information set cannot synchronize with the other regret minimizers, and they operate independently. Hence, unlike OMD, the individual iterates (not the average) obtained from these independent regret-minimizers may reach an NE of the game asynchronously. For the same reason, it is also challenging to extend the fast-rate no-regret learning results from NFGs (over simplex) to EFGs under the CFR framework.

We provide a detailed related work discussion in Appendix A. Here we provide Table 1 and Table 2 to compare our work with the literature.

## 2 PRELIMINARIES

**Notation.** We use $x_i$ to denote the *i*-th coordinate of vector $\boldsymbol{x}$ and $\|\boldsymbol{x}\|_p$ to denote its *p*-norm. By default, we use $\|\boldsymbol{x}\|$ to denote the 2-norm $\|\boldsymbol{x}\|_2$. We use $\Delta_m$ to denote the $m-1$ dimension probability simplex $\{\boldsymbol{x} \in [0,1]^m : \sum_{i=1}^m x_i = 1\}$, and we sometimes omit the subscript $m$ when

it is clear from the context. For any convex and differentiable function $\psi$, its associated Bregman divergence is defined as $D_\psi(\boldsymbol{u}, \boldsymbol{v}) := \psi(\boldsymbol{u}) - \psi(\boldsymbol{v}) - \langle \nabla \psi(\boldsymbol{v}), \boldsymbol{u} - \boldsymbol{v} \rangle$. Finally, we use $\prod_{\mathcal{C}}(\boldsymbol{u})$ to denote the projection of $\boldsymbol{u}$ to a convex set $\mathcal{C}$ with respect to Euclidean distance.

**Bilinear optimization problem.** Strategies in two-player zero-sum extensive-form games with perfect recall can be interpreted in sequence-form (Von Stengel, 1996). Thus, finding the Nash equilibrium reduces to solving a bilinear saddle-point problem,

$$\min_{\boldsymbol{x} \in \mathcal{X}} \max_{\boldsymbol{y} \in \mathcal{Y}} \quad \boldsymbol{x}^\top \boldsymbol{A} \boldsymbol{y} \tag{2.1}$$

where $\mathcal{X} \subset \mathbb{R}_T^M, \mathcal{Y} \subset \mathbb{R}_T^N$ are the decision sets for min/max players called *treeplexes* (to be defined next). In sequence-form representation, $x_i$ denotes the probability of reaching node $i$ in the treeplex when only counting the uncertainty incurred by the min-player, and $y_i$ can be interpreted similarly. The matrix $\boldsymbol{A} \in [-1, 1]^{M \times N}$, where $\boldsymbol{A}_{i,j}$ denotes the payoff of the max-player when the min-player reaches $i$ and max-player reaches $j$. Nash equilibria are just the solutions to Eq (2.1). We define $\mathcal{Z}^* = \mathcal{X}^* \times \mathcal{Y}^*$ to denote the set of NE, which is always convex for two player zero-sum game.

For convenience, we use $P := M + N$ to denote the dimension of problem (2.1), and concatenate the sequence form for both players by defining $\boldsymbol{z} := (\boldsymbol{x}, \boldsymbol{y}) \in \mathcal{Z} := \mathcal{X} \times \mathcal{Y}$ and the gradient of the bilinear form (2.1) by defining $F(\boldsymbol{z}) := (\boldsymbol{A}\boldsymbol{y}, -\boldsymbol{A}^\top \boldsymbol{x})$. By re-normalizing $\boldsymbol{A}$, we can assume $\|F(\boldsymbol{z})\|_\infty \leq 1$ without loss of generality.

**Treeplex and dilated regularizer.** The structure of a sequence-form is enforced implicitly by the *treeplexes*, which we define formally here:

**Definition 2.1** ( Hoda et al. (2010)). Treeplex is recursively defined as follows:

1. Each probability simplex is a treeplex.

2. The Cartesian product of multiple treeplexes is a treeplex.

3. The branching of two treeplexes is a treeplex, where for integers $m, n > 0$, the branching of two treeplexes $\mathcal{Z}_1 \subset \mathbb{R}_T^m, \mathcal{Z}_2 \subset \mathbb{R}_T^n$ on index $i \in \{1, 2, ..., m\}$ is defined as

$$\mathcal{Z}_1 \boxed{i} \mathcal{Z}_2 = \{(\boldsymbol{u}, u_i \cdot \boldsymbol{v}) : \boldsymbol{u} \in \mathcal{Z}_1, \boldsymbol{v} \in \mathcal{Z}_2\}. \tag{2.2}$$

See an illustration of treeplex in Figure 1 of Appendix A. The simplexes in the treeplex specify the decision points for both players, which are also called *information sets* in the EFG literature (Zinkevich et al., 2007; Tammelin et al., 2015; Farina et al., 2019a). The collection of information sets in treeplex $\mathcal{Z}$ is denoted as $\mathcal{H}^\mathcal{Z}$. For any $h \in \mathcal{H}^\mathcal{Z}$, we use $\Omega_h$ to denote the indices in $\mathcal{Z}$ belonging to decision point $h$ and $h(i)$ to denote the information set that index $i$ belongs to. That is, $h(i) = h$ if and only if $i \in \Omega_h$. We use $\sigma(h)$ to denote the index of the parent variable of $h$ and $\mathcal{H}_i = \{h \in \mathcal{H}^\mathcal{Z} : \sigma(h) = i\}$. For a simplex $\mathcal{Z}$, the parent of the only information set $h \in \mathcal{H}^\mathcal{Z}$ does not exist and we use $\sigma(h) = 0$ to denote it. And when applying Cartesian product on multiple treeplexes, it will not change the parent of any information set. When we branch two treeplexes, that is $\mathcal{Z}_1 \boxed{i} \mathcal{Z}_2$, then the parent of all information set $h \in \mathcal{H}^{\mathcal{Z}_2}$ with $\sigma(h) = 0$ will be updated to $\sigma(h) = i$. For convenience, we use $\boldsymbol{z}_h$ to denote the slice of $\boldsymbol{z}$ with indices in $\Omega_h$. Let $C_\Omega := \max_{h \in \mathcal{H}^\mathcal{Z}} |\Omega_h|$ denote the maximum number of indices in each individual information set. For convenience, we define vector $\boldsymbol{q} \in \mathbb{R}^P$ with $q_i := z_i / z_{\sigma(h(i))}$ for any $i$. In the EFG terminology, $\boldsymbol{q}_h \in \mathbb{R}^{|\Omega_h|}$, the slice of $\boldsymbol{q}$ in information set $h$, is the probability distribution of actions in information set $h$.

The treeplex structure motivates a natural *dilation* operation to generate regularizers that leads to efficient computation in EFGs (Hoda et al., 2010). For any strongly-convex base regularizer $\psi^\Delta$ defined on a simplex, the dilated regularizer is defined by

$$\psi^\mathcal{Z}(\boldsymbol{z}) := \sum_{h \in \mathcal{H}^\mathcal{Z}} \alpha_h z_{\sigma(h)} \psi^\Delta \left( \frac{\boldsymbol{z}_h}{z_{\sigma(h)}} \right), \tag{2.3}$$

where $z_{\sigma(h)}$ is the probability of reaching the parent variable of information set $h$. And $\alpha_h$ is the $h^{th}$ element of vector $\boldsymbol{\alpha} \in \mathbb{R}_+^{|\mathcal{H}^\mathcal{Z}|}$ which is some hyper-parameter set according to $\psi^\Delta$ to

guarantee that $\psi^{\mathcal{Z}}$ is 1-strongly convex with respect to 2-norm (Hoda et al., 2010; Kroer et al., 2020), i.e., $D_{\psi^{\mathcal{Z}}}(\boldsymbol{z}_1, \boldsymbol{z}_2) \geq \frac{1}{2}\|\boldsymbol{z}_1 - \boldsymbol{z}_2\|^2$. Two common base regularizers are the negative entropy $\psi^{\Delta}_{\text{Entropy}}(\boldsymbol{p}) = \sum_i p_i \log p_i$ and the Euclidean norm $\psi^{\Delta}_{\text{Euclidean}}(\boldsymbol{p}) = \sum_i p_i^2$, where $\boldsymbol{p} \in \Delta$ is a probability distribution.

**Finding NE and regret minimization.** Given a strategy $\boldsymbol{z}$ in sequence form, there are two criteria to evaluate the performance:

- the Euclidean distance to the set of NE $\| \prod_{\mathcal{Z}^*}(\boldsymbol{z}) - \boldsymbol{z}\|$,
- the duality gap $\max_{\widehat{\boldsymbol{z}} \in \mathcal{Z}} F(\boldsymbol{z})^\top(\boldsymbol{z} - \widehat{\boldsymbol{z}})$.

When one or both of the above quantities are close to zero, we find an approximate NE. A common approach to minimize duality gap is by *regret minimization*, where we define the (external) regret of the min-player as

$$R_T^{\mathcal{X}} := \sum_{t=1}^{T} l_t(\boldsymbol{x}_t) - \min_{\widehat{\boldsymbol{x}} \in \mathcal{X}} \sum_{t=1}^{T} l_t(\widehat{\boldsymbol{x}}), \tag{2.4}$$

where $l_t$ is the loss function at iteration $t$ and $\boldsymbol{x}_t$ is the output of the regret minimizer at iteration $t$. Regret of the max-player can be defined similarly.

When regret is growing sublinearly with respect to $T$, the average regret is converging to zero (hence the name no-regret). The following Nash folklore theorem implies that the average strategy will converge to NE.

**Lemma 2.2.** For a bilinear zero-sum game where $l_t^{\mathcal{X}}(\boldsymbol{x}_t) = -l_t^{\mathcal{Y}}(\boldsymbol{y}_t) = \boldsymbol{x}_t^\top \boldsymbol{A} \boldsymbol{y}_t$, the duality gap of the average strategy $(\frac{1}{T}\sum_{t=1}^{T}\boldsymbol{x}_t, \frac{1}{T}\sum_{t=1}^{T}\boldsymbol{y}_t)$ is bounded by $(R_T^{\mathcal{X}} + R_T^{\mathcal{Y}})/T$.

## 3 REGULARIZED DILATED OPTIMISTIC MIRROR DESCENT (REG-DOMD)

### 3.1 SOLVING A REGULARIZED PROBLEM

To obtain a faster convergence rate for OMD algorithms, we will solve the NE of the regularized problem below (and thus strongly convex-concave) as an intermediate step. In the literature (McKelvey and Palfrey, 1995), the solution to the regularized problem is called the quantal-response equilibrium (QRE), when the regularizer $\psi^{\mathcal{Z}}$ is entropy:

$$\min_{\boldsymbol{x} \in \mathcal{X}} \max_{\boldsymbol{y} \in \mathcal{Y}} \ \boldsymbol{x}^\top \boldsymbol{A} \boldsymbol{y} + \tau \psi^{\mathcal{Z}}(\boldsymbol{x}) - \tau \psi^{\mathcal{Z}}(\boldsymbol{y}) \tag{3.1}$$

where $\tau \in (0, 1]$ is the weight of regularization and $\psi^{\mathcal{Z}}$ is a strongly-convex regularizer. Thanks to the strong convexity of $\psi^{\mathcal{Z}}$, Eq (3.1) has a unique NE, denoted by $z_\tau^*$. For $t = 1, 2, ...$, the update rule of optimistic mirror descent for the regularized problem (3.1), which we refer to as Reg-DOMD, can be written as

$$\begin{aligned}
\boldsymbol{z}_t &= \underset{\boldsymbol{z} \in \mathcal{Z}}{\operatorname{argmin}} \ \langle \boldsymbol{z}, F(\boldsymbol{z}_{t-1}) + \tau \nabla \psi^{\mathcal{Z}}(\widehat{\boldsymbol{z}}_t) \rangle + \frac{1}{\eta} D_{\psi^{\mathcal{Z}}}(\boldsymbol{z}, \widehat{\boldsymbol{z}}_t) \\
\widehat{\boldsymbol{z}}_{t+1} &= \underset{\boldsymbol{z} \in \mathcal{Z}}{\operatorname{argmin}} \ \langle \boldsymbol{z}, F(\boldsymbol{z}_t) + \tau \nabla \psi^{\mathcal{Z}}(\widehat{\boldsymbol{z}}_t) \rangle + \frac{1}{\eta} D_{\psi^{\mathcal{Z}}}(\boldsymbol{z}, \widehat{\boldsymbol{z}}_t)
\end{aligned} \tag{3.2}$$

where we set $\boldsymbol{z}_0 = \widehat{\boldsymbol{z}}_1$ as uniform strategy, i.e., $\frac{\boldsymbol{z}_{0,h}}{\boldsymbol{z}_{0,\sigma(h)}}$ is uniform distribution in $\Delta_{|\Omega_h|}$, and $\eta > 0$ is the stepsize. The Dilated Optimistic Mirror Descent (DOMD) (Lee et al., 2021) now becomes a special case when $\tau = 0$. We call the update rule (3.2) Regularized Dilated Optimistic Multiplicative Weights Update (Reg-DOMWU) when the base regularizer $\psi^{\Delta}$ is negative entropy, and Regularized Dilated Optimistic Gradient Descent Ascent (Reg-DOGDA) when $\psi^{\Delta}$ is Euclidean norm.

As desired, $\widehat{\boldsymbol{z}}_{t+1}$ converges to $z_\tau^*$ at a linear rate for any fixed $\tau$.

**Theorem 3.1.** With $\eta \leq \frac{1}{8P}$, $\tau \leq 1$ and $\psi^{\mathcal{Z}}$ being a 1-strongly convex function with respect to the 2-norm, Reg-DOMD guarantees that $D_{\psi^{\mathcal{Z}}}(z_\tau^*, \widehat{\boldsymbol{z}}_{t+1}) \leq (1 - \eta\tau)^t D_{\psi^{\mathcal{Z}}}(z_\tau^*, \widehat{\boldsymbol{z}}_1)$ for any $t \geq 1$ when we initialize $\boldsymbol{z}_0 = \widehat{\boldsymbol{z}}_1$.

The results in Theorem 3.1 are for general dilated regularizers, and apply to the regularized version of two representative algorithms, `Reg-DOMWU` and `Reg-DOGDA`, as studied in Lee et al. (2021). The detailed proof is postponed to Appendix C. We sketch the proof below.

**Proof sketch of Theorem 3.1.** When $\psi^{\mathcal{Z}}$ is a 1-strongly convex function with respect to 2-norm and $\eta \le \frac{1}{8P}$, then for any $z \in \mathcal{Z}$ and $t \ge 1$, we have

$$\eta\tau\psi^{\mathcal{Z}}(z) - \eta\tau\psi^{\mathcal{Z}}(z_t) + \eta F(z_t)^\top(z_t - z) \tag{3.3}$$

$$\le (1 - \eta\tau)D_{\psi^{\mathcal{Z}}}(z, \widehat{z}_t) - D_{\psi^{\mathcal{Z}}}(z, \widehat{z}_{t+1}) - D_{\psi^{\mathcal{Z}}}(\widehat{z}_{t+1}, z_t) - \frac{7}{8}D_{\psi^{\mathcal{Z}}}(z_t, \widehat{z}_t) + \frac{1}{8}D_{\psi^{\mathcal{Z}}}(\widehat{z}_t, z_{t-1}),$$

which is adapted from the standard OMD analysis (Rakhlin and Sridharan, 2013), but for the regularized problem. See Lemma C.2 for the proof.

Taking $z = z_\tau^*$ in Eq (3.3), we have

$$(1 - \eta\tau)D_{\psi^{\mathcal{Z}}}(z_\tau^*, \widehat{z}_t) - D_{\psi^{\mathcal{Z}}}(z_\tau^*, \widehat{z}_{t+1}) - D_{\psi^{\mathcal{Z}}}(\widehat{z}_{t+1}, z_t) - \frac{7}{8}D_{\psi^{\mathcal{Z}}}(z_t, \widehat{z}_t) + \frac{1}{8}D_{\psi^{\mathcal{Z}}}(\widehat{z}_t, z_{t-1})$$

$$\ge \eta\tau\psi^{\mathcal{Z}}(z_t) - \eta\tau\psi^{\mathcal{Z}}(z_\tau^*) + \eta F(z_t)^\top(z_t - z_\tau^*) \overset{(i)}{\ge} 0, \tag{3.4}$$

where $(i)$ follows by definition of $z_\tau^*$.

Letting $\Theta_{t+1} = D_{\psi^{\mathcal{Z}}}(z_\tau^*, \widehat{z}_{t+1}) + D_{\psi^{\mathcal{Z}}}(\widehat{z}_{t+1}, z_t)$, inequality (3.4) can be written as

$$\Theta_{t+1} \le (1 - \eta\tau)\Theta_t - \frac{7}{8}D_{\psi^{\mathcal{Z}}}(z_t, \widehat{z}_t) - (\frac{7}{8} - \eta\tau)D_{\psi^{\mathcal{Z}}}(\widehat{z}_t, z_{t-1}) \le (1 - \eta\tau)\Theta_t \tag{3.5}$$

where the second inequality comes from $\eta\tau \le \eta \le \frac{7}{8}$. This justifies the linear convergence. □

In the existing work Wei et al. (2021); Lee et al. (2021) without regularization, i.e., when $\tau = 0$, the above argument cannot guarantee the linearly shrinking property of $\Theta_t$. With the unique NE assumption, one can prove some "slope" in the original bilinear objective which implies an explicit convergence rate (Lee et al., 2021, Lemma 15). It was unclear if such an assumption can be removed. Here, the regularization technique enables us to avoid such an assumption. See a more detailed and technical discussion below Lemma D.5.

## 3.2 From the Regularized Problem to the Original Problem

Intuitively, if the weight of regularization $\tau$ is sufficiently small, NE for the regularized problem should be close to the NE of the original problem (2.1). In the following we formalize this intuition and show how Theorem 3.1 implies a last-iterate guarantee.

We shrink the weight of regularization $\tau$ as follows: First initialize $\tau = \tau_0$ for some hyper-parameter $\tau_0$ at the beginning and run `Reg-DOMD` in episodes. In each episode, we update the parameters $z_t$ and $\widehat{z}_{t+1}$ for $\widetilde{\Theta}(1/\tau)$ iterations so that the duality gap of $\widehat{z}_t$ will be lower than $O(\tau)$ according to Lemma D.1 and Theorem 3.1. Then, we will shrink $\tau$ by one half and start the next episode from scratch. Notice that although $\tau$ is changing, the stepsize $\eta$ keeps fixed/constant, which differs from Hsieh et al. (2021), where the stepsize is *adaptive*.

**Theorem 3.2.** With the shrinking algorithm described above, the duality gap satisfies $\max_{z \in \mathcal{Z}} F(\widehat{z}_{t+1})^\top(\widehat{z}_{t+1} - z) \le \widetilde{O}(\frac{1}{t})$ for $t = 1, 2, ..., T$. Moreover, we have an iterate convergence rate of $\|\widehat{z}_{t+1} - \prod_{\mathcal{Z}^*}(\widehat{z}_{t+1})\| \le \widetilde{O}(\frac{1}{t})$.

In practice, we use an adaptive weight-shrinking rule proposed in Appendix A, which is motivated by Yang et al. (2020).

Note that Theorem 3.2 applies for both `Reg-DOMWU` and `Reg-DOGDA`. To the best of our knowledge, this is the first result to obtain convergence rate for duality gap and *the distance to the NE set* in EFGs without the unique NE assumption, when the mirror map is generated through a *dilated* operation (Lee et al., 2021).

**Technical overview.** We briefly sketch the intuition behind the proof and defer the full details to Appendix D. To prove the duality gap guarantee, first notice that in the regularized problem, $\hat{z}_t$ has a small duality gap thanks to the last-iterate guarantee in Theorem 3.1. So we only need to argue that the duality gap of $z_\tau^*$ in the original problem is also small, which turns out to be $O(\tau)$.

However, this argument does not imply a small distance to the NE set, because the distance between $z_\tau^*$ and $z^*$ is unknown. Instead, we need the result that the lower-bound of the "slope" of the duality gap is strictly positive, i.e., for any $z$, we have $\max_{z' \in \mathcal{Z}} F(z)^\top (z - z') \geq c\|z - \prod_{\mathcal{Z}^*}(z)\|$ for some constant $c > 0$. Moreover, compared to existing "slope" results (Gilpin et al., 2008; Wei et al., 2021), we provide a stronger one when the regularizer is entropy since we prove that $\max_{z'} F(z)^\top (z - z') \geq c\|z - \prod_{\mathcal{Z}^*}(z)\|$ when $z'$ is restricted to a subset of $\mathcal{Z}$ (see Lemma D.6).

Due to the regularization, our dependence on the EFG size $P$ is quite mild. There's only a $P\|\alpha\|_\infty$ dependence on the EFG size for the duality gap convergence result ($\|\alpha\|_\infty$ is usually $O(P^2)$ regarding the specific type of dilation (Hoda et al., 2010; Kroer et al., 2020; Farina et al., 2021)), which can be found in Appendix D. The convergence rate of the distance to the NE set of the original problem depends on the slope $c$, which also depends on the reward matrix.

## 4 REGULARIZED COUNTERFACTUAL REGRET MINIMIZATION (Reg-CFR)

*Counterfactual regret minimization* is the most widely used solution framework in EFGs in the past decades, and has achieved many successes including defeating the professional human player in Texas Hold'em (Brown and Sandholm, 2018; 2019b). Through the framework, the (global) regret of the EFG in (2.4) can be minimized by minimizing the *local regret* in each information set separately.

To describe the regret decomposition framework in its full generality, we first introduce some additional notation. $W^h(z)$ is the value at the treeplex rooted at information set $h$ of the player $h$ belongs to when both players play according to $z$. For any $h \in \mathcal{H}^\mathcal{X}$, $W^h(z)$ can be recursively defined as

$$W^h(z) = \sum_{i \in \Omega_h} q_i \big( (Ay)_i + \sum_{h' \in \mathcal{H}_i} W^{h'}(z) \big) + \tau \alpha_h \psi^\Delta(q_h)$$

where $q_i = z_i / z_{\sigma(h(i))} \in \Delta_{|\Omega_{h(i)}|}$ is the (conditional-form) strategy on information set $h(i)$ (it lies in a simplex due to the definition of treeplex in Definition 2.1) and $\alpha_h$ is the hyper-parameter defined in Eq (2.3). For $h \in \mathcal{H}^\mathcal{Y}$, $W^h(z)$ can be defined similarly.

The local loss $l_t^h(q_h) \colon \Delta_{|\Omega_h|} \to \mathbb{R}$ at any information set $h \in \mathcal{H}^\mathcal{Z}$ can be defined by

$$l_t^h(q_h) := \big\langle V^h(z_t), q_h \big\rangle + \tau \alpha_h \psi^\Delta(q_h), \quad \text{where } V^h(z) := \big( (Ay)_i + \sum_{h' \in \mathcal{H}_i} W^{h'}(z) \big)_{i \in \Omega_h}.$$

Notice that $W^h(z)$ is a scalar while $V^h(z)$ is a vector. Furthermore, the two quantities can be related to each other by $W^h(z) = \big\langle \frac{z_h}{z_{\sigma(h)}}, V^h(z) \big\rangle + \tau \alpha_h \psi^\Delta(\frac{z_h}{z_{\sigma(h)}})$.

The local difference at information set $h$ is just $G_T^h(q_h) := \sum_{t=1}^T l_t^h(q_{t,h}) - \sum_{t=1}^T l_t^h(q_h)$ and the local regret $R_T^h := \max_{\hat{q}_h \in \Delta_{|\Omega_h|}} G_T^h(\hat{q}_h)$. The following decomposition implies that the global regret can be controlled by the sum of local regrets:

**Lemma 4.1** (Laminar regret decomposition (Farina et al., 2019b)). For any $z_1, z_2, ..., z_T, z \in \mathcal{Z}$ and $\tau \geq 0$, we have

$$G_T^\mathcal{Z}(z) = \sum_{t=1}^T (F(z_t)^\top (z_t - z) + \tau \psi^\mathcal{Z}(z_t) - \tau \psi^\mathcal{Z}(z)) = \sum_{h \in \mathcal{H}^\mathcal{Z}} z_{\sigma(h)} G_T^h \big( \frac{z_h}{z_{\sigma(h)}} \big)$$

$$R_T^\mathcal{Z} = \max_{\hat{z} \in \mathcal{Z}} G_T^\mathcal{Z}(\hat{z}) \leq \max_{\hat{z} \in \mathcal{Z}} \sum_{h \in \mathcal{H}^\mathcal{Z}} \hat{z}_{\sigma(h)} R_T^h$$

(4.1)

where $R_T^\mathcal{Z}$ is the sum of the regret of min-player and max-player defined in Eq (2.4) instantiated with $l_t(z) = \langle F(z_t), z \rangle + \tau \psi^\mathcal{Z}(z)$. The proof is postponed to Appendix E. Hence, by minimizing $R_T^h$ at each information set $h \in \mathcal{H}^\mathcal{Z}$, $R_T^\mathcal{Z}$ will also be minimized. By Lemma 2.2, the average

strategy will converge to NE when $\tau = 0$. In fact, when $\tau > 0$, the average strategy will converge to the corresponding NE of the regularized problem $\boldsymbol{z}_\tau^*$ according to a stronger version of Lemma 2.2 (Theorem 3 ; Farina et al., 2019b). For completeness, we provide the formal version as Lemma F.3.

To describe our main results in full generality, we introduce the notion of perturbed EFGs before diving into the algorithm and analysis.

### 4.1 Perturbed extensive-form game and extensive-form perfect Nash equilibrium

Although NE specifies a natural notion of optimality in EFGs, an NE strategy is not necessarily behaving reasonably in information sets that it will not reach almost surely. To avoid this issue, a stronger and refined notion of equilibirum, *extensive-form perfect equilibria*, has been proposed in Selten (1975), which takes every information set into consideration by *perturbing* the EFG to force the players to reach every information set. We formally introduce the definitions below.

**Definition 4.2.** For any $\gamma \geq 0$, a $\gamma$-perturbed EFG is an EFG with a $\gamma$-perturbed treeplex $\mathcal{Z}^\gamma := \mathcal{X}^\gamma \times \mathcal{Y}^\gamma$ which restricts that $q_i = \frac{z_i}{z_{\sigma(h(i))}} \geq \gamma$ for any $\boldsymbol{z} \in \mathcal{Z}^\gamma$ and index $i$. An extensive-form perfect equilibrium is a limit point of $\{\boldsymbol{z}^{\gamma,*}\}_{\gamma \to 0}$ where $\boldsymbol{z}^{\gamma,*}$ is the NE of the $\gamma$-perturbed EFG.

The simplest instance of $\gamma$-perturbed treeplex is a $\gamma$-*perturbed probability simplex* $\Delta^\gamma$ where all entries have a probability larger than $\gamma$. Since the standard EFG is just a perturbed EFG with $\gamma = 0$, we will only describe our results in $\gamma$-perturbed EFG to keep the argument unified and general, and only translate our result to the $\gamma = 0$ case when necessary. Correspondingly, we use $\boldsymbol{z}_\tau^{\gamma,*}$ to denote the Nash equilibrium of the regularized game in Eq (3.1) when $(\boldsymbol{x}, \boldsymbol{y}) \in \mathcal{Z}^\gamma$. When $\gamma > 0$, $\boldsymbol{z}^{\gamma,*}$ is empirically used as an approximation to the EFPE (Kroer et al., 2017; Farina et al., 2017). We prove that $\boldsymbol{z}^{\gamma,*}$ could been seen as an approximation of EFPE in terms of duality gap (See Lemma F.4 for more details about this approximation, which might be of independent interest).

### 4.2 Main result

Given the regret decomposition in Lemma 4.1, we instantiate the regret minimizer in each information set by the regularized version of the Dual Stabilized Optimistic Mirror Descent algorithm (Hsieh et al., 2021), i.e., `Reg-DS-OptMD`. The DS-OptMD algorithm in (Hsieh et al., 2021) achieves constant regret in two player zero-sum NFGs, which to the best of our knowledge, is the state-of-the-art result that achieves this desired property. Hence, we develop our local regret minimizer based on this algorithm. For any information set $h \in \mathcal{H}^{\mathcal{Z}}$ and $t = 1, 2, ..., T$, the full update rule of our proposed algorithm, Regularized Counterfactual Regret Minimization (`Reg-CFR`), follows

$$\boldsymbol{q}_{t,h} = \operatorname*{argmin}_{\boldsymbol{q}_h \in \Delta_{|\Omega_h|}^\gamma} \left\langle V^h(\boldsymbol{z}_{t-\frac{1}{2}}) + \tau\alpha_h\nabla\psi^\Delta(\boldsymbol{q}_{t-1,h}), \boldsymbol{q}_h \right\rangle + \lambda_{t-1}^h D_{\psi^\Delta}(\boldsymbol{q}_h, \boldsymbol{q}_{t-1,h})$$

$$+ (\lambda_t^h - \lambda_{t-1}^h)D_{\psi^\Delta}(\boldsymbol{q}_h, \boldsymbol{q}_{1,h}) \quad (4.2)$$

$$\boldsymbol{q}_{t+\frac{1}{2},h} = \operatorname*{argmin}_{\boldsymbol{q}_h \in \Delta_{|\Omega_h|}^\gamma} \left\langle V^h(\boldsymbol{z}_{t-\frac{1}{2}}) + \tau\alpha_h\nabla\psi^\Delta(\boldsymbol{q}_{t,h}), \boldsymbol{q}_h \right\rangle + \lambda_t^h D_{\psi^\Delta}(\boldsymbol{q}_h, \boldsymbol{q}_{t,h}),$$

where the adaptive stepsize is defined by $\lambda_t^h := \sqrt{\kappa + \sum_{s=1}^{t-1} \delta_s^h}$ and $\kappa \geq 1$ is a hyper-parameter. $\delta_s^h := \|V^h(\boldsymbol{z}_{s+\frac{1}{2}}) - V^h(\boldsymbol{z}_{s-\frac{1}{2}})\|^2$ is the variation of value function and $\boldsymbol{q}_{t,h} = \frac{\boldsymbol{z}_{t,h}}{z_{t,\sigma(h)}}$. Again, for any $h \in \mathcal{H}^{\mathcal{Z}}$, $\boldsymbol{q}_{0,h} = \boldsymbol{q}_{\frac{1}{2},h}$ are intialized as uniform distribution in $\Delta_{|\Omega_h|}$. With the adaptive stepsize in `Reg-DS-OptMD`, we no longer need to tune the stepsize for each individual information set.

`Reg-CFR` enjoys a desirable last-iterate convergence guarantee of the actual iterate as follows:

**Theorem 4.3.** Consider the case when $\tau > 0$. In $\gamma$-perturbed EFGs, if we use Euclidean norm as the regularizer $\psi^\Delta$ in `Reg-CFR`, then $\sum_{t=1}^T D_{\psi^{\mathcal{Z}}}(\boldsymbol{z}_\tau^{\gamma,*}, \boldsymbol{z}_t) \leq \frac{C_\gamma}{\tau}$, where $C_\gamma$ is some positive variable depending on $\gamma$. As a result,

- When $\gamma > 0$ and $\tau \leq \frac{1}{2\|\boldsymbol{\alpha}\|_\infty}$, $C_\gamma$ is a constant which implies asymptotic last-iterate convergence to $\boldsymbol{z}_\tau^{\gamma,*}$ in terms of Bregman distance.

- When $\gamma = 0$, $C_\gamma \leq O(T^{1/4})$, implying a $O(T^{-3/4})$ best-iterate convergence rate to $\boldsymbol{z}_\tau^*$ in terms of Bregman distance.

To the best of our knowledge, under the regret decomposition framework, although some CFR-type algorithms, like CFR+ (Tammelin et al., 2015), have been empirically observed to have last-iterate convergence (Bowling et al., 2015), there is no theoretical justifications for them in the literature yet. Our results appear to be the first to establish the provable best- and last-iterate convergence results under the regret decomposition framework of CFR. Even in terms of empirical performance, our algorithm `Reg-CFR` can achieve faster last-iterate convergence rate comparing to previous ones. More interestingly, by applying regularization to CFR (Zinkevich et al., 2007) and CFR+ (Tammelin et al., 2015), we empirically show that regularization can improve the last-iterate performance. We will discuss them in Appendix B.

**Significance of last-iterate convergence for CFR.** We believe that Theorem 4.3 paves the way for more tractable CFR-type algorithms with function approximation in large-scale EFGs like Texas Hold'em. Previously, although Brown and Sandholm (2018; 2019b) achieved super-human level performance in Texas Hold'em, they utilized domain-specific abstraction techniques (Ganzfried and Sandholm, 2014; Brown et al., 2015), which will merge the similar nodes in Texas Hold'em into one to make the total number of nodes tractable. However, the existing abstraction methods are highly restricted to the poker games. Therefore, it is crucial to design algorithms with function approximation to do such abstraction in an end-to-end manner.

Currently, the *average-iterate* convergence of CFR is an obstacle to using function approximation. In the seminal work Deep-CFR (Brown et al., 2019), the authors trained an additional network to maintain the average policy, which caused additional approximation errors. In the subsequent work (Steinberger, 2019; Steinberger et al., 2020), to get the average policy, they stored the networks at every iteration on disk and sampled one randomly to follow. Though sampling successfully eliminates the additional approximation error, given that it takes at least $10^5$ iterations to converge in large poker games, storing all networks on disk is not tractable for large games like Texas Hold'em.

With Theorem 4.3, we can easily run CFR with function approximation since we only need to take the best model during learning due to the best-iterate guarantee[2].

A direct consequence of the theorem above is the following corollary.

**Corollary 4.4.** For any desired duality gap $\epsilon$, we can set $\tau = \Theta(\epsilon)$. The best-iterate convergence to the NE $\boldsymbol{z}^*$ when $\gamma = 0$ would be $O(T^{-1/4})$. When $\gamma > 0$, we will still have asymptotic last-iterate convergence to $\boldsymbol{z}^{*,\gamma}$, the NE of the $\gamma$-perturbed EFG, both in terms of duality gap.

**Remark 4.5** (Technical challenges in showing best-iterate convergence for CFR-type algorithms)**.** Although OMD achieves last-iterate convergence (Daskalakis and Panageas, 2019; Wei et al., 2021) and fast average-iterate convergence (Rakhlin and Sridharan, 2013; Syrgkanis et al., 2015), applying OMD as local regret minimizer in the CFR framework does not enjoy those results since the loss function for the regret minimizer depends on the global strategy in the treeplex which is not totally controlled by the local regret minimizer as in the NFGs. Therefore, the local regret minimizer could be seen as deployed in a changing environment where the previous results do not apply.

Moreover, as a by-product, we find that the average strategy produced by `Reg-CFR` is also superior comparing to the previous variants of CFR algorithms to our best knowledge. Notice that when picking $\tau = 0$, the algorithm will converge to the NE $\boldsymbol{z}^{*,\gamma}$ of the $\gamma$-perturbed EFG.

**Theorem 4.6.** Consider the case when $\tau \geq 0$ and the regularizer is Euclidean norm. In $\gamma$-perturbed EFGs with $\gamma > 0$ and $\tau \leq \frac{1}{2\|\boldsymbol{\alpha}\|_\infty}$, the average strategy output by `Reg-CFR` converges to $\boldsymbol{z}_\tau^{\gamma,*}$ with convergence rate $O(1/T)$, which is the optimal rate. In the original EFG with $\gamma = 0$, the average strategy output by `Reg-CFR` converges to $\boldsymbol{z}_\tau^*$ with convergence rate $O(1/T^{3/4})$.

To the best of our knowledge, `Reg-CFR` is the first CFR-type algorithm that achieves the theoretically optimal average-iterate convergence rate $O(1/T)$ when $\gamma > 0$ (for both $\tau > 0$ and $\tau = 0$). Furthermore, it maintains the current state-of-the-art average-iterate $O(1/T^{3/4})$ convergence rate established by Farina et al. (2019a) in the original EFG where $\gamma = 0$.

---

[2]In fact, we found that just taking the last iterate is good enough empirically. This part can be referred to Figure 3 and Figure 5.

ACKNOWLEDGEMENT

T.Y. was supported by NSF CCF-2112665 (TILOS AI Research Institute). A.O and K.Z. were supported by MIT-DSTA grant 031017-00016. K.Z. also acknowledges support from Simons-Berkeley Research Fellowship. The authors also thank Suvrit Sra for the valuable discussions.

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

# Supplementary Materials for
# "The Power of Regularization in Solving Extensive-Form Games"

## A  OMITTED DETAILS

Here we present some details omitted in the maintext.

### A.1  RELATED WORK

**Regularization.**  In reinforcement learning, regularization has been widely used to accelerate convergence and encourage exploration (Tuyls et al., 2003; Geist et al., 2019; Cen et al., 2021a; Mei et al., 2020). In game theory, regularization can be used to turn the bilinear objective in normal-form games into a strongly-convex-strongly-concave one (Hofbauer and Hopkins, 2005; Cen et al., 2021b). However, Hofbauer and Hopkins (2005) only gave asymptotic convergence to the NE of the regularized game under the best-response dynamics and Cen et al. (2021b) only provided convergence of OMWU to the original NE in terms of duality gap. Similar ideas could be dated back to the smoothing techniques led by Nesterov (2003). This way, the linear convergence rate to the saddle point of the new objective can be guaranteed. Letting the regularization be small, the solution to the regularized problem can be close to the NE of the original problem, in terms of duality gap (Cen et al., 2021b). In contrast, we aim to show the convergence in terms of not only the duality gap, but also the *distance to the NE set (of the original problem)*, and for the more complicated setting of EFGs. The idea of using regularization in learning in games has also been explored recently in various different settings (Perolat et al., 2021; Leonardos et al., 2021). Specifically, Perolat et al. (2021); Leonardos et al. (2021) study *continuous-time* dynamics and establish convergence to NE, either only gave rate to the NE of the regularized game, or only guaranteed asymptotic convergence to the NE of the original game, using techniques based on Lyapunov arguments. Instead, our focus was on discrete-time optimistic mirror-descent algorithms with constant stepsizes, with *convergence rates* for both duality gap and iterate-distance. Finally, we note that the framework of CFR (for solving EFGs) was not investigated in these recent works.

**Last-iterate convergence.**  Finding the NE in EFGs could be formulated as finding the saddle point of a bilinear objective function. While mirror descent diverges in simple cases (in terms of the last-iterate) (Mertikopoulos et al., 2018; Bailey and Piliouras, 2018), its optimistic version receives great success in finding the saddle point, enabling both faster and last-iterate convergence guarantees (Rakhlin and Sridharan, 2013; Mertikopoulos et al., 2019; Lei et al., 2021; Daskalakis et al., 2018; Mokhtari et al., 2020). However, these previous works either only consider the case without constraints (which do not apply to the NFG/EFG setting), or provide only asymptotic convergence without explicit rate. Recently, with the unique NE assumption, Daskalakis and Panageas (2019) gives an asymptotic last-iterate convergence result for OMWU in NFGs. Wei et al. (2021) further improves the result by showing that both OMWU and OGDA converge to the NE with a global sublinear convergence rate $O(1/T)$ and a local linear convergence rate in NFGs. Among them, OMWU requires the unique NE assumption. Very recently, Cai et al. (2022) provides a tight last-iterate convergence for OGDA. Finally, Lee et al. (2021) extends the result of OMWU from NFGs in Wei et al. (2021) to EFGs, and still requires the unique NE assumption. Concurrent to our submission, we are aware of Piliouras et al. (2022), which studies network zero-sum EFGs with last-iterate convergence rate guarantees, also without the unique NE assumption. However, the regularizer therein for the OMD update rule is neither dilated nor entropy-based, which makes the algorithm less scalable than the one we study, with dilated and entropy-based regularizer, see Lee et al. (2021) for a related discussion.

**Counterfactual regret minimization (CFR).**  CFR-type algorithms are based on the idea that the regret in an EFG could be decomposed into the local regret of each information set. By minimizing the local regret, the global regret will be minimized and the algorithms will achieve average-iterate convergence thereby. Recent work Farina et al. (2019a) utilizes the progress in the aforementioned optimistic methods, and achieves a faster average-iterate convergence rate of $O(1/T^{3/4})$ in EFGs.

However, since CFR-type methods rely on the regret decomposition that breaks the structure of the strategy, up to now no CFR (and variant) algorithms are able to inherit the optimal rate optimistic algorithms have enjoyed in NFGs, to the best of our knowledge. Also, due to the decomposition, although Bowling et al. (2015) has found that the last iterate of CFR+ (Tammelin et al., 2015), a variant of CFR, converges empirically, no CFR-type algorithm have the last-iterate convergence guarantee theoretically.

**Extensive-form perfect equilibrium and perturbed EFGs.** Nash equilibrium in EFGs does not have any guarantee at the places with zero probability to reach when all players follow the NE. Therefore, in reality when players make an error that leads to an impossible state in the NE, still following the NE may be suboptimal. The concept of extensive-form perfect equilibria has thus been proposed to resolve the issue (Selten, 1975). To find the EFPEs, Miltersen and Sørensen (2010); Farina and Gatti (2017) formulate the problem as a linear programming (LP), which is not tractable for large EFGs. Kroer et al. (2017) and Farina et al. (2017) extend the first-order method (Nesterov, 2005) and CFR to the perturbed extensive-form game (Selten, 1975) (which can be used for finding approximate EFPEs), where players have a small probability choosing to act randomly at every information set. Both of the results do not have last-iterate convergence.

## A.2 A GRAPHICAL ILLUSTRATION OF TREEPLEX

For better understanding of the structure of treeplex, we show the treeplex of the player who moves first in Kuhn Poker in Figure 1.

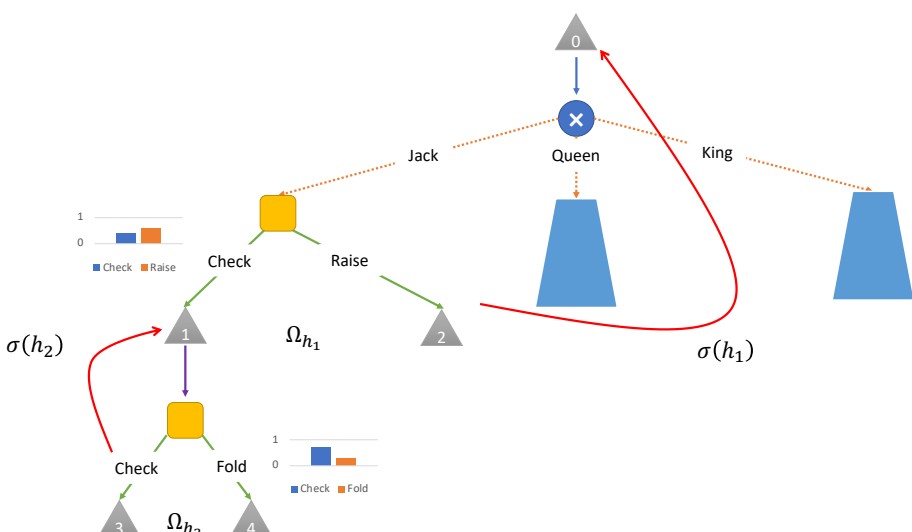

Figure 1: Treeplex of the player who moves first in Kuhn Poker, say player $x$. The blue circle denotes the chance node and the grey triangles denote the indices in $x$. This is the place where Cartesian product is applied to. And the squares denote the information sets of player $x$, which are the simplexes. The purple arrow is the place applied Branching once ($i = 1$). We omit the same structure as Jack under Queen & King. The dotted square represented the indices belongs to information set $h_1$ and $h_2$ and the red line represents the parent index of $h_1$ and $h_2$.

The treeplex is built up from 6 simplexes (2 each under different private card). Here's how the treeplex is built up.

- Branching: $h_1 \boxed{1} h_2$.
- Cartesian Product: Cartesian product of 3 similar treeplexes under Jack, Queen & King individually.

And the whole game tree of Kuhn Poker is shown in Figure 2.

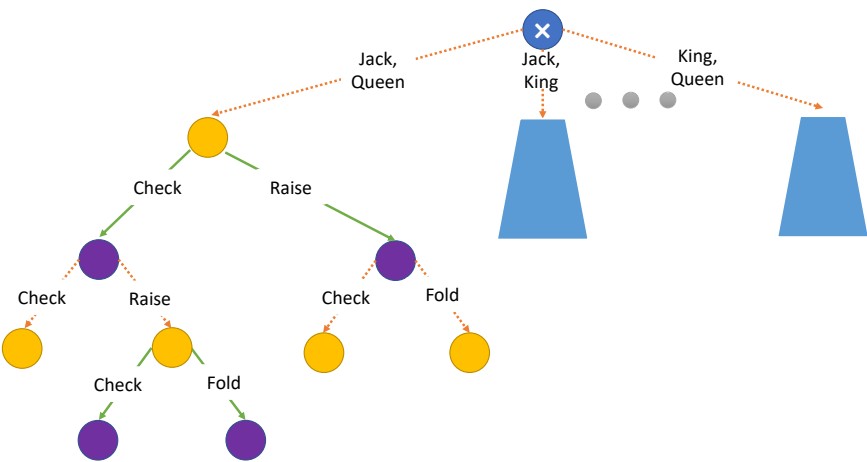

Figure 2: The full game tree of Kuhn Poker. The yellow nodes belong to the player who moves first and the purple nodes belong to the other player. The blue node is the chance node which dealt the private cards for each player. The first line is the private card for the player moving first and the second line is for the other player. The game tree under different private card composition are the same so we only plot the first-move player get `Jack` and the second-move player get `Queen`.

### A.3 PSEUDOCODE OF THE ADAPTIVE WEIGHT-SHRINKING ALGORITHM

Here's the practical version of adaptively shrinking $\tau$ framework mentioned in §3.2.

---

**Algorithm 1** Adaptive Weight-Shrinking

---

1: $\tau \leftarrow \tau_0$
2: $\delta_{\tau_0} \leftarrow \max_{\boldsymbol{z}'} F(\boldsymbol{z}_0)^\top (\boldsymbol{z}_0 - \boldsymbol{z}') + \tau_0 \psi^{\mathcal{Z}}(\boldsymbol{z}_0) - \tau_0 \psi^{\mathcal{Z}}(\boldsymbol{z}')$
3: $\boldsymbol{z}_0, \widehat{\boldsymbol{z}}_1 \leftarrow$ Uniform Strategy
4: **for** $t = 1, 2, ...$ **do**
5: $\quad \boldsymbol{z}_t, \widehat{\boldsymbol{z}}_{t+1} \leftarrow$ Reg-DOMD$(\boldsymbol{z}_{t-1}, \widehat{\boldsymbol{z}}_t)$
6: $\quad$ **if** $\max_{\boldsymbol{z}'} F(\widehat{\boldsymbol{z}}_t)^\top (\boldsymbol{z}_t - \boldsymbol{z}') + \tau \psi^{\mathcal{Z}}(\widehat{\boldsymbol{z}}_t) - \tau \psi^{\mathcal{Z}}(\boldsymbol{z}') \leq \frac{\delta_\tau}{4}$ **then**
7: $\quad\quad \tau \leftarrow \frac{\tau}{2}$
8: $\quad\quad \delta_\tau \leftarrow \max_{\boldsymbol{z}'} F(\widehat{\boldsymbol{z}}_t)^\top (\boldsymbol{z}_t - \boldsymbol{z}') + \tau \psi^{\mathcal{Z}}(\widehat{\boldsymbol{z}}_t) - \tau \psi^{\mathcal{Z}}(\boldsymbol{z}')$
9: $\quad\quad \boldsymbol{z}_t \leftarrow \widehat{\boldsymbol{z}}_{t+1}$
10: $\quad$ **end if**
11: **end for**

---

Notice that this framework can also be applied to `Reg-CFR` by simply changing `Reg-DOMD` to `Reg-CFR`.

### A.4 EXPERIMENT ENVIRONMENTS

**Kuhn Poker (Kuhn, 1950).** In Kuhn Poker, there are two players and three cards, `Jack`, `Queen` and `King`. And at the beginning, each player should place 1 chip into the pot and then 1 private card will be dealt to each player. And each player can call, raise or fold in each round. If a player call, then she should ensure that each player contributes equally to the pot. If a player raise, she should put 1 more chip in the pot than the other. If a player fold, then the other player takes all the chips in the pot. There will be at most 1 raise in the game. And a betting round ends when both players call or one of them fold.

After the game ends and nobody folds, the two players reveal their private cards and the one with higher rank takes all the chips in the pot.

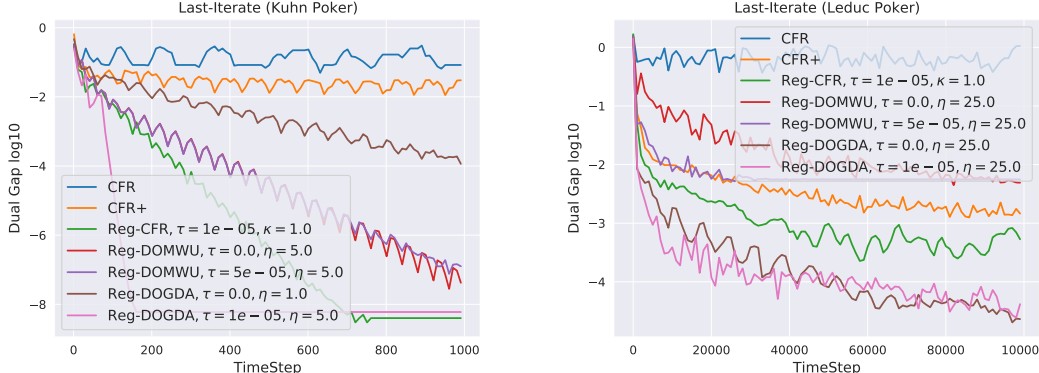

Figure 3: The last-iterate convergence result in Kuhn Poker (left) and Leduc Poker (right). CFR (Zinkevich et al., 2007), CFR+ (Tammelin et al., 2015) are tested as baselines. We can see that the last-iterate performance of `Reg-DOMWU` and `Reg-DOGDA` is much better than their versions when $\tau = 0$.

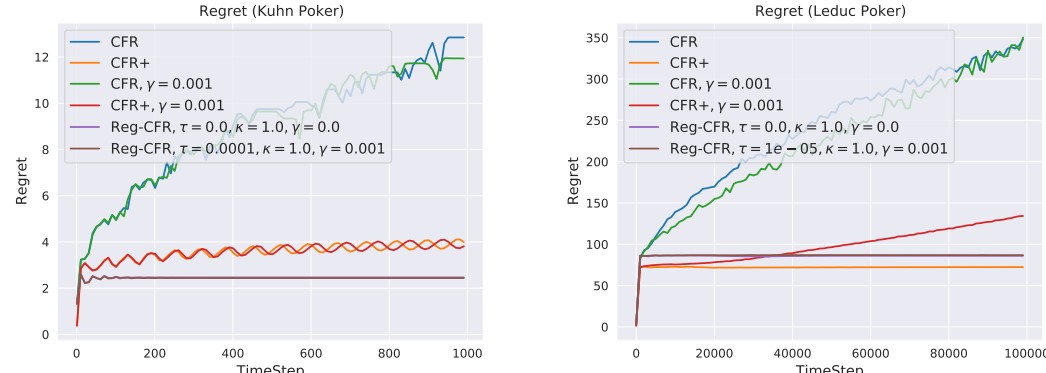

Figure 4: The regret upper-bound $\max_{\hat{z} \in \mathcal{Z}} \sum_{h \in \mathcal{H}^{\mathcal{Z}}} \hat{z}_{\sigma(h)} R_T^h$ in Kuhn Poker (left) and Leduc Poker (right). The regret of `Reg-CFR` is constant while that of CFR is increased in $O(\sqrt{T})$. The regret of CFR+ is much lower than $O(\sqrt{T})$ but not constant, which matches previous empirical result (Tammelin et al., 2015).

**Leduc Poker (Southey et al., 2005) .** Leduc Poker is similar to Kuhn Poker. It has 6 cards, three ranks ($\{J, Q, K\}$) with two suits ($\{a, b\}$) each. There are two betting rounds in Leduc Poker, each round admits two raises. The player who raises should place 1 more chip in the first round and 2 chips in the second. If the game ends and nobody folds, then the players reveal their private cards. The one who has the same private card as the public card wins. If nobody has the same private card as the public card, then the one with higher rank wins. Otherwise the game draws and the two players share the pot equally.

# B EXPERIMENT RESULTS

Beyond sharp theoretical guarantees, regularized algorithms in EFG also have superior performance in practice, which we showcase in this section through numerical experiments in Kuhn Poker (Kuhn, 1950) and Leduc Poker (Southey et al., 2005). The details of the experiment setup are illustrated in Appendix A.

The results are shown in Figure 3 for the last-iterate convergence in duality gap. We used grid search to find the best parameters for each algorithm. The algorithms `Reg-DOMWU`, `Reg-DOGDA`,

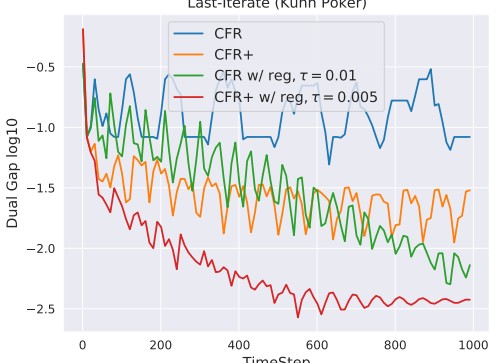 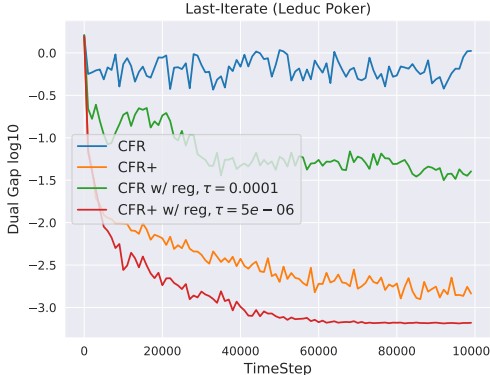

Figure 5: The last-iterate convergence results of CFR and CFR+, in Kuhn Poker (left) and Leduc Poker (right). We can see that with regularization, the last iterate produced by CFR and CFR+ significantly outperforms the original version without regularization.

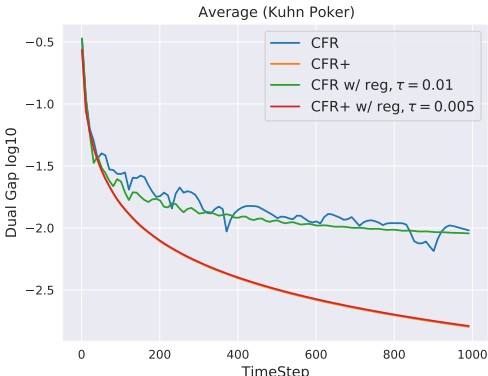 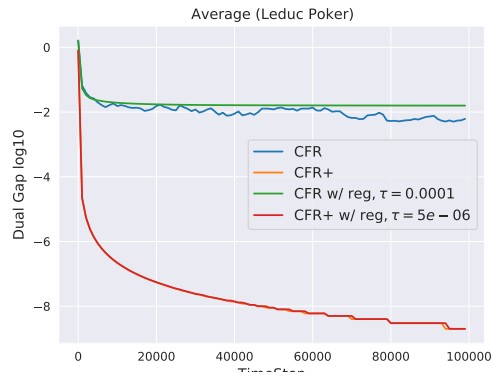

Figure 6: The average-iterate convergence results of CFR and CFR+, in Kuhn Poker (left) and Leduc Poker (right). We can see that by adding additional regularization, the average-iterate convergence is still competitive with the original version.

`Reg-CFR` all apply the adaptive weight-shrinking framework proposed as Algorithm 1 in Appendix A.

As shown in Figure 4, we show the regret upper-bound $\max_{\widehat{z} \in \mathcal{Z}} \sum_{h \in \mathcal{H}^{\mathcal{Z}}} \widehat{z}_{\sigma(h)} R_T^h$. We can see that `Reg-CFR` has *constant regret* even in a non-perturbed EFG in practice.

Moreover, we further empirically show that regularization is also helpful for CFR and CFR+. That is, with RM and RM+ as local regret minimizers, adding regularization still helps the algorithm enjoy last-iterate convergence. See Figure 5 for the details. To minimize the regret of a convex but non-linear loss function $l_t(\boldsymbol{x}_t)$, we feed $\langle \nabla l_t(\boldsymbol{x}_t), \boldsymbol{x}_t \rangle$ into RM and RM+ as the loss function. See (Farina et al., 2019d, §2.1) for more details. Moreover, the average-iterate convergence rate of this regularized version is still competitive with the original version. See Figure 6 for details.

Figure 7 illustrates the duality gap of average iterate. We can see that `Reg-CFR` is faster than CFR in both environments and has a comparable performance with CFR+ in smaller environments like Kuhn Poker.

Figure 8 illustrates the maximum cumulative regret across all information sets, conditioned on reaching that information set. This is also used as metric in Farina et al. (2017); Kroer et al. (2017). This metric can be used to measure the "closeness" to EFPEs. We can see that with $\gamma > 0$, `Reg-CFR` significantly outperforms `CFR` and `CFR+` in finding EFPEs.

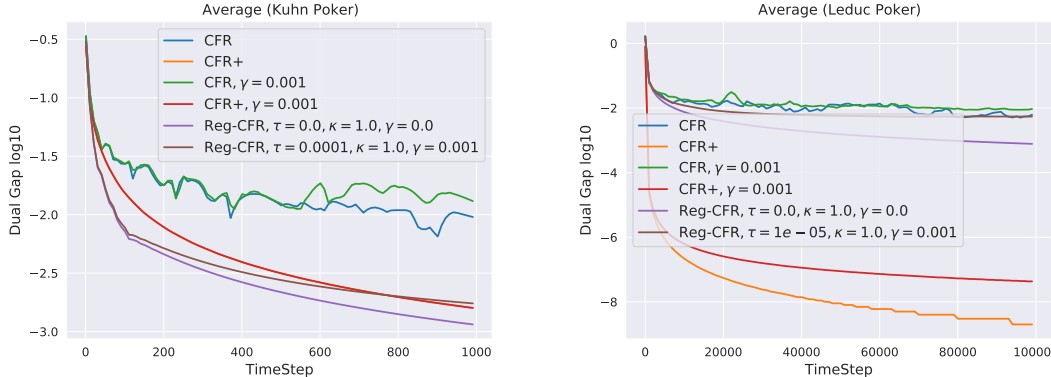

Figure 7: The duality gap of average iterate in Kuhn Poker (left) and Leduc Poker (right).

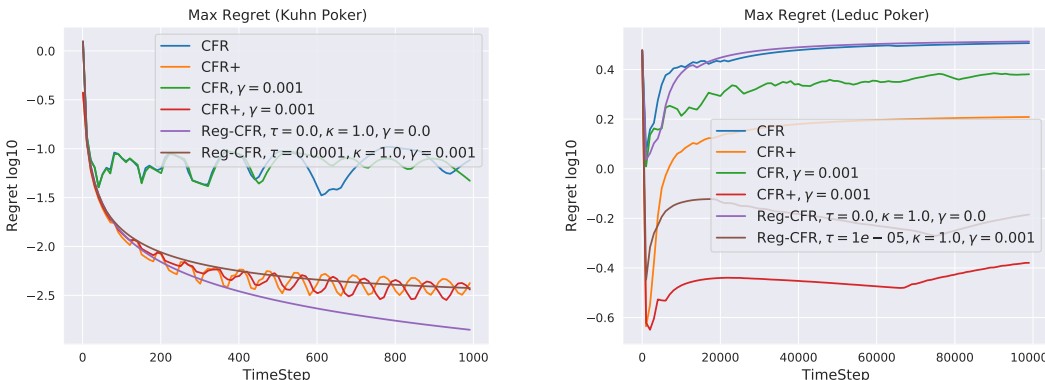

Figure 8: The maximum cumulative regret across all information sets, conditioned on reaching that information set. We test our algorithm in both Kuhn Poker (left) and Leduc Poker (right).

## C    PROOF OF THEOREM 3.1

**Lemma C.1.** For any $\tau \leq 1$ and $z \in \mathcal{Z}$, the NE of the regularized problem Eq (3.1) satisfies that

$$F(z)^{\top}(z - z_{\tau}^{*}) - \tau \psi^{\mathcal{Z}}(z_{\tau}^{*}) + \tau \psi^{\mathcal{Z}}(z) \geq 0. \tag{C.1}$$

**Lemma C.2.** Consider the update rule in Eq (3.2). When $\psi^{\mathcal{Z}}$ satisfies Eq (C.6) with $p = 2$ and $\eta \leq \frac{1}{8P}$, then for any $z \in \mathcal{Z}$ and $t \geq 1$, we have

$$\eta \tau \psi^{\mathcal{Z}}(z) - \eta \tau \psi^{\mathcal{Z}}(z_t) + \eta F(z_t)^{\top}(z_t - z) \leq (1 - \eta \tau) D_{\psi^{\mathcal{Z}}}(z, \widehat{z}_t) - D_{\psi^{\mathcal{Z}}}(z, \widehat{z}_{t+1})$$
$$- D_{\psi^{\mathcal{Z}}}(\widehat{z}_{t+1}, z_t) - \frac{7}{8} D_{\psi^{\mathcal{Z}}}(z_t, \widehat{z}_t) + \frac{1}{8} D_{\psi^{\mathcal{Z}}}(\widehat{z}_t, z_{t-1}).$$

*Proof of Theorem 3.1.* Taking $z = z_{\tau}^{*}$ in Lemma C.2, we have

$$(1 - \eta \tau) D_{\psi^{\mathcal{Z}}}(z_{\tau}^{*}, \widehat{z}_t) - D_{\psi^{\mathcal{Z}}}(z_{\tau}^{*}, \widehat{z}_{t+1}) - D_{\psi^{\mathcal{Z}}}(\widehat{z}_{t+1}, z_t) - \frac{7}{8} D_{\psi^{\mathcal{Z}}}(z_t, \widehat{z}_t) + \frac{1}{8} D_{\psi^{\mathcal{Z}}}(\widehat{z}_t, z_{t-1})$$

$$\geq \eta \tau \psi^{\mathcal{Z}}(z_t) - \eta \tau \psi^{\mathcal{Z}}(z_{\tau}^{*}) + \eta F(z_t)^{\top}(z_t - z_{\tau}^{*}) \overset{(i)}{\geq} 0, \tag{C.2}$$

where $(i)$ is by Lemma C.1.

Letting $\Theta_{t+1} = D_{\psi^z}(\boldsymbol{z}_\tau^*, \widehat{\boldsymbol{z}}_{t+1}) + D_{\psi^z}(\widehat{\boldsymbol{z}}_{t+1}, \boldsymbol{z}_t)$, inequality (C.2) can be written as

$$
\begin{aligned}
\Theta_{t+1} &\leq (1 - \eta\tau)\Theta_t - \frac{7}{8}D_{\psi^z}(\boldsymbol{z}_t, \widehat{\boldsymbol{z}}_t) - (\frac{7}{8} - \eta\tau)D_{\psi^z}(\widehat{\boldsymbol{z}}_t, \boldsymbol{z}_{t-1}) \\
&\leq (1 - \eta\tau)\Theta_t
\end{aligned} \tag{C.3}
$$

where the second inequality comes from $\eta\tau \leq \eta \leq \frac{7}{8}$.

As a result,

$$
D_{\psi^z}(\boldsymbol{z}_\tau^*, \widehat{\boldsymbol{z}}_{t+1}) \leq \Theta_{t+1} \leq (1 - \eta\tau)^t\Theta_1 = (1 - \eta\tau)^t D_{\psi^z}(\boldsymbol{z}_\tau^*, \widehat{\boldsymbol{z}}_1) \tag{C.4}
$$

where the last equation is satisfied when we initialize $\boldsymbol{z}_0 = \widehat{\boldsymbol{z}}_1$. $\square$

**Lemma C.3.** $F(\boldsymbol{z})$ is $P$-Lipschitz for any $\boldsymbol{z} \in \mathcal{Z}$. That is, for any $\boldsymbol{z}, \boldsymbol{z}' \in \mathcal{Z}$, we have

$$
\|F(\boldsymbol{z}) - F(\boldsymbol{z}')\| \leq P\|\boldsymbol{z} - \boldsymbol{z}'\|. \tag{C.5}
$$

*Proof.*

$$
\begin{aligned}
\|F(\boldsymbol{z}) - F(\boldsymbol{z}')\| = \sqrt{\|\boldsymbol{A}^\top(\boldsymbol{x} - \boldsymbol{x}')\|^2 + \|\boldsymbol{A}(\boldsymbol{y} - \boldsymbol{y}')\|^2} &\leq \sqrt{P\|\boldsymbol{x} - \boldsymbol{x}'\|_1^2 + P\|\boldsymbol{y} - \boldsymbol{y}'\|_1^2} \\
&\leq \sqrt{P\|\boldsymbol{z} - \boldsymbol{z}'\|_1^2} \\
&\leq P\|\boldsymbol{z} - \boldsymbol{z}'\|. \qquad \square
\end{aligned}
$$

**Lemma C.4.** Let $\mathcal{C}$ be a convex set and $\boldsymbol{u}_1 = \operatorname{argmin}_{\widehat{\boldsymbol{u}}_1 \in \mathcal{C}}\{\langle \widehat{\boldsymbol{u}}_1, \boldsymbol{g} + \tau\nabla\psi^{\mathcal{C}}(\boldsymbol{u})\rangle + \frac{1}{\eta}D_{\psi^{\mathcal{C}}}(\widehat{\boldsymbol{u}}_1, \boldsymbol{u})\}$ where $\psi^{\mathcal{C}}$ is a strongly-convex function in $\mathcal{C}$. Then for any $\boldsymbol{u}_2 \in \mathcal{C}, \tau \in [0, 1], \eta > 0$,

$$
\eta\tau\psi^{\mathcal{C}}(\boldsymbol{u}_1) - \eta\tau\psi^{\mathcal{C}}(\boldsymbol{u}_2) + \eta\langle\boldsymbol{u}_1 - \boldsymbol{u}_2, \boldsymbol{g}\rangle \leq (1 - \eta\tau)D_{\psi^{\mathcal{C}}}(\boldsymbol{u}_2, \boldsymbol{u}) - D_{\psi^{\mathcal{C}}}(\boldsymbol{u}_2, \boldsymbol{u}_1) - (1 - \eta\tau)D_{\psi^{\mathcal{C}}}(\boldsymbol{u}_1, \boldsymbol{u}).
$$

*Proof.* Plug in the definition of Bregman divergence $D_{\psi^{\mathcal{C}}}(\boldsymbol{u}_1, \boldsymbol{u}) = \psi^{\mathcal{C}}(\boldsymbol{u}_1) - \psi^{\mathcal{C}}(\boldsymbol{u}) - \langle\nabla\psi^{\mathcal{C}}(\boldsymbol{u}), \boldsymbol{u}_1 - \boldsymbol{u}\rangle$, the right-hand side of it is equal to,

$$
\begin{aligned}
&(1 - \eta\tau)D_{\psi^{\mathcal{C}}}(\boldsymbol{u}_2, \boldsymbol{u}) - D_{\psi^{\mathcal{C}}}(\boldsymbol{u}_2, \boldsymbol{u}_1) - (1 - \eta\tau)D_{\psi^{\mathcal{C}}}(\boldsymbol{u}_1, \boldsymbol{u}) \\
={}&(1 - \eta\tau)(\psi^{\mathcal{C}}(\boldsymbol{u}_2) - \psi^{\mathcal{C}}(\boldsymbol{u}) - \langle\nabla\psi^{\mathcal{C}}(\boldsymbol{u}), \boldsymbol{u}_2 - \boldsymbol{u}\rangle) \\
&+ (-\psi^{\mathcal{C}}(\boldsymbol{u}_2) + \psi^{\mathcal{C}}(\boldsymbol{u}_1) + \langle\nabla\psi^{\mathcal{C}}(\boldsymbol{u}_1), \boldsymbol{u}_2 - \boldsymbol{u}_1\rangle) \\
&+ (1 - \eta\tau)(-\psi^{\mathcal{C}}(\boldsymbol{u}_1) + \psi^{\mathcal{C}}(\boldsymbol{u}) + \langle\nabla\psi^{\mathcal{C}}(\boldsymbol{u}), \boldsymbol{u}_1 - \boldsymbol{u}\rangle) \\
={}&\eta\tau\psi^{\mathcal{C}}(\boldsymbol{u}_1) - \eta\tau\psi^{\mathcal{C}}(\boldsymbol{u}_2) + \langle\nabla\psi^{\mathcal{C}}(\boldsymbol{u}_1) - (1 - \eta\tau)\nabla\psi^{\mathcal{C}}(\boldsymbol{u}), \boldsymbol{u}_2 - \boldsymbol{u}_1\rangle \\
\overset{(i)}{\geq}{}&\eta\tau\psi^{\mathcal{C}}(\boldsymbol{u}_1) - \eta\tau\psi^{\mathcal{C}}(\boldsymbol{u}_2) + \eta\langle\boldsymbol{u}_1 - \boldsymbol{u}_2, \boldsymbol{g}\rangle,
\end{aligned}
$$

where $(i)$ is by the first order optimality of $\boldsymbol{u}_1$, i.e.,

$$
(\eta\boldsymbol{g} + \nabla\psi^{\mathcal{C}}(\boldsymbol{u}_1) - (1 - \eta\tau)\nabla\psi^{\mathcal{C}}(\boldsymbol{u}))^\top(\boldsymbol{u}_2 - \boldsymbol{u}_1) \geq 0. \qquad \square
$$

**Lemma C.5.** Suppose that $\psi^{\mathcal{C}}$ is a 1-strongly convex function with respect to $p$-norm in $\mathcal{C}$ such that

$$
D_{\psi^{\mathcal{C}}}(\boldsymbol{x}, \boldsymbol{x}') \geq \frac{1}{2}\|\boldsymbol{x} - \boldsymbol{x}'\|_p^2 \tag{C.6}
$$

for some $p \geq 1$, and $\boldsymbol{u}, \boldsymbol{u}_1, \boldsymbol{u}_2$ are members of a convex set $\mathcal{C}$ such that,

$$
\begin{aligned}
\boldsymbol{u}_1 &= \operatorname*{argmin}_{\boldsymbol{u}' \in \mathcal{C}}\{\langle\boldsymbol{u}', \boldsymbol{g}_1 + \tau\nabla\psi^{\mathcal{C}}(\boldsymbol{u})\rangle + D_{\psi^{\mathcal{C}}}(\boldsymbol{u}', \boldsymbol{u})\}, \\
\boldsymbol{u}_2 &= \operatorname*{argmin}_{\boldsymbol{u}' \in \mathcal{C}}\{\langle\boldsymbol{u}', \boldsymbol{g}_2 + \tau\nabla\psi^{\mathcal{C}}(\boldsymbol{u})\rangle + D_{\psi^{\mathcal{C}}}(\boldsymbol{u}', \boldsymbol{u})\}.
\end{aligned} \tag{C.7}
$$

Then we have,

$$
\|\boldsymbol{u}_1 - \boldsymbol{u}_2\|_p \leq \|\boldsymbol{g}_1 - \boldsymbol{g}_2\|_q, \tag{C.8}
$$

where $q \geq 1$ and $\frac{1}{p} + \frac{1}{q} = 1$.

*Proof.* By the first-order optimality of $\boldsymbol{u}_1, \boldsymbol{u}_2$, we have

$$\begin{aligned}(\boldsymbol{g}_1 + \nabla\psi^{\mathcal{C}}(\boldsymbol{u}_1) - (1-\tau)\nabla\psi^{\mathcal{C}}(\boldsymbol{u}))^{\top}(\boldsymbol{u}_2 - \boldsymbol{u}_1) &\geq 0, \\ (\boldsymbol{g}_2 + \nabla\psi^{\mathcal{C}}(\boldsymbol{u}_2) - (1-\tau)\nabla\psi^{\mathcal{C}}(\boldsymbol{u}))^{\top}(\boldsymbol{u}_1 - \boldsymbol{u}_2) &\geq 0.\end{aligned} \tag{C.9}$$

Summing up and rearranging the terms,

$$\langle \boldsymbol{u}_2 - \boldsymbol{u}_1, \boldsymbol{g}_1 - \boldsymbol{g}_2 \rangle \geq \langle \nabla\psi^{\mathcal{C}}(\boldsymbol{u}_1) - \nabla\psi^{\mathcal{C}}(\boldsymbol{u}_2), \boldsymbol{u}_1 - \boldsymbol{u}_2 \rangle. \tag{C.10}$$

To bound the right-hand side of inequality (C.10), By the lower bound of Bregman divergence (C.6), we have

$$\langle \nabla\psi^{\mathcal{C}}(\boldsymbol{u}_1), \boldsymbol{u}_1 - \boldsymbol{u}_2 \rangle \geq \psi^{\mathcal{C}}(\boldsymbol{u}_1) - \psi^{\mathcal{C}}(\boldsymbol{u}_2) + \frac{1}{2}\|\boldsymbol{u}_1 - \boldsymbol{u}_2\|_p^2,$$

$$\langle \nabla\psi^{\mathcal{C}}(\boldsymbol{u}_2), \boldsymbol{u}_2 - \boldsymbol{u}_1 \rangle \geq \psi^{\mathcal{C}}(\boldsymbol{u}_2) - \psi^{\mathcal{C}}(\boldsymbol{u}_1) + \frac{1}{2}\|\boldsymbol{u}_1 - \boldsymbol{u}_2\|_p^2.$$

Summing them up we have

$$\langle \nabla\psi^{\mathcal{C}}(\boldsymbol{u}_1) - \nabla\psi^{\mathcal{C}}(\boldsymbol{u}_2), \boldsymbol{u}_1 - \boldsymbol{u}_2 \rangle \geq \|\boldsymbol{u}_1 - \boldsymbol{u}_2\|_p^2.$$

Combining with inequality (C.10),

$$\langle \boldsymbol{u}_2 - \boldsymbol{u}_1, \boldsymbol{g}_1 - \boldsymbol{g}_2 \rangle \geq \|\boldsymbol{u}_1 - \boldsymbol{u}_2\|_p^2. \tag{C.11}$$

Finally, by Hölder's inequality,

$$\langle \boldsymbol{u}_2 - \boldsymbol{u}_1, \boldsymbol{g}_1 - \boldsymbol{g}_2 \rangle \leq \|\boldsymbol{u}_1 - \boldsymbol{u}_2\|_p \cdot \|\boldsymbol{g}_1 - \boldsymbol{g}_2\|_q,$$

and as a result $\|\boldsymbol{u}_1 - \boldsymbol{u}_2\|_p \leq \|\boldsymbol{g}_1 - \boldsymbol{g}_2\|_q$ as claimed. □

*Proof of Lemma C.1* By definition of NE, we have

$$\begin{aligned}F(\boldsymbol{z})^{\top}&(\boldsymbol{z} - \boldsymbol{z}_\tau^*) \\ =&(-\boldsymbol{x}_\tau^{*\top}\boldsymbol{A}\boldsymbol{y} + \boldsymbol{x}^{\top}\boldsymbol{A}\boldsymbol{y}_\tau^*) \\ =&\left(-\boldsymbol{x}_\tau^{*\top}\boldsymbol{A}\boldsymbol{y} + \tau\psi^{\mathcal{Z}}(\boldsymbol{y})\right) + \left(\boldsymbol{x}^{\top}\boldsymbol{A}\boldsymbol{y}_\tau^* + \tau\psi^{\mathcal{Z}}(\boldsymbol{x})\right) - \tau(\psi^{\mathcal{Z}}(\boldsymbol{x}) + \psi^{\mathcal{Z}}(\boldsymbol{y})) \\ \geq& -\boldsymbol{x}_\tau^{*\top}\boldsymbol{A}\boldsymbol{y}_\tau^* + \tau\psi^{\mathcal{Z}}(\boldsymbol{y}_\tau^*) + \boldsymbol{x}_\tau^{*\top}\boldsymbol{A}\boldsymbol{y}_\tau^* + \tau\psi^{\mathcal{Z}}(\boldsymbol{x}_\tau^*) - \tau(\psi^{\mathcal{Z}}(\boldsymbol{x}) + \psi^{\mathcal{Z}}(\boldsymbol{y})) \\ =& \tau\psi^{\mathcal{Z}}(\boldsymbol{z}_\tau^*) - \tau\psi^{\mathcal{Z}}(\boldsymbol{z}). \qquad \square\end{aligned}$$

*Proof of Lemma C.2.* Plug $\boldsymbol{u} = \widehat{\boldsymbol{z}}_t, \boldsymbol{u}_1 = \widehat{\boldsymbol{z}}_{t+1}, \boldsymbol{u}_2 = \boldsymbol{z}, \boldsymbol{g} = F(\boldsymbol{z}_t), \psi^{\mathcal{C}} = \psi^{\mathcal{Z}}$ into Lemma C.4,

$$\eta\tau\psi^{\mathcal{Z}}(\widehat{\boldsymbol{z}}_{t+1}) - \eta\tau\psi^{\mathcal{Z}}(\boldsymbol{z}) + \eta\langle\widehat{\boldsymbol{z}}_{t+1} - \boldsymbol{z}, F(\boldsymbol{z}_t)\rangle \leq (1-\eta\tau)D_{\psi^{\mathcal{Z}}}(\boldsymbol{z}, \widehat{\boldsymbol{z}}_t) - D_{\psi^{\mathcal{Z}}}(\boldsymbol{z}, \widehat{\boldsymbol{z}}_{t+1}) - (1-\eta\tau)D_{\psi^{\mathcal{Z}}}(\widehat{\boldsymbol{z}}_{t+1}, \widehat{\boldsymbol{z}}_t).$$

Plug $\boldsymbol{u} = \widehat{\boldsymbol{z}}_t, \boldsymbol{u}_1 = \boldsymbol{z}_t, \boldsymbol{u}_2 = \boldsymbol{z}_{t+1}, \boldsymbol{g} = F(\boldsymbol{z}_{t-1})$ and $\psi^{\mathcal{C}} = \psi^{\mathcal{Z}}$ into Lemma C.4,

$$\eta\tau\psi^{\mathcal{Z}}(\boldsymbol{z}_t) - \eta\tau\psi^{\mathcal{Z}}(\widehat{\boldsymbol{z}}_{t+1}) + \eta\langle\boldsymbol{z}_t - \widehat{\boldsymbol{z}}_{t+1}, F(\boldsymbol{z}_{t-1})\rangle \leq (1-\eta\tau)D_{\psi^{\mathcal{Z}}}(\widehat{\boldsymbol{z}}_{t+1}, \widehat{\boldsymbol{z}}_t) - D_{\psi^{\mathcal{Z}}}(\widehat{\boldsymbol{z}}_{t+1}, \boldsymbol{z}_t) - (1-\eta\tau)D_{\psi^{\mathcal{Z}}}(\boldsymbol{z}_t, \widehat{\boldsymbol{z}}_t).$$

Summing them up and adding $\langle F(\boldsymbol{z}_t) - F(\boldsymbol{z}_{t-1}), \boldsymbol{z}_t - \widehat{\boldsymbol{z}}_{t+1}\rangle$ to both sides, we have

$$\begin{aligned}\eta\tau\psi^{\mathcal{Z}}(\boldsymbol{z}_t) - \eta\tau\psi^{\mathcal{Z}}(\boldsymbol{z}) + \eta\langle F(\boldsymbol{z}_t), \boldsymbol{z}_t - \boldsymbol{z}\rangle \leq& (1-\eta\tau)D_{\psi^{\mathcal{Z}}}(\boldsymbol{z}, \widehat{\boldsymbol{z}}_t) - D_{\psi^{\mathcal{Z}}}(\boldsymbol{z}, \widehat{\boldsymbol{z}}_{t+1}) - D_{\psi^{\mathcal{Z}}}(\widehat{\boldsymbol{z}}_{t+1}, \boldsymbol{z}_t) \\ &- (1-\eta\tau)D_{\psi^{\mathcal{Z}}}(\boldsymbol{z}_t, \widehat{\boldsymbol{z}}_t) + \eta\langle F(\boldsymbol{z}_t) - F(\boldsymbol{z}_{t-1}), \boldsymbol{z}_t - \widehat{\boldsymbol{z}}_{t+1}\rangle.\end{aligned}$$

It remains to bound the last term, which is

$$\begin{aligned}\eta\langle F&(\boldsymbol{z}_t) - F(\boldsymbol{z}_{t-1}), \boldsymbol{z}_t - \widehat{\boldsymbol{z}}_{t+1}\rangle \\ &\overset{(i)}{\leq} \eta\|\boldsymbol{x}_t - \widehat{\boldsymbol{x}}_{t+1}\| \cdot \|\eta\boldsymbol{A}\boldsymbol{y}_t - \eta\boldsymbol{A}\boldsymbol{y}_{t-1}\| + \eta\|\boldsymbol{y}_t - \widehat{\boldsymbol{y}}_{t+1}\| \cdot \|\eta\boldsymbol{A}\boldsymbol{x}_t - \eta\boldsymbol{A}\boldsymbol{x}_{t-1}\| \\ &\overset{(ii)}{\leq} \eta^2(\|\boldsymbol{A}\boldsymbol{y}_t - \boldsymbol{A}\boldsymbol{y}_{t-1}\|^2 + \|\boldsymbol{A}\boldsymbol{x}_t - \boldsymbol{A}\boldsymbol{x}_{t-1}\|^2) \\ &\overset{(iii)}{\leq} 2\eta^2 P^2\|\boldsymbol{z}_t - \boldsymbol{z}_{t-1}\|^2 \\ &\overset{(iv)}{\leq} \frac{1}{32}\|\boldsymbol{z}_t - \boldsymbol{z}_{t-1}\|^2 \leq \frac{1}{16}(\|\boldsymbol{z}_t - \widehat{\boldsymbol{z}}_t\|^2 + \|\widehat{\boldsymbol{z}}_t - \boldsymbol{z}_{t-1}\|^2) \leq \frac{1}{8}(D_{\psi^{\mathcal{Z}}}(\boldsymbol{z}_t, \widehat{\boldsymbol{z}}_t) + D_{\psi^{\mathcal{Z}}}(\widehat{\boldsymbol{z}}_t, \boldsymbol{z}_{t-1}))\end{aligned}$$

where $(i)$ is by Hölder's inequality, $(ii)$ is by Lemma C.5 with $p = q = 2$, $(iii)$ is by Lemma C.3, and $(iv)$ is by $\eta \leq \frac{1}{8P}$.

The proof of the claim is completed by putting everything together. □

# D   PROOF OF THEOREM 3.2

Firstly, we will prove that the approximate NE of the regularized problem is close to the NE of the original problem in terms of duality gap.

**Lemma D.1.** For any $\tau > 0$ and $z \in \mathcal{Z}$, we have

$$\max_{\widehat{z} \in \mathcal{Z}} F(z)^\top (z - \widehat{z}) \leq 2\tau C_B + 2P\sqrt{D_{\psi^{\mathcal{Z}}}(z_\tau^*, z)}, \tag{D.1}$$

where $C_B$ is the upper-bound of the regularizer $\psi^{\mathcal{Z}}$.

*Proof.*

$$
\begin{aligned}
\max_{\widehat{z} \in \mathcal{Z}} F(z)^\top (z - \widehat{z}) =& \max_{\widehat{z} \in \mathcal{Z}} \{ x_\tau^{*\top} A\widehat{y} - \widehat{x}^\top Ay_\tau^* + \tau\psi^{\mathcal{Z}}(z_\tau^*) - \tau\psi^{\mathcal{Z}}(\widehat{z}) \\
& - \tau\psi^{\mathcal{Z}}(z_\tau^*) + \tau\psi^{\mathcal{Z}}(\widehat{z}) + (x - x_\tau^*)^\top A\widehat{y} + \widehat{x}^\top A(y_\tau^* - y) \} \\
\leq& \max_{\widehat{z} \in \mathcal{Z}} \{ x_\tau^{*\top} A\widehat{y} - \widehat{x}^\top Ay_\tau^* + \tau\psi^{\mathcal{Z}}(z_\tau^*) - \tau\psi^{\mathcal{Z}}(\widehat{z}) \} \\
& + \max_{\widehat{z} \in \mathcal{Z}} \{ -\tau\psi^{\mathcal{Z}}(z_\tau^*) + \tau\psi^{\mathcal{Z}}(\widehat{z}) + (x - x_\tau^*)^\top A\widehat{y} + \widehat{x}^\top A(y_\tau^* - y) \} \\
\overset{(i)}{\leq}& 0 + 2\tau C_B + \|x - x_\tau^*\|_1 + \|y_\tau^* - y\|_1 \\
\overset{(ii)}{\leq}& 2\tau C_B + 2P\|z - z_\tau^*\| \\
\leq& 2\tau C_B + 2P\sqrt{D_{\psi^{\mathcal{Z}}}(z_\tau^*, z)}
\end{aligned}
\tag{D.2}
$$

where $(i)$ is because of the definition of $z_\tau^*$ and $\|F(z)\|_\infty \leq 1$ for any $z \in \mathcal{Z}$. $(ii)$ is by $\|x - x_\tau^*\|_1 \leq \sqrt{P}\|x - x_\tau^*\|$, $\|y - y_\tau^*\|_1 \leq \sqrt{P}\|y - y_\tau^*\|$ and $a + b \leq 2\sqrt{a^2 + b^2}$. $C_B$ is the upper-bound of the regularizer $\psi^{\mathcal{Z}}$. It would be $P\|\alpha\|_\infty \log C_\Omega$ for entropy regularizer and $\frac{P\|\alpha\|_\infty}{C_\Omega}$ for Euclidean regularizer, where $C_\Omega = \max_{h \in \mathcal{H}^{\mathcal{Z}}} |\Omega_h|$. $\square$

A direct consequence of the lemma is that for any $\epsilon > 0$, we can set $\tau = \frac{\epsilon}{4C_B}$, then after $\frac{2(\log \epsilon - \log 4P) - \log D_{\psi^{\mathcal{Z}}}(z_\tau^*, \widehat{z}_1)}{\log(1-\tau)} \leq \frac{-2(\log \epsilon - \log 4P) + \log D_{\psi^{\mathcal{Z}}}(z_\tau^*, \widehat{z}_1)}{\tau}$ iterations, $\widehat{z}_t$ produced by `Reg-DOMD` will satisfies that

$$\max_{z \in \mathcal{Z}} \{\widehat{x}_t^\top Ay - x^\top A\widehat{y}_t\} \leq \frac{\epsilon}{2} + 2P\sqrt{\frac{\epsilon^2}{16P^2 D_{\psi^{\mathcal{Z}}}(z_\tau^*, \widehat{z}_1)} D_{\psi^{\mathcal{Z}}}(z_\tau^*, \widehat{z}_1)} \leq \epsilon \tag{D.3}$$

by Theorem 3.1.

*Proof of Theorem 3.2.* **Sublinear convergence rate of duality gap.**

For any $\epsilon$, the number of iterations that the duality gap reach $\epsilon$ is no larger than $4C_B \frac{-2(\log \epsilon - \log 4P) + \log D_{\psi^{\mathcal{Z}}}(\widehat{z}_\tau^*, \widehat{z}_1)}{\epsilon}$ by the discussion above. Therefore, while duality gap reaching $\epsilon = \frac{\epsilon_0}{2^K}$, the number of iterations performed so far is no larger than

$$
\begin{aligned}
& \sum_{k=0}^{K} 4C_B \cdot 2^k \frac{-2\log \epsilon_0 + 2k\log 2 + 2\log 4P + \log D_{\psi^{\mathcal{Z}}}(z_\tau^*, \widehat{z}_1)}{\epsilon_0} \\
\leq& 4C_B 2^{K+2} \frac{-\log \epsilon_0 + K\log 2 + \log 4P + \log D_{\psi^{\mathcal{Z}}}(z_\tau^*, \widehat{z}_1)}{\epsilon_0} \\
=& \widetilde{O}(1/\epsilon).
\end{aligned}
\tag{D.4}
$$

**Iterate convergence.**

From the proof of Theorem 5 in Wei et al. (2021), we have the following lemma.

**Lemma D.2** (Proved in Theorem 5 of Wei et al. (2021)). Consider a bilinear zero-sum game. Let $\rho := \min_{x \in \mathcal{X}} \max_{y \in \mathcal{Y}} x^\top A y$ be the game value. When $\mathcal{X}, \mathcal{Y}$ are polytopes, we have $\max_{\widehat{y} \in \mathcal{Y}} x^\top A \widehat{y} - \rho \geq c \|x - \prod_{\mathcal{X}^*}(x)\|$ $(\rho - \min_{\widehat{x} \in \mathcal{X}} \widehat{x}^\top A y \geq c \|y - \prod_{\mathcal{Y}^*}(y)\|)$ for some constant $c > 0$ where $\prod_{\mathcal{X}^*}(x)$ $(\prod_{\mathcal{Y}^*}(y))$ is the projection of $x$ $(y)$ to the NE set $\mathcal{X}^*$ $(\mathcal{Y}^*)$ of the min-player (max-player).

Then, since the treeplex is a polytope by definition, we have

$$
\begin{aligned}
\max_{z \in \mathcal{Z}} F(\widehat{z}_t)^\top (\widehat{z}_t - z) &= \max_{y \in \mathcal{Y}} \widehat{x}_t^\top A y - \min_{x \in \mathcal{X}} x^\top A \widehat{y}_t \\
&\geq c(\|\widehat{x}_t - \prod_{\mathcal{X}^*}(\widehat{x}_t)\| + \|\widehat{y}_t - \prod_{\mathcal{Y}^*}(\widehat{y}_t)\|) \\
&\geq c\|\widehat{z}_t - \prod_{\mathcal{Z}^*}(\widehat{z}_t)\|
\end{aligned}
\tag{D.5}
$$

where the last inequality comes from $\sqrt{a+b} \leq \sqrt{a} + \sqrt{b}$. Therefore, $\|\widehat{z}_t - \prod_{\mathcal{Z}^*}(\widehat{z}_t)\| \leq \frac{1}{c} \max_{z \in \mathcal{Z}} F(\widehat{z}_t)^\top (z_t - z) \leq \widetilde{O}(\frac{1}{t})$. $\square$

Notice that comparing to the results in Gilpin et al. (2008); Wei et al. (2021), our slope result (Lemma D.6) is based on different techniques. In Lemma D.6, we prove that $\max_{\widehat{y} \in V^*(\prod_{\mathcal{X}^*}(x))} x^\top A \widehat{y} - \rho \geq c_x \|x - \prod_{\mathcal{X}^*}(x)\|$ where $V^*(\prod_{\mathcal{X}^*}(x)) \subseteq \mathcal{Y}$ when $x \in \mathcal{F}_x$ and $\mathcal{F}_x \subseteq \mathcal{X}$ contains all possible iterates generated by DOMWU. That is, our result is stronger than the existing results, when the algorithm is DOMWU[3]. Moreover, our result can be viewed as an extension of (Lee et al., 2021, Lemma 14) to the non-unique NE cases. Given that (Lee et al., 2021, Lemma 14) plays an critical role in proving the last-iterate convergence with unique NE assumption, Lemma D.6 may be useful when proving last-iterate convergence in EFGs without unique NE assumption and regularization.

## D.1 COMPLEMENTARY SLACKNESS

This part of discussion is similar to the one in Lee et al. (2021). From Definition 2.1, we have

$$
\forall h \in \mathcal{H}^\mathcal{Y}, \quad \sum_{i \in \Omega_h} y_i = y_{\sigma(h)}, \qquad y_0 = 1
\tag{D.6}
$$

which can be written compactly as $E_\mathcal{Y} y = e_\mathcal{Y}$ where $E_\mathcal{Y} \in \mathbb{R}^{(|\mathcal{H}^\mathcal{Y}|+1) \times N}$ and $e_\mathcal{Y} = (1, 0, 0, ..., 0) \in \mathbb{R}^{|\mathcal{H}^\mathcal{Y}|+1}$. Except the first row of $E_\mathcal{Y}$ where there's 1 on index 0 and 0 otherwise, all other rows have 1 on index $\sigma(h)$ and $-1$ on all $i \in \Omega_h$. Therefore, for any fixed $x$, the objective of $y$ can be written as

$$
\begin{aligned}
\max_{y \in \mathcal{Y}} \quad & x^\top A y \\
\text{s.t.} \quad & E_\mathcal{Y} y = e_\mathcal{Y}, \quad y \geq 0
\end{aligned}
\tag{D.7}
$$

whose dual problem is

$$
\begin{aligned}
\min_{g} \quad & e_\mathcal{Y}^\top g \\
\text{s.t.} \quad & E_\mathcal{Y}^\top g \geq A^\top x
\end{aligned}
\tag{D.8}
$$

where $e_\mathcal{Y}^\top g = g_0$ since $e_\mathcal{Y} = (1, 0, 0, ..., 0)$.

Remind that the primal formulation of the original problem is

$$
\begin{aligned}
\min_{x \in \mathcal{X}} \max_{y \in \mathcal{Y}} \quad & x^\top A y \\
\text{s.t.} \quad & E_\mathcal{X} x = e_\mathcal{X}, \quad x \geq 0 \\
& E_\mathcal{Y} y = e_\mathcal{Y}, \quad y \geq 0.
\end{aligned}
\tag{D.9}
$$

---

[3]In fact, here we only require that the regularization is entropy to make Lemma D.7 hold.

Therefore, every solution $y^*$ of the original problem would be a solution of the following problem.

$$\min_{\boldsymbol{x} \in \mathcal{X}, \boldsymbol{g}} \; g_0$$
$$\text{s.t. } \boldsymbol{E}_{\mathcal{Y}}^\top \boldsymbol{g} \geq \boldsymbol{A}^\top \boldsymbol{x} \quad \boldsymbol{E}_{\mathcal{X}} \boldsymbol{x} = \boldsymbol{e}_{\mathcal{X}} \quad \boldsymbol{x} \geq 0. \tag{D.10}$$

The dual of this one is

$$\max_{\boldsymbol{y} \in \mathcal{Y}, \boldsymbol{f}} \; f_0$$
$$\text{s.t. } \boldsymbol{E}_{\mathcal{X}}^\top \boldsymbol{f} \leq \boldsymbol{A} \boldsymbol{y} \quad \boldsymbol{E}_{\mathcal{Y}} \boldsymbol{y} = \boldsymbol{e}_{\mathcal{Y}} \quad \boldsymbol{y} \geq 0. \tag{D.11}$$

Note that $\mathcal{X}^*, \mathcal{Y}^*$ are the optimal solution of Eq (D.10) and Eq (D.11). By complementary slackness, for any optimal solution pair $(\boldsymbol{x}^*, \boldsymbol{g}^*), (\boldsymbol{y}^*, \boldsymbol{f}^*)$, we have slackness variables $\boldsymbol{w}^* \in \mathbb{R}^M, \boldsymbol{s}^* \in \mathbb{R}^N$ so that

$$\boldsymbol{E}_{\mathcal{X}}^\top \boldsymbol{f} + \boldsymbol{w}^* = \boldsymbol{A} \boldsymbol{y} \qquad \boldsymbol{E}_{\mathcal{Y}}^\top \boldsymbol{g} - \boldsymbol{s}^* = \boldsymbol{A}^\top \boldsymbol{x}$$
$$\boldsymbol{x}^* \odot \boldsymbol{w}^* = 0 \quad \boldsymbol{y}^* \odot \boldsymbol{s}^* = 0 \qquad \boldsymbol{w}^* \geq 0 \quad \boldsymbol{s}^* \geq 0 \tag{D.12}$$

where $\odot$ denotes the element-wise product.

As a direct consequence, we have the following lemma.

**Lemma D.3.** For any optimal solution pair $(\boldsymbol{x}^*, \boldsymbol{g}^*), (\boldsymbol{y}^*, \boldsymbol{f}^*)$ of Eq (D.10) and Eq (D.11), we have

$$\sum_{h \in \mathcal{H}_i} f_h^* + (\boldsymbol{A} \boldsymbol{y}^*)_i = f_{h(i)}^* \quad \forall i \in \text{supp}(\mathcal{X}^*)$$

$$\sum_{h \in \mathcal{H}_i} f_h^* + (\boldsymbol{A} \boldsymbol{y}^*)_i \geq f_{h(i)}^* \quad \forall i \notin \text{supp}(\mathcal{X}^*)$$

$$\sum_{h \in \mathcal{H}_i} g_h^* + (\boldsymbol{A}^\top \boldsymbol{x}^*)_i = g_{h(i)}^* \quad \forall i \in \text{supp}(\mathcal{Y}^*) \tag{D.13}$$

$$\sum_{h \in \mathcal{H}_i} g_h^* + (\boldsymbol{A}^\top \boldsymbol{x}^*)_i \leq g_{h(i)}^* \quad \forall i \notin \text{supp}(\mathcal{Y}^*)$$

where $\text{supp}(\boldsymbol{x})$ denotes the support set of vector $\boldsymbol{x}$ and $\text{supp}(\mathcal{C}) = \bigcup_{\boldsymbol{x} \in \mathcal{C}} \text{supp}(\boldsymbol{x})$ denotes the support set of a convex set $\mathcal{C}$.

*Proof.* Since $(\boldsymbol{E}_{\mathcal{X}}^\top \boldsymbol{f})_i = f_{h(i)}^* - \sum_{h \in \mathcal{H}_i} f_h^*$ by definition of $\boldsymbol{E}$, from Eq (D.12), we have

$$\sum_{h \in \mathcal{H}_i} f_h^* + (\boldsymbol{A} \boldsymbol{y}^*)_i = w_i^* + f_{h(i)}^* \geq f_{h(i)}^*. \tag{D.14}$$

For any $i$ where there's $\boldsymbol{x}^* \in \mathcal{X}^*$ and $x_i^* > 0$, from $\boldsymbol{x}^* \odot \boldsymbol{w}^* = 0$, we have $w_i^* = 0$. Thus, the above inequality takes the equality. So the first two lines of Lemma D.3 are proved. Similarly, we can prove the last two lines. $\square$

We further introduce the following definitions.

**Definition D.4.**

$$\rho = \boldsymbol{x}^{*\top} \boldsymbol{A} \boldsymbol{y}^*$$
$$PS(\boldsymbol{x}^*) = \{\boldsymbol{y} : \boldsymbol{y} \text{ is a pure strategy, } \boldsymbol{x}^{*\top} \boldsymbol{A} \boldsymbol{y} = \rho\}$$
$$PS(\boldsymbol{y}^*) = \{\boldsymbol{x} : \boldsymbol{x} \text{ is a pure strategy, } \boldsymbol{x}^\top \boldsymbol{A} \boldsymbol{y}^* = \rho\}$$
$$V^*(\boldsymbol{x}^*) = \mathcal{C}(PS(\boldsymbol{x}^*)) \tag{D.15}$$
$$V^*(\boldsymbol{y}^*) = \mathcal{C}(PS(\boldsymbol{y}^*))$$
$$\text{supp}(\boldsymbol{x}) = \{i : x_i > 0\}$$
$$\text{supp}(\mathcal{C}) = \{i : \exists \boldsymbol{x} \in \mathcal{C}, \; x_i > 0\}$$

where $\mathcal{C}(S)$ denotes the minimum convex set covering all points in $S$.

A fact from the definition is that $\forall \boldsymbol{y} \in V^*(\boldsymbol{x}^*), \boldsymbol{x}^{*\top} \boldsymbol{A} \boldsymbol{y} = \rho$ and $\forall \boldsymbol{x} \in V^*(\boldsymbol{y}^*), \boldsymbol{x}^\top \boldsymbol{A} \boldsymbol{y}^* = \rho$.

**Lemma D.5.** $V^*(\boldsymbol{x}^*), V^*(\boldsymbol{y}^*)$ are not empty for any $\boldsymbol{x}^* \in \mathcal{X}^*, \boldsymbol{y}^* \in \mathcal{Y}^*$.

*Proof.* For any $\boldsymbol{x} \in \mathcal{X}, \boldsymbol{y}^* \in \mathcal{Y}^*, \boldsymbol{f}^*$ so that $\mathrm{supp}(\boldsymbol{x}) \subseteq \mathrm{supp}(\mathcal{X}^*)$ and $(\boldsymbol{f}^*, \boldsymbol{y}^*)$ is a pair of optimal solution of Eq (D.11), we have

$$
\begin{aligned}
\boldsymbol{x}^\top \boldsymbol{A} \boldsymbol{y}^* &= \sum_i x_i (\boldsymbol{A}\boldsymbol{y}^*)_i \\
&= \sum_i x_i (f^*_{h(i)} - \sum_{h \in \mathcal{H}_i} f^*_h) \\
&= \sum_{h \in \mathcal{H}^\mathcal{X}} f^*_h \sum_{i \in \Omega_h} x_i - \sum_{h \in \mathcal{H}^\mathcal{X}, h \neq 0} f^*_h x_{\sigma(h)} \\
&= \sum_{h \in \mathcal{H}^\mathcal{X}} f^*_h x_{\sigma(h)} - \sum_{h \in \mathcal{H}^\mathcal{X}, h \neq 0} f^*_h x_{\sigma(h)} \\
&= f^*_0 = \rho
\end{aligned}
\tag{D.16}
$$

where the second equality is because $\mathrm{supp}(\boldsymbol{x}) \subseteq \mathrm{supp}(\mathcal{X}^*)$ and Lemma D.3. The fourth equality comes from the fact that $\sum_{i \in \Omega_h} x_i = x_{\sigma(h)}$. Therefore, $V^*(\boldsymbol{y}^*)$ is not empty for any $\boldsymbol{y}^* \in \mathcal{Y}^*$. Similarly, $V^*(\boldsymbol{x}^*)$ is not empty for any $\boldsymbol{x}^* \in \mathcal{X}^*$. $\qquad\square$

When assuming unique NE as in Lee et al. (2021), the second line and the fourth line in Lemma D.3 will be strictly larger than and strictly less than by strict complementary slackness. The discussion in Lemma D.5 turns out to be *if and only if* $\mathrm{supp}(\boldsymbol{x}) \subseteq \mathrm{supp}(\mathcal{X}^*)$, we have $\boldsymbol{x}^\top \boldsymbol{A}\boldsymbol{y}^* = \rho$ which strengthen our conclusion here.

## D.2 CONNECTION BETWEEN DUALITY GAP AND ITERATE DISTANCE

**Lemma D.6.** The constants $c_x, c_y$ defined below satisfy that $c_x, c_y > 0$.

$$
\begin{aligned}
c_x &= \inf_{\boldsymbol{x} \in \mathcal{F}_x \setminus \mathcal{X}^*} \max_{\boldsymbol{y} \in V^*(\prod_{\mathcal{X}^*}(\boldsymbol{x}))} \frac{(\boldsymbol{x} - \prod_{\mathcal{X}^*}(\boldsymbol{x}))^\top \boldsymbol{A}\boldsymbol{y}}{\|\boldsymbol{x} - \prod_{\mathcal{X}^*}(\boldsymbol{x})\|} \\
c_y &= \inf_{\boldsymbol{y} \in \mathcal{F}_y \setminus \mathcal{Y}^*} \max_{\boldsymbol{x} \in V^*(\prod_{\mathcal{Y}^*}(\boldsymbol{y}))} \frac{\boldsymbol{x}^\top \boldsymbol{A}(\prod_{\mathcal{Y}^*}(\boldsymbol{y}) - \boldsymbol{y})}{\|\boldsymbol{y} - \prod_{\mathcal{Y}^*}(\boldsymbol{y})\|}
\end{aligned}
\tag{D.17}
$$

where

$$
\begin{aligned}
\mathcal{F}_x &= \{\boldsymbol{x} | \boldsymbol{x} \in \mathcal{X}, \forall i \in \mathrm{supp}(\mathcal{X}^*)\ x_i \geq \epsilon_{\mathrm{dil}}\} \\
\mathcal{F}_y &= \{\boldsymbol{y} | \boldsymbol{y} \in \mathcal{Y}, \forall i \in \mathrm{supp}(\mathcal{Y}^*)\ y_i \geq \epsilon_{\mathrm{dil}}\},
\end{aligned}
\tag{D.18}
$$

and $\epsilon_{\mathrm{dil}}$ is some game dependent constant defined in Lemma D.7.

*Proof.* Define the set $\mathcal{X}' = \{\boldsymbol{x} | \boldsymbol{x} \in \mathcal{X}, \|\boldsymbol{x} - \prod_{\mathcal{X}^*}(\boldsymbol{x})\| \geq \epsilon_{\mathrm{dil}}\}$. In the following, we will show that we only need to consider $x \in \mathcal{X}'$ instead of $\mathcal{F}_x \setminus \mathcal{X}^*$. Formally we will prove that for any $\boldsymbol{x} \in \mathcal{F}_x \setminus \mathcal{X}^*$, we have $\boldsymbol{x}' \in \mathcal{X}'$ so that

$$
\forall \boldsymbol{y}, \frac{(\boldsymbol{x} - \prod_{\mathcal{X}^*}(\boldsymbol{x}))^\top \boldsymbol{A}\boldsymbol{y}}{\|\boldsymbol{x} - \prod_{\mathcal{X}^*}(\boldsymbol{x})\|} = \frac{(\boldsymbol{x}' - \prod_{\mathcal{X}^*}(\boldsymbol{x}'))^\top \boldsymbol{A}\boldsymbol{y}}{\|\boldsymbol{x}' - \prod_{\mathcal{X}^*}(\boldsymbol{x}')\|}.
\tag{D.19}
$$

The claim trivially holds if $x \in \mathcal{X}'$. Otherwise, let $\boldsymbol{x}' = \prod_{\mathcal{X}^*}(\boldsymbol{x}) + \frac{\epsilon_{\mathrm{dil}}}{\|\boldsymbol{x} - \prod_{\mathcal{X}^*}(\boldsymbol{x})\|}(\boldsymbol{x} - \prod_{\mathcal{X}^*}(\boldsymbol{x}))$. For any element that $x_i \geq \prod_{\mathcal{X}^*}(\boldsymbol{x})_i \geq 0$, we know that $x'_i \geq 0$.

For elements that $\prod_{\mathcal{X}^*}(\boldsymbol{x})_i > x_i \geq 0$, we can ensure that $i \in \mathrm{supp}(\mathcal{X}^*)$, which means that $\prod_{\mathcal{X}^*}(\boldsymbol{x})_i > x_i \geq \epsilon_{\mathrm{dil}}$ since $\boldsymbol{x} \in \mathcal{F}_x \setminus \mathcal{X}^*$. Therefore, we have $x'_i \geq \prod_{\mathcal{X}^*}(\boldsymbol{x})_i - |x_i - \prod_{\mathcal{X}^*}(\boldsymbol{x})_i| \cdot \frac{\epsilon_{\mathrm{dil}}}{\|\boldsymbol{x} - \prod_{\mathcal{X}^*}(\boldsymbol{x})\|} \geq \prod_{\mathcal{X}^*}(\boldsymbol{x})_i - \epsilon_{\mathrm{dil}} \geq 0$. Also, for any $h \in \mathcal{H}^\mathcal{X}$,

$$\sum_{i \in \Omega_h} x'_i = \frac{\epsilon_{\text{dil}}}{\|\boldsymbol{x} - \prod_{\mathcal{X}^*}(\boldsymbol{x})\|} \sum_{i \in \Omega_h} x_i + (1 - \frac{\epsilon_{\text{dil}}}{\|\boldsymbol{x} - \prod_{\mathcal{X}^*}(\boldsymbol{x})\|}) \sum_{i \in \Omega_h} \prod_{\mathcal{X}^*}(\boldsymbol{x})_i$$

$$= \frac{\epsilon_{\text{dil}}}{\|\boldsymbol{x} - \prod_{\mathcal{X}^*}(\boldsymbol{x})\|} x_{\sigma(h)} + (1 - \frac{\epsilon_{\text{dil}}}{\|\boldsymbol{x} - \prod_{\mathcal{X}^*}(\boldsymbol{x})\|}) \prod_{\mathcal{X}^*}(\boldsymbol{x})_{\sigma(h)} \tag{D.20}$$

$$= x'_{\sigma(h)}.$$

Therefore, $\boldsymbol{x}' \in \mathcal{X}$ and we can conclude that $\boldsymbol{x}' \in \mathcal{X}'$ since $\prod_{\mathcal{X}^*}(\boldsymbol{x}) = \prod_{\mathcal{X}^*}(\boldsymbol{x}')$.

Moreover, since $\boldsymbol{x}' - \prod_{\mathcal{X}^*}(\boldsymbol{x})$ and $\boldsymbol{x} - \prod_{\mathcal{X}^*}(\boldsymbol{x})$ are parallel and $\prod_{\mathcal{X}^*}(\boldsymbol{x}) = \prod_{\mathcal{X}^*}(\boldsymbol{x}')$, we can conclude that Eq (D.19) is satisfied. Because $\mathcal{X}'$ is closed, we can define

$$c'_x = \min_{\boldsymbol{x} \in \mathcal{X}'} \max_{\boldsymbol{y} \in \mathcal{V}^*(\prod_{\mathcal{X}^*}(\boldsymbol{x}))} \frac{(\boldsymbol{x} - \prod_{\mathcal{X}^*}(\boldsymbol{x}))^\top \boldsymbol{A} \boldsymbol{y}}{\|\boldsymbol{x} - \prod_{\mathcal{X}^*}(\boldsymbol{x})\|}$$

$$c'_y = \min_{\boldsymbol{y} \in \mathcal{Y}'} \max_{\boldsymbol{x} \in \mathcal{V}^*(\prod_{\mathcal{Y}^*}(\boldsymbol{y}))} \frac{\boldsymbol{x}^\top \boldsymbol{A}(\prod_{\mathcal{Y}^*}(\boldsymbol{y}) - \boldsymbol{y})}{\|\boldsymbol{y} - \prod_{\mathcal{Y}^*}(\boldsymbol{y})\|} \tag{D.21}$$

with the inequality that $c_x \geq c'_x$ and $c_y \geq c'_y$ by the discussion above. Then, we will prove that $c'_x, c'_y > 0$.

Firstly, we will prove that $c'_y \geq 0$. If $c'_y < 0$, then it says that there's some $\boldsymbol{y}$ so that

$$\min_{\boldsymbol{x} \in \mathcal{V}^*(\prod_{\mathcal{Y}^*}(\boldsymbol{y}))} \boldsymbol{x}^\top \boldsymbol{A} \boldsymbol{y} > \rho \tag{D.22}$$

which implies that for any $\boldsymbol{x}^* \in \mathcal{X}^*$, $\boldsymbol{x}^{*\top} \boldsymbol{A} \boldsymbol{y} > \rho$. And it contradicts with the definition of $\mathcal{X}^*$.

If $c'_y = 0$, then for some $\boldsymbol{y} \notin \mathcal{Y}^*$,

$$\max_{\boldsymbol{x} \in V^*(\prod_{\mathcal{Y}^*}(\boldsymbol{y}))} \boldsymbol{x}^\top \boldsymbol{A}(\prod_{\mathcal{Y}^*}(\boldsymbol{y}) - \boldsymbol{y}) = 0. \tag{D.23}$$

Let $PS^{\mathcal{X}}$ denote all pure strategies of $\boldsymbol{x}$. If $PS(\boldsymbol{y}^*) = PS^{\mathcal{X}}$, then $V^*(\prod_{\mathcal{Y}^*}(\boldsymbol{y})) = \mathcal{X}$. Eq (D.23) implies that $\min_{\boldsymbol{x} \in \mathcal{X}} \boldsymbol{x}^\top \boldsymbol{A} \boldsymbol{y} = \rho$ so that $\boldsymbol{y} \in \mathcal{Y}^*$. But this contradicts with the definition that $\boldsymbol{y} \notin \mathcal{Y}^*$.

If $PS(\boldsymbol{y}^*) \neq PS^{\mathcal{X}}$, we define

$$\xi(\boldsymbol{y}^*) = \min_{\boldsymbol{x} \in PS^{\mathcal{X}} \setminus PS(\boldsymbol{y}^*)} \{\boldsymbol{x}^\top \boldsymbol{A} \boldsymbol{y}^* - \rho\}. \tag{D.24}$$

And we can prove that $\xi(\boldsymbol{y}^*) \in (0, 2M]$. The lower bound is directly from Lemma D.3 and the upperbound is from the assumption on $\boldsymbol{A}$ that $\forall \boldsymbol{y} \in \mathcal{Y}, \|\boldsymbol{A} \boldsymbol{y}\|_\infty \leq 1$.

Let $\boldsymbol{y}' = \prod_{\mathcal{Y}^*}(\boldsymbol{y}) + \frac{\xi(\prod_{\mathcal{Y}^*}(\boldsymbol{y}))}{2N \cdot M}(\boldsymbol{y} - \prod_{\mathcal{Y}^*}(\boldsymbol{y})) \in \mathcal{Y}$. For any pure strategy $\boldsymbol{x} \in PS^{\mathcal{X}} \setminus PS(\boldsymbol{y}^*)$, we have

$$\boldsymbol{x}^\top \boldsymbol{A} \boldsymbol{y}' = \boldsymbol{x}^\top \boldsymbol{A} \prod_{\mathcal{Y}^*}(\boldsymbol{y}) - \boldsymbol{x}^\top \left(\boldsymbol{A}(\prod_{\mathcal{Y}^*}(\boldsymbol{y}) - \boldsymbol{y}')\right)$$

$$\geq \boldsymbol{x}^\top \boldsymbol{A} \prod_{\mathcal{Y}^*}(\boldsymbol{y}) - \|\boldsymbol{x}\|_\infty \cdot \|\prod_{\mathcal{Y}^*}(\boldsymbol{y}) - \boldsymbol{y}'\|_1$$

$$\geq \boldsymbol{x}^\top \boldsymbol{A} \prod_{\mathcal{Y}^*}(\boldsymbol{y}) - \frac{\xi(\prod_{\mathcal{Y}^*}(\boldsymbol{y}))}{M} \tag{D.25}$$

$$\geq \rho$$

where the last inequality comes from the definition of $\xi(\prod_{\mathcal{Y}^*}(\boldsymbol{y}))$ in Eq (D.24).

For any pure strategy $\boldsymbol{x} \in PS(\boldsymbol{y}^*)$, we have

$$\boldsymbol{x}^\top \boldsymbol{A} \boldsymbol{y}' = \boldsymbol{x}^\top \boldsymbol{A} \prod_{\mathcal{Y}^*}(\boldsymbol{y}) + \frac{\xi(\prod_{\mathcal{Y}^*}(\boldsymbol{y}))}{2N \cdot M} \boldsymbol{x}^\top \boldsymbol{A}(\boldsymbol{y} - \prod_{\mathcal{Y}^*}(\boldsymbol{y}))$$

$$\geq \boldsymbol{x}^\top \boldsymbol{A} \prod_{\mathcal{Y}^*}(\boldsymbol{y}) \tag{D.26}$$

$$= \rho.$$

Therefore, $\min_{\boldsymbol{x} \in \mathcal{X}} \boldsymbol{x}^\top \boldsymbol{A} \boldsymbol{y}' \geq \rho$ since any $\boldsymbol{x} \in \mathcal{X}$ is a linear combination of pure strategies. And it implies that $\boldsymbol{y}' \notin \mathcal{Y}^*$ is also a maximin point, contradicting with the definition of $\mathcal{Y}^*$.

So, $c_y' > 0$ and so does $c_x'$. And further we have that $c_x, c_y > 0$. $\qquad\square$

**Lemma D.7.** For any $t = 1, 2, ...,$ and $i \in \mathrm{supp}(\mathcal{Z}^*)$, and $\eta \leq \frac{1}{8P}$, `Reg-DOMWU` ensures that $\widehat{z}_{t,i} \geq \epsilon_{\mathrm{dil}}$ where $\epsilon_{\mathrm{dil}}$ is some game-dependent constant.

*Proof.* By Lemma C.2, `Reg-DOMD` satisfies

$$\eta\tau\psi^{\mathcal{Z}}(\boldsymbol{z}) - \eta\tau\psi^{\mathcal{Z}}(\boldsymbol{z}_t) + \eta F(\boldsymbol{z}_t)^\top(\boldsymbol{z}_t - \boldsymbol{z}) \leq (1 - \eta\tau)D_{\psi^z}(\boldsymbol{z}, \widehat{\boldsymbol{z}}_t) - D_{\psi^z}(\boldsymbol{z}, \widehat{\boldsymbol{z}}_{t+1})$$
$$- D_{\psi^z}(\widehat{\boldsymbol{z}}_{t+1}, \boldsymbol{z}_t) - \frac{7}{8}D_{\psi^z}(\boldsymbol{z}_t, \widehat{\boldsymbol{z}}_t) + \frac{1}{8}D_{\psi^z}(\widehat{\boldsymbol{z}}_t, \boldsymbol{z}_{t-1}). \tag{D.27}$$

Pick $\boldsymbol{z} = \boldsymbol{z}^*$ such that $\mathrm{supp}(\boldsymbol{z}^*) = \mathrm{supp}(\mathcal{Z}^*)$ (note that such a $\boldsymbol{z}^* \in \mathcal{Z}^*$ must exist since $\mathcal{Z}^*$ is convex). Then, we have

$$\eta\tau\psi^{\mathcal{Z}}(\boldsymbol{z}^*) - \eta\tau\psi^{\mathcal{Z}}(\boldsymbol{z}_t) \leq \eta\tau\psi^{\mathcal{Z}}(\boldsymbol{z}^*) - \eta\tau\psi^{\mathcal{Z}}(\boldsymbol{z}_t) + \eta F(\boldsymbol{z}_t)^\top(\boldsymbol{z}_t - \boldsymbol{z}^*)$$
$$\leq (1 - \eta\tau)D_{\psi^z}(\boldsymbol{z}^*, \widehat{\boldsymbol{z}}_t) - D_{\psi^z}(\boldsymbol{z}^*, \widehat{\boldsymbol{z}}_{t+1}) \tag{D.28}$$
$$- D_{\psi^z}(\widehat{\boldsymbol{z}}_{t+1}, \boldsymbol{z}_t) - \frac{7}{8}D_{\psi^z}(\boldsymbol{z}_t, \widehat{\boldsymbol{z}}_t) + \frac{1}{8}D_{\psi^z}(\widehat{\boldsymbol{z}}_t, \boldsymbol{z}_{t-1})$$

where the first inequality comes from $F(\boldsymbol{z}_t)^\top(\boldsymbol{z}_t - \boldsymbol{z}^*) = \boldsymbol{x}_t^\top \boldsymbol{A} \boldsymbol{y}^* - \boldsymbol{x}^{*\top} \boldsymbol{A} \boldsymbol{y}_t \geq 0$ by definition of NE. And it further implies that

$$D_{\psi^z}(\boldsymbol{z}^*, \widehat{\boldsymbol{z}}_{t+1}) + D_{\psi^z}(\widehat{\boldsymbol{z}}_{t+1}, \boldsymbol{z}_t) \leq (1 - \eta\tau)\Big(D_{\psi^z}(\boldsymbol{z}^*, \widehat{\boldsymbol{z}}_t) + D_{\psi^z}(\widehat{\boldsymbol{z}}_t, \boldsymbol{z}_{t-1})\Big)$$
$$- \frac{1}{2}\Big(D_{\psi^z}(\boldsymbol{z}_t, \widehat{\boldsymbol{z}}_t) + D_{\psi^z}(\widehat{\boldsymbol{z}}_t, \boldsymbol{z}_{t-1})\Big) - \eta\tau\psi^{\mathcal{Z}}(\boldsymbol{z}^*) + \eta\tau\psi^{\mathcal{Z}}(\boldsymbol{z}_t)$$
$$\leq (1 - \eta\tau)\Big(D_{\psi^z}(\boldsymbol{z}^*, \widehat{\boldsymbol{z}}_t) + D_{\psi^z}(\widehat{\boldsymbol{z}}_t, \boldsymbol{z}_{t-1})\Big)$$
$$- \frac{1}{2}\Big(D_{\psi^z}(\boldsymbol{z}_t, \widehat{\boldsymbol{z}}_t) + D_{\psi^z}(\widehat{\boldsymbol{z}}_t, \boldsymbol{z}_{t-1})\Big) - \eta\tau\psi^{\mathcal{Z}}(\boldsymbol{z}^*) \tag{D.29}$$

when $\eta\tau \leq \eta \leq \frac{3}{8}$.

When $\tau = 0$, we have

$$D_{\psi^z}(\boldsymbol{z}^*, \widehat{\boldsymbol{z}}_{t+1}) \leq D_{\psi^z}(\boldsymbol{z}^*, \widehat{\boldsymbol{z}}_1) + D_{\psi^z}(\widehat{\boldsymbol{z}}_1, \boldsymbol{z}_0) = D_{\psi^z}(\boldsymbol{z}^*, \widehat{\boldsymbol{z}}_1). \tag{D.30}$$

And when $\tau > 0$, we have

$$D_{\psi^z}(\boldsymbol{z}^*, \widehat{\boldsymbol{z}}_{t+1}) \leq (1 - \eta\tau)^t D_{\psi^z}(\boldsymbol{z}^*, \widehat{\boldsymbol{z}}_1) - \psi^{\mathcal{Z}}(\boldsymbol{z}^*) \leq D_{\psi^z}(\boldsymbol{z}^*, \widehat{\boldsymbol{z}}_1) - \psi^{\mathcal{Z}}(\boldsymbol{z}^*). \tag{D.31}$$

Therefore, for any $i \in \mathrm{supp}(\mathcal{Z}^*) = \mathrm{supp}(\boldsymbol{z}^*)$,

$$z_i^* \log \frac{1}{\widehat{q}_{t+1,i}} \leq \sum_j \alpha_{h(j)} z_j^* \log \frac{1}{\widehat{q}_{t+1,j}} = D_{\psi^z}(\boldsymbol{z}^*, \widehat{\boldsymbol{z}}_{t+1}) - \sum_j \alpha_{h(j)} z_j^* \log q_j^*$$
$$\leq D_{\psi^z}(\boldsymbol{z}^*, \widehat{\boldsymbol{z}}_1) - \psi^{\mathcal{Z}}(\boldsymbol{z}^*) - \sum_j \alpha_{h(j)} z_j^* \log q_j^*$$
$$= \sum_j \alpha_{h(j)} z_j^* \log \frac{1}{\widehat{q}_{1,j}} - \psi^{\mathcal{Z}}(\boldsymbol{z}^*) \tag{D.32}$$
$$\leq 2P\|\boldsymbol{\alpha}\|_\infty \log C_\Omega$$

where the last inequality comes from the fact that $\widehat{\boldsymbol{z}}_1$ is initialized as a uniform strategy. Therefore,

$$\widehat{q}_{t+1,i} \geq \exp\Big(-2P\|\boldsymbol{\alpha}\|_\infty \frac{\log C_\Omega}{\min_{i \in \mathrm{supp}(\mathcal{Z}^*)} z_i^*}\Big) \tag{D.33}$$

for any $i \in \text{supp}(\mathcal{Z}^*)$.

And we further have

$$
\begin{aligned}
\widehat{z}_{t+1,i} &= \widehat{z}_{t+1,\sigma(h(i))} \cdot \widehat{q}_{t+1,i} \\
&= \widehat{z}_{t+1,\sigma(h(\sigma(h(i))))} \cdot \widehat{q}_{t+1,\sigma(h(i))} \cdot \widehat{q}_{t+1,i} \\
&= ... \\
&\geq \exp\Big(-2P^2 \|\boldsymbol{\alpha}\|_\infty \frac{\log C_\Omega}{\min_{i \in \text{supp}(\mathcal{Z}^*)} \boldsymbol{z}_i^*}\Big) \\
&=: \epsilon_{\text{dil}} > 0,
\end{aligned}
\tag{D.34}
$$

completing the proof. □

## E    PROOF OF LEMMA 4.1

Our regret decomposition framework follows the laminar regret decomposition (Farina et al., 2019b), which is a more general case of the original counterfactual regret minimization (Zinkevich et al., 2007). The second part of Lemma 4.1, the boundedness of regret, also appears in (Farina et al., 2019b, Theorem 2). But here we use Lemma E.1 to prove it which is more concise.

**Lemma E.1** (First part of Lemma 4.1). The difference satisfies that $G_T^{\mathcal{Z}}(\boldsymbol{z}) = \sum_{h \in \mathcal{H}^{\mathcal{Z}}} z_{\sigma(h)} G_T^h(\boldsymbol{z})$ for any $\boldsymbol{z} \in \mathcal{Z}^\gamma$ and $\gamma \geq 0$.

*Proof.* We define the scalar *subtree value* $S_t^h(\boldsymbol{z})$ recursively,

$$
S_t^h(\boldsymbol{z}) := \sum_{i \in \Omega_h} q_i\big((\boldsymbol{A}\boldsymbol{y}_t)_i + \sum_{h' \in \mathcal{H}_i} S_t^{h'}(\boldsymbol{z})\big) + \tau\alpha_h \psi^\Delta(\boldsymbol{q}_h).
\tag{E.1}
$$

For terminal nodes, $\mathcal{H}_i$ will be empty set and thus $S_t^h(\boldsymbol{z}) = \sum_{i \in \Omega_h} q_i(\boldsymbol{A}\boldsymbol{y}_t)_i + \tau\alpha_h \psi^\Delta(\boldsymbol{q}_h)$.

By definition, for any $\boldsymbol{z} \in \mathcal{Z}^\gamma$, we have

$$
\begin{aligned}
G_T^{\mathcal{Z}}(\boldsymbol{z}) &= \sum_{t=1}^T (\langle F(\boldsymbol{z}_t), \boldsymbol{z}_t \rangle + \tau\psi^{\mathcal{Z}}(\boldsymbol{z}_t)) - \sum_{t=1}^T (\langle F(\boldsymbol{z}_t), \boldsymbol{z} \rangle + \tau\psi^{\mathcal{Z}}(\boldsymbol{z})) \\
&= \sum_{t=1}^T \sum_{h \in \mathcal{H}_0} S_t^h(\boldsymbol{z}_t) - \sum_{t=1}^T \sum_{h \in \mathcal{H}_0} S_t^h(\boldsymbol{z}) \\
&= \sum_{h \in \mathcal{H}_0} \Big(\sum_{t=1}^T S_t^h(\boldsymbol{z}_t) - \sum_{t=1}^T S_t^h(\boldsymbol{z})\Big)
\end{aligned}
\tag{E.2}
$$

where $\mathcal{H}_0 = \{h : h \in \mathcal{H}^{\mathcal{Z}}, \sigma(h) = 0\}$ is the set of information set at the root of treeplex. Note that $\mathcal{Z}^\gamma = \mathcal{Z}_{h_1}^\gamma \times \mathcal{Z}_{h_2}^\gamma \times ... \times \mathcal{Z}_{h_m}$ where $\mathcal{H}_0 = \{h_1, h_2, ..., h_m\}$. Then, the inequality in the second line is simply by expanding the definition of $S_t^h(\boldsymbol{z})$ from the recursive manner.

We further define $G^h_{T,\text{sub}} := \sum_{t=1}^T S^h_t(\boldsymbol{z}_t) - \sum_{t=1}^T S^h_t(\boldsymbol{z})$. Then,

$$
\begin{aligned}
&G^h_{T,\text{sub}}(\boldsymbol{z}) \\
=&\sum_{t=1}^T S^h_t(\boldsymbol{z}_t) - \sum_{t=1}^T S^h_t(\boldsymbol{z}) \\
=&\sum_{t=1}^T S^h_t(\boldsymbol{z}_t) - \Big(\sum_{t=1}^T \big(\sum_{i\in\Omega_h} q_i(\boldsymbol{A}\boldsymbol{y}_t)_i + \tau\alpha_h\psi^\Delta(\boldsymbol{q}_h)\big) + \sum_{i\in\Omega_h} q_i \sum_{h'\in\mathcal{H}_i} \sum_{t=1}^T S^{h'}_t(\boldsymbol{z})\Big) \\
\overset{(i)}{=}&\sum_{t=1}^T S^h_t(\boldsymbol{z}_t) - \Big(\sum_{t=1}^T \big(\sum_{i\in\Omega_h} q_i(\boldsymbol{A}\boldsymbol{y}_t)_i + \tau\alpha_h\psi^\Delta(\boldsymbol{q}_h)\big) + \sum_{i\in\Omega_h} q_i \sum_{h'\in\mathcal{H}_i} \big(\sum_{t=1}^T S^{h'}_t(\boldsymbol{z}_t) - G^{h'}_{\text{sub}}(\boldsymbol{z})\big)\Big) \\
=&\sum_{t=1}^T S^h_t(\boldsymbol{z}_t) - \Big(\sum_{t=1}^T \big(\sum_{i\in\Omega_h} q_i\big((\boldsymbol{A}\boldsymbol{y}_t)_i + \sum_{h'\in\mathcal{H}_i} S^{h'}_t(\boldsymbol{z}_t)\big) + \tau\alpha_h\psi^\Delta(\boldsymbol{q}_h)\big)\Big) - \Big(\sum_{i\in\Omega_h} q_i \sum_{h'\in\mathcal{H}_i} -G^{h'}_{\text{sub}}(\boldsymbol{z})\Big) \\
=&G^h_T(\boldsymbol{q}_h) + \sum_{i\in\Omega_h} q_i \sum_{h'\in\mathcal{H}_i} G^{h'}_{\text{sub}}(\boldsymbol{z})
\end{aligned}
$$
(E.3)

where $(i)$ comes from $\sum_{t=1}^T S^{h'}_t(\boldsymbol{z}) = \sum_{t=1}^T S^{h'}_t(\boldsymbol{z}_t) - G^{h'}_{\text{sub}}(\boldsymbol{z})$.

By applying it recursively, we will get for any $\boldsymbol{z} \in \mathcal{Z}^\gamma$,

$$
G^{\mathcal{Z}}_T(\boldsymbol{z}) = \sum_{h\in\mathcal{H}^{\mathcal{Z}}} z_{\sigma(h)} G^h_T(\boldsymbol{q}_h),
$$
(E.4)

which completes the proof. $\qquad\square$

**Lemma E.2** (Second part of Lemma 4.1)**.** The regret satisfies that $R^{\mathcal{Z}}_T \leq \max_{\widehat{\boldsymbol{z}}\in\mathcal{Z}^\gamma} \sum_{h\in\mathcal{H}^{\mathcal{Z}}} \widehat{z}_{\sigma(h)} R^h_T$ for any $\gamma \geq 0$.

*Proof.* By Lemma E.1, we have

$$
\begin{aligned}
R^{\mathcal{Z}}_T = \max_{\widehat{\boldsymbol{z}}\in\mathcal{Z}^\gamma} G^{\mathcal{Z}}_T(\widehat{\boldsymbol{z}}) &= \max_{\widehat{\boldsymbol{z}}\in\mathcal{Z}^\gamma} \sum_{h\in\mathcal{H}^{\mathcal{Z}}} \widehat{z}_{\sigma(h)} G^h_T\big(\frac{\widehat{\boldsymbol{z}}_h}{\widehat{z}_{\sigma(h)}}\big) \\
&\leq \max_{\widehat{\boldsymbol{z}}\in\mathcal{Z}^\gamma} \sum_{h\in\mathcal{H}^{\mathcal{Z}}} \widehat{z}_{\sigma(h)} \max_{\boldsymbol{q}_h\in\Delta^\gamma_{|\Omega_h|}} G^h_T(\boldsymbol{q}_h) \\
&= \max_{\widehat{\boldsymbol{z}}\in\mathcal{Z}^\gamma} \sum_{h\in\mathcal{H}^{\mathcal{Z}}} \widehat{z}_{\sigma(h)} R^h_T
\end{aligned}
$$

which completes the proof. $\qquad\square$

# F PROOF OF THEOREM 4.3 AND THEOREM 4.6

## F.1 PROOF OF LEMMA F.1

**Lemma F.1.** For any information set $h \in \mathcal{H}^{\mathcal{Z}}$, $\boldsymbol{q}_h \in \Delta^\gamma_{|\Omega_h|}$ and $\tau \leq \frac{1}{2\|\boldsymbol{\alpha}\|_\infty}$, Reg-CFR guarantees

$$
\begin{aligned}
G^h_T(\boldsymbol{q}_h) \leq& \lambda^h_{T+1} D_{\psi^\Delta}(\boldsymbol{q}_h, \boldsymbol{q}_{1,h}) + \|V^h(\boldsymbol{z}_{\frac{3}{2}}) - V^h(\boldsymbol{z}_{\frac{1}{2}})\|^2 - \alpha_h\tau\sum_{t=2}^T D_{\psi^\Delta}(\boldsymbol{q}_h, \boldsymbol{q}_{t,h}) \\
&+ \sum_{t=2}^T \Big(\frac{\|V^h(\boldsymbol{z}_{t+\frac{1}{2}}) - V^h(\boldsymbol{z}_{t-\frac{1}{2}})\|^2}{\lambda^h_t} - \frac{\lambda^h_{t-1}}{8}\|\boldsymbol{q}_{t+\frac{1}{2},h} - \boldsymbol{q}_{t-\frac{1}{2},h}\|^2\Big).
\end{aligned}
$$
(F.1)

*Proof.* By Lemma F.6,

$$
\begin{aligned}
G_T^h(\boldsymbol{q}_h) &= \sum_{t=1}^T \Big[ \Big\langle V^h(\boldsymbol{z}_{t+\frac{1}{2}}), \boldsymbol{q}_{t+\frac{1}{2},h} - \boldsymbol{q}_h \Big\rangle + \tau\alpha_h\psi^\Delta(\boldsymbol{q}_{t+\frac{1}{2},h}) - \tau\alpha_h\psi^\Delta(\boldsymbol{q}_h) \Big] \\
&\leq (\lambda_1^h - \tau\alpha_h)D_{\psi^\Delta}(\boldsymbol{q}_h, \boldsymbol{q}_{1,h}) - \lambda_{T+1}^h D_{\psi^\Delta}(\boldsymbol{q}_h, \boldsymbol{q}_{T+1,h}) + (\lambda_{T+1}^h - \lambda_1^h)D_{\psi^\Delta}(\boldsymbol{q}_h, \boldsymbol{q}_{1,h}) \\
&\quad - (\lambda_1^h - \tau\alpha_h)D_{\psi^\Delta}(\boldsymbol{q}_{\frac{3}{2},h}, \boldsymbol{q}_{1,h}) - \frac{\lambda_T^h}{2}D_{\psi^\Delta}(\boldsymbol{q}_{T+1,h}, \boldsymbol{q}_{T+\frac{1}{2},h}) \\
&\quad - \sum_{t=2}^T \Big( \frac{\lambda_{t-1}^h}{2}D_{\psi^\Delta}(\boldsymbol{q}_{t,h}, \boldsymbol{q}_{t-\frac{1}{2},h}) + (\lambda_t^h - \tau\alpha_h)D_{\psi^\Delta}(\boldsymbol{q}_{t+\frac{1}{2},h}, \boldsymbol{q}_{t,h}) \Big) \\
&\quad + \sum_{t=1}^T \Big( \Big\langle V^h(\boldsymbol{z}_{t+\frac{1}{2}}) - V^h(\boldsymbol{z}_{t-\frac{1}{2}}), \boldsymbol{q}_{t+\frac{1}{2},h} - \boldsymbol{q}_{t+1,h} \Big\rangle - \frac{\lambda_t^h}{2}D_{\psi^\Delta}(\boldsymbol{q}_{t+1,h}, \boldsymbol{q}_{t+\frac{1}{2},h}) \Big) \\
&\quad - \tau\alpha_h \sum_{t=2}^T D_{\psi^\Delta}(\boldsymbol{q}_h, \boldsymbol{q}_{t,h}).
\end{aligned}
\tag{F.2}
$$

By the strong convexity of $\psi^\Delta$,

$$
\begin{aligned}
\|\boldsymbol{q}_{t+\frac{1}{2},h} - \boldsymbol{q}_{t-\frac{1}{2},h}\|^2 &\leq 2\|\boldsymbol{q}_{t+\frac{1}{2},h} - \boldsymbol{q}_{t,h}\|^2 + 2\|\boldsymbol{q}_{t,h} - \boldsymbol{q}_{t-\frac{1}{2},h}\|^2 \\
&\leq 4D_{\psi^\Delta}(\boldsymbol{q}_{t+\frac{1}{2},h}, \boldsymbol{q}_{t,h}) + 4D_{\psi^\Delta}(\boldsymbol{q}_{t,h}, \boldsymbol{q}_{t-\frac{1}{2},h}).
\end{aligned}
\tag{F.3}
$$

Also,

$$
\begin{aligned}
&\Big\langle V^h(\boldsymbol{z}_{t+\frac{1}{2}}) - V^h(\boldsymbol{z}_{t-\frac{1}{2}}), \boldsymbol{q}_{t+\frac{1}{2},h} - \boldsymbol{q}_{t+1,h} \Big\rangle - \frac{\lambda_t^h}{2}D_{\psi^\Delta}(\boldsymbol{q}_{t+1,h}, \boldsymbol{q}_{t+\frac{1}{2},h}) \\
&\leq \frac{\|V^h(\boldsymbol{z}_{t+\frac{1}{2}}) - V^h(\boldsymbol{z}_{t-\frac{1}{2}})\|^2}{2\lambda_t^h} + \frac{\lambda_t^h}{2}\|\boldsymbol{q}_{t+\frac{1}{2},h} - \boldsymbol{q}_{t+1,h}\|^2 - \frac{\lambda_t^h}{2}D_{\psi^\Delta}(\boldsymbol{q}_{t+1,h}, \boldsymbol{q}_{t+\frac{1}{2},h}) \\
&\leq \frac{\|V^h(\boldsymbol{z}_{t+\frac{1}{2}}) - V^h(\boldsymbol{z}_{t-\frac{1}{2}})\|^2}{2\lambda_t^h} \\
&\leq \frac{\|V^h(\boldsymbol{z}_{t+\frac{1}{2}}) - V^h(\boldsymbol{z}_{t-\frac{1}{2}})\|^2}{\lambda_t^h}
\end{aligned}
\tag{F.4}
$$

where the second inequality is by Young's inequality.

Therefore, with $\tau\alpha_h \leq \frac{1}{2} \leq \frac{\lambda_{t-1}^h}{2}$,

$$
\begin{aligned}
&G_T^h(\boldsymbol{q}_h) \\
&= \sum_{t=1}^T \Big[ \Big\langle V^h(\boldsymbol{z}_{t+\frac{1}{2}}), \boldsymbol{q}_{t+\frac{1}{2},h} - \boldsymbol{q}_h \Big\rangle + \tau\alpha_h\psi^\Delta(\boldsymbol{q}_{t+\frac{1}{2},h}) - \tau\alpha_h\psi^\Delta(\boldsymbol{q}_h) \Big] \\
&\leq (\lambda_{T+1}^h - \tau\alpha_h)D_{\psi^\Delta}(\boldsymbol{q}_h, \boldsymbol{q}_{1,h}) \\
&\quad + \sum_{t=1}^T \frac{\|V^h(\boldsymbol{z}_{t+\frac{1}{2}}) - V^h(\boldsymbol{z}_{t-\frac{1}{2}})\|^2}{\lambda_t^h} - \frac{1}{8}\sum_{t=2}^T \lambda_{t-1}^h\|\boldsymbol{q}_{t+\frac{1}{2},h} - \boldsymbol{q}_{t-\frac{1}{2},h}\|^2 - \tau\alpha_h\sum_{t=2}^T D_{\psi^\Delta}(\boldsymbol{q}_h, \boldsymbol{q}_{t,h}) \\
&\leq \lambda_{T+1}^h D_{\psi^\Delta}(\boldsymbol{q}_h, \boldsymbol{q}_{1,h}) + \|V^h(\boldsymbol{z}_{\frac{3}{2}}) - V^h(\boldsymbol{z}_{\frac{1}{2}})\|^2 \\
&\quad + \sum_{t=2}^T \Big( \frac{\|V^h(\boldsymbol{z}_{t+\frac{1}{2}}) - V^h(\boldsymbol{z}_{t-\frac{1}{2}})\|^2}{\lambda_t^h} - \frac{\lambda_{t-1}^h}{8}\|\boldsymbol{q}_{t+\frac{1}{2},h} - \boldsymbol{q}_{t-\frac{1}{2},h}\|^2 \Big) - \tau\alpha_h\sum_{t=2}^T D_{\psi^\Delta}(\boldsymbol{q}_h, \boldsymbol{q}_{t,h}),
\end{aligned}
\tag{F.5}
$$

which completes the proof. $\qquad\square$

For simplicity, we use constant $M^h$ as the maximum value of $D_{\psi^\Delta}(\boldsymbol{q}_h, \boldsymbol{q}_{1,h})$ in information set $h$. $D_{\psi^\Delta}(\boldsymbol{q}_h, \boldsymbol{q}_{1,h})$ is upper-bounded since $\boldsymbol{q}_{1,h}$ is initialized as uniform distribution in $\Delta_{|\Omega_h|}$.

## F.2 Proof of Theorem 4.3

By Lemma E.1, we have

$$0 \le G_T^{\mathcal{Z}}(\boldsymbol{z}_\tau^{\gamma,*}) = \sum_{h \in \mathcal{H}} z_{\tau,\sigma(h)}^* G_T^h(\frac{\boldsymbol{z}_{\tau,h}^{\gamma,*}}{z_{\tau,\sigma(h)}^{\gamma,*}})$$

where the first inequality is by definition of $\boldsymbol{z}_\tau^{\gamma,*}$.

Now by Lemma F.1 taking $\boldsymbol{q}_h = \boldsymbol{q}_{\tau,h}^{\gamma,*} = \frac{\boldsymbol{z}_{\tau,h}^{\gamma,*}}{z_{\tau,\sigma(h)}^{\gamma,*}}$,

$$0 \le \sum_{h \in \mathcal{H}^{\mathcal{Z}}} z_{\tau,\sigma(h)}^{\gamma,*} \Big( \lambda_{T+1}^h M^h + \|V^h(\boldsymbol{z}_{\frac{3}{2}}) - V^h(\boldsymbol{z}_{\frac{1}{2}})\|^2 + \sum_{t=2}^T \Big( \frac{\|V^h(\boldsymbol{z}_{t+\frac{1}{2}}) - V^h(\boldsymbol{z}_{t-\frac{1}{2}})\|^2}{\lambda_t^h}$$
$$- \frac{\lambda_{t-1}^h}{8} \|\boldsymbol{q}_{t+\frac{1}{2},h} - \boldsymbol{q}_{t-\frac{1}{2},h}\|^2 \Big) - \tau\alpha_h \sum_{t=2}^T D_{\psi^\triangle}(\boldsymbol{q}_{\tau,h}^{\gamma,*}, \boldsymbol{q}_{t,h}) \Big)$$

where constant $M^h$ is the maximum value of $D_{\psi^\triangle}(\boldsymbol{q}_h, \boldsymbol{q}_{1,h})$ in information set $h$. $D_{\psi^\triangle}(\boldsymbol{q}_h, \boldsymbol{q}_{1,h})$ is upper-bounded since $\boldsymbol{q}_{1,h}$ is initialized as uniform distribution in $\Delta_{|\Omega_h|}$.

By rearranging the terms, we have

$$\tau \sum_{t=2}^T D_{\psi^{\mathcal{Z}}}(\boldsymbol{z}_\tau^{\gamma,*}, \boldsymbol{z}_t) \overset{(i)}{=} \tau \sum_{t=2}^T \sum_{h \in \mathcal{H}^{\mathcal{Z}}} \alpha_h z_{\tau,\sigma(h)}^{\gamma,*} D_{\psi^\triangle}(\boldsymbol{q}_{\tau,h}^{\gamma,*}, \boldsymbol{q}_{t,h}) \le C_\gamma \tag{F.6}$$

where $(i)$ is by the expanded form of the (dilated) Bregman divergence $D_{\psi^{\mathcal{Z}}}$ (see Lemma F.8 for a detailed proof) and the constant $C_\gamma$ is defined by

$$C_\gamma := \sum_{h \in \mathcal{H}^{\mathcal{Z}}} z_{\tau,\sigma(h)}^{\gamma,*} \Big( \lambda_{T+1}^h M^h + \|V^h(\boldsymbol{z}_{\frac{3}{2}}) - V^h(\boldsymbol{z}_{\frac{1}{2}})\|^2 + \sum_{t=2}^T \Big( \frac{\|V^h(\boldsymbol{z}_{t+\frac{1}{2}}) - V^h(\boldsymbol{z}_{t-\frac{1}{2}})\|^2}{\lambda_t^h}$$
$$- \frac{\lambda_{t-1}^h}{8} \|\boldsymbol{q}_{t+\frac{1}{2},h} - \boldsymbol{q}_{t-\frac{1}{2},h}\|^2 \Big).$$
$$\tag{F.7}$$

**Non-perturbed EFG best-iterate convergence.** To bound the quantity $\lambda_{T+1}^h M^h + \sum_{t=2}^T \frac{\|V^h(\boldsymbol{z}_{t+\frac{1}{2}}) - V^h(\boldsymbol{z}_{t-\frac{1}{2}})\|^2}{\lambda_t^h}$ in $C_\gamma$ (other parts of $C_\gamma$ have been already bounded by constant), we introduce the following Lemma, whose proof is postponed to F.5.

**Lemma F.2.** Consider update-rule in Eq (4.2). For any $h \in \mathcal{H}^{\mathcal{Z}}$, by taking $\kappa = T^{\frac{1}{2}}$, Reg-CFR satisfies that

$$\lambda_{T+1}^h M^h + \sum_{t=2}^T \frac{\|V^h(\boldsymbol{z}_{t+\frac{1}{2}}) - V^h(\boldsymbol{z}_{t-\frac{1}{2}})\|^2}{\lambda_t^h} \le O(T^{\frac{1}{4}}) \tag{F.8}$$

where constant $M^h$ is the maximum value of $D_{\psi^\triangle}(\boldsymbol{q}_h, \boldsymbol{q}_{1,h})$ in information set $h$.

By Lemma F.2, we know that $C_\gamma \le O(T^{\frac{1}{4}})$, which is

$$\tau \sum_{t=2}^T D_{\psi^{\mathcal{Z}}}(\boldsymbol{z}_\tau^*, \boldsymbol{z}_t) \le O(T^{\frac{1}{4}}). \tag{F.9}$$

Therefore, there exists $t' \in \{2, 3, ..., T\}$,

$$D_{\psi^{\mathcal{Z}}}(\boldsymbol{z}_\tau^*, \boldsymbol{z}_{t'}) \le \frac{1}{\tau} O(T^{-\frac{3}{4}}). \tag{F.10}$$

So, $\boldsymbol{z}_{t'}$ converges to $\boldsymbol{z}_\tau^*$ with convergence rate $O(T^{-\frac{3}{4}})$. □

**Perturbed EFG asymptotic last-iterate convergence.** From the form of constant $C_\gamma$ Eq (F.7) and $\lambda_{t-1}^h \geq \kappa \geq 1$, we have

$$C_\gamma \leq \sum_{h \in \mathcal{H}^{\mathcal{Z}}} z_{\tau,\sigma(h)}^{\gamma,*} \Big( \lambda_{T+1}^h M^h + \|V^h(\boldsymbol{z}_{\frac{3}{2}}) - V^h(\boldsymbol{z}_{\frac{1}{2}})\|^2 + \sum_{t=2}^T \Big( \frac{\|V^h(\boldsymbol{z}_{t+\frac{1}{2}}) - V^h(\boldsymbol{z}_{t-\frac{1}{2}})\|^2}{\lambda_t^h} - \frac{1}{8}\|\boldsymbol{q}_{t+\frac{1}{2},h} - \boldsymbol{q}_{t-\frac{1}{2},h}\|^2 \Big) \tag{F.11}$$

where constant $M^h$ is the maximum value of $D_{\psi^\triangle}(\boldsymbol{q}_h, \boldsymbol{q}_{1,h})$ in information set $h$.

We will prove that $C_\gamma \leq O(1)$ when $\gamma > 0$. By the Lipschitz property of $V^h(\boldsymbol{z})$ (see Lemma F.10 for a full proof), we have

$$\|V^h(\boldsymbol{z}_{t+\frac{1}{2}}) - V^h(\boldsymbol{z}_{t-\frac{1}{2}})\|^2 \leq \Big( L_2 \sum_{h \in \mathcal{H}^{\mathcal{Z}}} \|\boldsymbol{q}_{t+\frac{1}{2},h} - \boldsymbol{q}_{t-\frac{1}{2},h}\| \Big)^2$$

$$\leq P L_2^2 \sum_{h \in \mathcal{H}^{\mathcal{Z}}} \|\boldsymbol{q}_{t+\frac{1}{2},h} - \boldsymbol{q}_{t-\frac{1}{2},h}\|^2 \tag{F.12}$$

$$\leq P \frac{L_2^2}{\gamma^P} \sum_{h \in \mathcal{H}^{\mathcal{Z}}} z_{\tau,\sigma(h)}^{\gamma,*} \|\boldsymbol{q}_{t+\frac{1}{2},h} - \boldsymbol{q}_{t-\frac{1}{2},h}\|^2$$

where the last inequality is because $\frac{z_{\tau,i}^{\gamma,*}}{z_{\tau,\sigma(h(i))}^{\gamma,*}} \geq \gamma$ for any $i$ by definition of $\gamma$-perturbed EFG so that $z_{\tau,i}^{\gamma,*} \geq \gamma^P$. Since $z_{\tau,\sigma(h)}^{\gamma,*} \leq 1$,

$$\sum_{h \in \mathcal{H}^{\mathcal{Z}}} z_{\tau,\sigma(h)}^{\gamma,*} \|\boldsymbol{q}_{t+\frac{1}{2},h} - \boldsymbol{q}_{t-\frac{1}{2},h}\|^2 \geq \frac{\gamma^P}{P L_2^2} z_{\tau,\sigma(h)}^{\gamma,*} \|V^h(\boldsymbol{z}_{t+\frac{1}{2}}) - V^h(\boldsymbol{z}_{t-\frac{1}{2}})\|^2. \tag{F.13}$$

for any $h \in \mathcal{H}^{\mathcal{Z}}$.

Plugging inequality (F.13) to equation (F.11), we have

$$C_\gamma \leq \sum_{h \in \mathcal{H}^{\mathcal{Z}}} z_{\tau,\sigma(h)}^{\gamma,*} \|V^h(\boldsymbol{z}_{\frac{3}{2}}) - V^h(\boldsymbol{z}_{\frac{1}{2}})\|^2$$

$$+ \sum_{h \in \mathcal{H}^{\mathcal{Z}}} z_{\tau,\sigma(h)}^{\gamma,*} \Big( \lambda_{T+1}^h M^h - \frac{\gamma^P}{16 P^2 L_2^2} \|V^h(\boldsymbol{z}_{t+\frac{1}{2}}) - V^h(\boldsymbol{z}_{t-\frac{1}{2}})\|^2 \Big)$$

$$+ \sum_{h \in \mathcal{H}^{\mathcal{Z}}} z_{\tau,\sigma(h)}^{\gamma,*} \sum_{t=2}^T \Big( \frac{\|V^h(\boldsymbol{z}_{t+\frac{1}{2}}) - V^h(\boldsymbol{z}_{t-\frac{1}{2}})\|^2}{\lambda_t^h} - \frac{\gamma^P}{16 P^2 L_2^2} \|V^h(\boldsymbol{z}_{t+\frac{1}{2}}) - V^h(\boldsymbol{z}_{t-\frac{1}{2}})\|^2 \Big). \tag{F.14}$$

As a result, it remains to bound the following two quantities in Eq (F.15) and Eq (F.16) separately by some constant:

$$\lambda_{T+1}^h M^h - \iota \sum_{t=2}^T \|V^h(\boldsymbol{z}_{t+\frac{1}{2}}) - V^h(\boldsymbol{z}_{t-\frac{1}{2}})\|^2. \tag{F.15}$$

$$\sum_{t=2}^T \Big( \frac{\|V^h(\boldsymbol{z}_{t+\frac{1}{2}}) - V^h(\boldsymbol{z}_{t-\frac{1}{2}})\|^2}{\lambda_t^h} - \iota \|V^h(\boldsymbol{z}_{t+\frac{1}{2}}) - V^h(\boldsymbol{z}_{t-\frac{1}{2}})\|^2 \Big), \tag{F.16}$$

where we use $\iota := \frac{\gamma^P}{16 P^2 L_2^2}$ for convenience.

For Eq (F.15), since $\lambda_{T+1}^h = \sqrt{\kappa + \sum_{t=1}^T \delta_t^h}$ where $\delta_t^h = \|V^h(\boldsymbol{z}_{t+\frac{1}{2}}) - V^h(\boldsymbol{z}_{t-\frac{1}{2}})\|^2$, we can get

$$M^h \sqrt{\kappa + \sum_{t=1}^T \delta_t^h} - \iota \sum_{t=2}^T \delta_t^h \leq M^h \sqrt{\kappa + \delta_1^h} + M^h \sqrt{\sum_{t=2}^T \delta_t^h} - \iota \sum_{t=2}^T \delta_t^h = f^h\Big(\sqrt{\sum_{t=2}^T \delta_t^h}\Big) \tag{F.17}$$

where the second inequality comes from $\sqrt{a+b} \leq \sqrt{a} + \sqrt{b}$ and $f$ is a quadratic function with negative coefficient on the quadratic term. Therefore, it is upper-bounded by a constant.

As for Eq (F.16), we discuss the two possible cases separately.

When $\lim_{t\to\infty} \lambda_t^h < +\infty$, then $\sum_{t=1}^{\infty} \|V^h(z_{t+\frac{1}{2}}) - V^h(z_{t-\frac{1}{2}})\|^2 < +\infty$ from definition of $\lambda_t^h$ so that Eq (F.16) is bounded by a constant.

When $\lim_{t\to\infty} \lambda_t^h = +\infty$, then we must have $t' = \min_t \{t : 1/\lambda_t^h \leq \iota\}$. Therefore, Eq (F.16) is bounded by $\sum_{t=1}^{t'} \|V^h(z_{t+\frac{1}{2}}) - V^h(z_{t-\frac{1}{2}})\|^2 < +\infty$.

Therefore,

$$\sum_{t=2}^{T} D_{\psi^{\mathcal{Z}}}(z_\tau^{\gamma,*}, z_t) \leq \frac{O(1)}{\tau} \tag{F.18}$$

so that $z_t$ converges asymptotically to $z_\tau^{\gamma,*}$. □

*Proof of Corollary 4.4.* By Lemma D.1, we know that when $\tau = \frac{\epsilon}{4C_B}$, we will get

$$\max_{\widehat{z} \in \mathcal{Z}} F(z_t)^\top (z_t - \widehat{z}) \leq O(\epsilon) + 2P\sqrt{D_{\psi^{\mathcal{Z}}}(z_\tau^*, z_t)}. \tag{F.19}$$

Using Theorem 4.3, the proof is done. □

### F.3  PROOF OF THEOREM 4.6

We first state a stronger version of the folklore theorem here (Theorem 3 ; Farina et al., 2019b), to provide gurantees for average iterate below.

**Lemma F.3.** For a EFG where $l_t^{\mathcal{X}}(x_t) = x^\top A y_t + \tau \psi^{\mathcal{Z}}(x), l_t^{\mathcal{Y}}(y) = -x_t^\top A y + \tau \psi^{\mathcal{Z}}(y)$, the saddle point residual $\max_{\widehat{z} \in \mathcal{Z}} F(z)^\top (z - \widehat{z}) + \tau \psi(z) - \tau \psi(\widehat{z})$ of the average strategy $(\frac{1}{T} \sum_{t=1}^{T} x_t, \frac{1}{T} \sum_{t=1}^{T} y_t)$ is bounded by $\frac{R^{\mathcal{X}} + R^{\mathcal{Y}}}{T}$.

**Non-perturbed EFG average-iterate convergence.** From Lemma E.2 and Lemma F.1, by taking $\widehat{z} = \operatorname{argmax}_{z \in \mathcal{Z}} \sum_{h \in \mathcal{H}^{\mathcal{Z}}} z_{\sigma(h)} R_T^h$, we have

$$R_T^{\mathcal{Z}} \leq \sum_{h \in \mathcal{H}^{\mathcal{Z}}} \widehat{z}_{\sigma(h)} R_T^h$$

$$\leq \sum_{h \in \mathcal{H}} \widehat{z}_{\sigma(h)} \Big( \lambda_{T+1}^h M^h + \|V^h(z_{\frac{3}{2}}) - V^h(z_{\frac{1}{2}})\|^2 + \sum_{t=2}^{T} \Big( \frac{\|V^h(z_{t+\frac{1}{2}}) - V^h(z_{t-\frac{1}{2}})\|^2}{\lambda_t^h} \tag{F.20}$$

$$- \frac{\lambda_{t-1}^h}{8} \|q_{t+\frac{1}{2}}^h - q_{t-\frac{1}{2},h}^h\|^2 \Big) \Big) \leq O(T^{1/4})$$

where the last inequality is by Lemma F.2 and constant $M^h$ is the maximum value of $D_{\psi^\Delta}(q_h, q_{1,h})$ in information set $h$. Therefore, by Lemma F.3, the average iterate enjoys $O(T^{-\frac{3}{4}})$ convergence rate in terms of duality gap.

**Perturbed EFG average-iterate convergence.** By taking $q_h = \frac{\widehat{z}_h}{\widehat{z}_{\sigma(h)}}$ where $\widehat{z} = \operatorname{argmax}_{z \in \mathcal{Z}^\gamma} \sum_{h \in \mathcal{H}^{\mathcal{Z}}} z_{\sigma(h)} R_T^h$, from Lemma F.1, we have

$$R_T^{\mathcal{Z}} \leq \sum_{h \in \mathcal{H}^{\mathcal{Z}}} \widehat{z}_{\sigma(h)} R_T^h$$

$$\leq \sum_{h \in \mathcal{H}^{\mathcal{Z}}} \widehat{z}_{\sigma(h)} \Big( \lambda_{T+1}^h M^h + \|V^h(z_{\frac{3}{2}}) - V^h(z_{\frac{1}{2}})\|^2 \Big) \tag{F.21}$$

$$+ \sum_{h \in \mathcal{H}^{\mathcal{Z}}} \widehat{z}_{\sigma(h)} \sum_{t=2}^{T} \Big( \frac{\|V^h(z_{t+\frac{1}{2}}) - V^h(z_{t-\frac{1}{2}})\|^2}{\lambda_t^h} - \frac{1}{8} \|q_{t+\frac{1}{2},h} - q_{t-\frac{1}{2},h}\|^2 \Big)$$

where constant $M^h$ is the maximum value of $D_{\psi^{\Delta}}(\boldsymbol{q}_h, \boldsymbol{q}_{1,h})$ in information set $h$.

Follow the same analysis in F.2, we will get $R_T^{\mathcal{Z}} \leq O(1)$ which means the duality gap converges with convergence rate $O(\frac{1}{T})$ by Lemma F.3. $\qquad\square$

### F.4 APPROXIMATE EXTENSIVE-FORM PERFECT EQUILIBRIA

To illustrate why the NE of $\mathcal{Z}^{\gamma}$ for some fixed $\gamma > 0$ is a good approximation to the extensive-form perfect equilibria, we propose the following lemma.

**Lemma F.4.** The approximate NE $\boldsymbol{z}^{\gamma}$ of a $\gamma$-perturbed EFG is an approximation of the EFPE in terms of duality gap. That is,

$$\max_{\widehat{\boldsymbol{z}} \in \mathcal{Z}^0} F(\boldsymbol{z}^{\gamma})^{\top}(\boldsymbol{z}^{\gamma} - \widehat{\boldsymbol{z}}) \leq \max_{\widehat{\boldsymbol{z}} \in \mathcal{Z}^{\gamma}} F(\boldsymbol{z}^{\gamma})^{\top}(\boldsymbol{z}^{\gamma} - \widehat{\boldsymbol{z}}) + \gamma P^2 \qquad (F.22)$$

where $\mathcal{Z}^0$ is an infinitely small perturbed treeplex whose NE is exactly EFPE.

*Proof.* For any $\boldsymbol{z} \in \mathcal{Z}^0$, we can define $\boldsymbol{z}' \in \mathcal{Z}^{\gamma}$ as

$$\frac{z_i'}{z_{\sigma(h(i))}'} = (1 - \gamma|\Omega_{h(i)}|)\frac{z_i}{z_{\sigma(h(i))}} + \gamma. \qquad (F.23)$$

Then, we will use induction to prove that $\|\boldsymbol{z} - \boldsymbol{z}'\|_{\infty} \leq \gamma P$. Note that we will use $anc(i) := \{i, \sigma(h(i)), \sigma(h(\sigma(h(i)))), ..., i'\}$ where $\sigma(h(i')) = 0$ to denote the set of ancestors of index $i$ in the treeplex. Firstly, for index $i$ which satisfies that $\sigma(h(i)) = 0$, we have

$$\Big| \prod_{j \in anc(i)} \Big( (1 - \gamma|\Omega_{h(j)}|)q_j + \gamma \Big) - \prod_{j \in anc(i)} q_j \Big| = |-\gamma|\Omega_{h(i)}|q_i + \gamma| \leq \gamma|\Omega_{h(i)}|. \qquad (F.24)$$

Then, assume that we already prove that $\Big| \prod_{j \in anc(\sigma(h(i)))} \Big( (1 - \gamma|\Omega_{h(j)}|)q_j + \gamma \Big) - \prod_{j \in anc(\sigma(h(i)))} q_j \Big| \leq \gamma C_{\sigma(h(i))}$ for an index $i$ where $C_{\sigma(h(i))} = \sum_{j \in anc(\sigma(h(i)))} |\Omega_{h(j)}|$, then

$$\prod_{j \in anc(i)} \Big( (1 - \gamma|\Omega_{h(j)}|)q_j + \gamma \Big) - \prod_{j \in anc(i)} q_j$$

$$\geq \Big( (1 - \gamma|\Omega_{h(i)}|)q_i + \gamma \Big) \Big( \prod_{j \in anc(\sigma(h(i)))} q_j - \gamma C_{\sigma(h(i))} \Big) - \prod_{j \in anc(i)} q_j$$

$$= -\gamma|\Omega_{h(i)}| \prod_{j \in anc(i)} q_j + \gamma \prod_{j \in anc(i)} q_j - \gamma C_{\sigma(h(i))} \Big( (1 - \gamma|\Omega_{h(i)}|)q_i + \gamma \Big)$$

$$\geq -\gamma(|\Omega_{h(i)}| + C_{\sigma(h(i))})$$

and similarly, we have the upperbound $\gamma(1 + C_{\sigma(h(i))})$. Therefore, we have $\|\boldsymbol{z} - \boldsymbol{z}'\|_{\infty} \leq \gamma P$.

Therefore, for $\boldsymbol{y} = \operatorname{argmax}_{\widehat{\boldsymbol{y}} \in \mathcal{Y}^0} \boldsymbol{x}^{\gamma\top} \boldsymbol{A}\widehat{\boldsymbol{y}}$ where $\boldsymbol{z}^{\gamma} = (\boldsymbol{x}^{\gamma}, \boldsymbol{y}^{\gamma})$ is an approximate NE in a $\gamma$-perturbed EFG, we have

$$\boldsymbol{x}^{\gamma\top}\boldsymbol{A}\boldsymbol{y} = \boldsymbol{x}^{\gamma\top}\boldsymbol{A}\Big( \boldsymbol{y}' + (\boldsymbol{y} - \boldsymbol{y}') \Big)$$

$$= \boldsymbol{x}^{\gamma\top}\boldsymbol{A}\boldsymbol{y}' + \boldsymbol{x}^{\gamma\top}\boldsymbol{A}(\boldsymbol{y} - \boldsymbol{y}') \qquad (F.25)$$

$$\leq \max_{\widehat{\boldsymbol{y}} \in \mathcal{Y}^{\gamma}} \boldsymbol{x}^{\gamma\top}\boldsymbol{A}\widehat{\boldsymbol{y}} + \|\boldsymbol{A}^{\top}\boldsymbol{x}^{\gamma}\|_1 \cdot \|\boldsymbol{y} - \boldsymbol{y}'\|_{\infty}$$

which implies that

$$\max_{\widehat{\boldsymbol{z}} \in \mathcal{Z}^0} F(\boldsymbol{z}^{\gamma})^{\top}(\boldsymbol{z}^{\gamma} - \widehat{\boldsymbol{z}})$$

$$\leq \max_{\widehat{\boldsymbol{z}} \in \mathcal{Z}^{\gamma}} F(\boldsymbol{z}^{\gamma})^{\top}(\boldsymbol{z}^{\gamma} - \widehat{\boldsymbol{z}}) + \|\boldsymbol{A}^{\top}\boldsymbol{x}^{\gamma}\|_1 \cdot \|\boldsymbol{y} - \boldsymbol{y}'\|_{\infty} + \|\boldsymbol{A}\boldsymbol{y}^{\gamma}\|_1 \cdot \|\boldsymbol{x} - \boldsymbol{x}'\|_{\infty}$$

$$\leq \max_{\widehat{\boldsymbol{z}} \in \mathcal{Z}^{\gamma}} F(\boldsymbol{z}^{\gamma})^{\top}(\boldsymbol{z}^{\gamma} - \widehat{\boldsymbol{z}}) + \gamma P^2$$

where the last inequality comes from $\|F(\boldsymbol{z})\|_{\infty} \leq 1$ for any $\boldsymbol{z} \in \mathcal{Z}$. $\qquad\square$

### F.5 PROPERTIES OF `Reg-DS-OptMD` (4.2)

We first prove some standard results in DS-OptMD (Hsieh et al., 2021) when adding regularization.

**Lemma F.5.** For any convex set $\mathcal{C}$ and $\boldsymbol{u}_0, \boldsymbol{u} \in \mathcal{C}$, consider the update rule

$$\boldsymbol{u}_1 = \operatorname*{argmin}_{\widehat{\boldsymbol{u}}_1 \in \mathcal{C}} \{\langle \widehat{\boldsymbol{u}}_1, \boldsymbol{g} + \tau \nabla \psi^{\mathcal{C}}(\boldsymbol{u}) \rangle + \lambda_1 D_{\psi^{\mathcal{C}}}(\widehat{\boldsymbol{u}}_1, \boldsymbol{u}) + (\lambda_2 - \lambda_1) D_{\psi^{\mathcal{C}}}(\widehat{\boldsymbol{u}}_1, \boldsymbol{u}_0)\}$$

where $\psi^{\mathcal{C}}$ is a strongly convex function in $\mathcal{C}$. Then for any $\boldsymbol{u}_2 \in \mathcal{C}$,

$$
\begin{aligned}
&\tau \psi^{\mathcal{C}}(\boldsymbol{u}_1) - \tau \psi^{\mathcal{C}}(\boldsymbol{u}_2) + \langle \boldsymbol{g}, \boldsymbol{u}_1 - \boldsymbol{u}_2 \rangle \\
\leq &\lambda_1((1 - \frac{\tau}{\lambda_1}) D_{\psi^{\mathcal{C}}}(\boldsymbol{u}_2, \boldsymbol{u}) - D_{\psi^{\mathcal{C}}}(\boldsymbol{u}_2, \boldsymbol{u}_1) - (1 - \frac{\tau}{\lambda_1}) D_{\psi^{\mathcal{C}}}(\boldsymbol{u}_1, \boldsymbol{u})) \\
&+ (\lambda_2 - \lambda_1)(D_{\psi^{\mathcal{C}}}(\boldsymbol{u}_2, \boldsymbol{u}_0) - D_{\psi^{\mathcal{C}}}(\boldsymbol{u}_2, \boldsymbol{u}_1) - D_{\psi^{\mathcal{C}}}(\boldsymbol{u}_1, \boldsymbol{u}_0)).
\end{aligned}
\tag{F.26}
$$

*Proof.* Since

$$\boldsymbol{u}_1 = \operatorname*{argmin}_{\widehat{\boldsymbol{u}}_1 \in \mathcal{C}} \left\{ \left\langle \boldsymbol{g} - \lambda_1(1 - \frac{\tau}{\lambda_1}) \nabla \psi^{\mathcal{C}}(\boldsymbol{u}) - (\lambda_2 - \lambda_1) \nabla \psi^{\mathcal{C}}(\boldsymbol{u}_0), \widehat{\boldsymbol{u}}_1 \right\rangle + \lambda_2 \psi^{\mathcal{C}}(\widehat{\boldsymbol{u}}_1) \right\}, \tag{F.27}$$

by first-order optimality condition,

$$\left( \boldsymbol{g} + \lambda_2 \nabla \psi^{\mathcal{C}}(\boldsymbol{u}_1) - \lambda_1(1 - \frac{\tau}{\lambda_1}) \nabla \psi^{\mathcal{C}}(\boldsymbol{u}) - (\lambda_2 - \lambda_1) \nabla \psi^{\mathcal{C}}(\boldsymbol{u}_0) \right)^{\top} (\boldsymbol{u}_2 - \boldsymbol{u}_1) \geq 0. \tag{F.28}$$

Notice that

$$
\begin{aligned}
&\lambda_1((1 - \frac{\tau}{\lambda_1}) D_{\psi^{\mathcal{C}}}(\boldsymbol{u}_2, \boldsymbol{u}) - D_{\psi^{\mathcal{C}}}(\boldsymbol{u}_2, \boldsymbol{u}_1) - (1 - \frac{\tau}{\lambda_1}) D_{\psi^{\mathcal{C}}}(\boldsymbol{u}_1, \boldsymbol{u})) \\
= &\lambda_1 \left\langle \nabla \psi^{\mathcal{C}}(\boldsymbol{u}_1) - (1 - \frac{\tau}{\lambda_t}) \nabla \psi^{\mathcal{C}}(\boldsymbol{u}), \boldsymbol{u}_2 - \boldsymbol{u}_1 \right\rangle - \tau \psi^{\mathcal{C}}(\boldsymbol{u}_2) + \tau \psi^{\mathcal{C}}(\boldsymbol{u}_1),
\end{aligned}
\tag{F.29}
$$

and

$$
\begin{aligned}
&(\lambda_2 - \lambda_1)(D_{\psi^{\mathcal{C}}}(\boldsymbol{u}_2, \boldsymbol{u}_0) - D_{\psi^{\mathcal{C}}}(\boldsymbol{u}_2, \boldsymbol{u}_1) - D_{\psi^{\mathcal{C}}}(\boldsymbol{u}_1, \boldsymbol{u}_0)) \\
= &(\lambda_2 - \lambda_1) \left\langle \nabla \psi^{\mathcal{C}}(\boldsymbol{u}_1) - \nabla \psi^{\mathcal{C}}(\boldsymbol{u}_0), \boldsymbol{u}_2 - \boldsymbol{u}_1 \right\rangle.
\end{aligned}
\tag{F.30}
$$

Sum them up,

$$
\begin{aligned}
&\lambda_1((1 - \frac{\tau}{\lambda_1}) D_{\psi^{\mathcal{C}}}(\boldsymbol{u}_2, \boldsymbol{u}) - D_{\psi^{\mathcal{C}}}(\boldsymbol{u}_2, \boldsymbol{u}_1) - (1 - \frac{\tau}{\lambda_1}) D_{\psi^{\mathcal{C}}}(\boldsymbol{u}_1, \boldsymbol{u})) \\
&+ (\lambda_2 - \lambda_1)(D_{\psi^{\mathcal{C}}}(\boldsymbol{u}_2, \boldsymbol{u}_0) - D_{\psi^{\mathcal{C}}}(\boldsymbol{u}_2, \boldsymbol{u}_1) - D_{\psi^{\mathcal{C}}}(\boldsymbol{u}_1, \boldsymbol{u}_0)) \\
= &\left\langle \lambda_2 \nabla \psi^{\mathcal{C}}(\boldsymbol{u}_1) - \lambda_1(1 - \frac{\tau}{\lambda_t}) \nabla \psi^{\mathcal{C}}(\boldsymbol{u}) - (\lambda_2 - \lambda_1) \nabla \psi^{\mathcal{C}}(\boldsymbol{u}_0), \boldsymbol{u}_2 - \boldsymbol{u}_1 \right\rangle \\
&- \tau \psi^{\mathcal{C}}(\boldsymbol{u}_2) + \tau \psi^{\mathcal{C}}(\boldsymbol{u}_1) \\
\geq &\langle \boldsymbol{g}, \boldsymbol{u}_1 - \boldsymbol{u}_2 \rangle - \tau \psi^{\mathcal{C}}(\boldsymbol{u}_2) + \tau \psi^{\mathcal{C}}(\boldsymbol{u}_1)
\end{aligned}
\tag{F.31}
$$

where the last equation comes from Eq (F.28). $\qquad\square$

**Lemma F.6.** Consider the update rule Eq (4.2). For any information set $h \in \mathcal{H}^{\mathcal{Z}}$, $\boldsymbol{q}_h \in \Delta_{|\Omega_h|}^{\gamma}$ and $t = 1, 2, ..., T$, we have

$$
\begin{aligned}
&\tau \alpha_h \psi^{\Delta}(\boldsymbol{q}_{t+\frac{1}{2}, h}) - \tau \alpha_h \psi^{\Delta}(\boldsymbol{q}_h) + \left\langle V^h(\boldsymbol{z}_{t+\frac{1}{2}}), \boldsymbol{q}_{t+\frac{1}{2}, h} - \boldsymbol{q}_h \right\rangle \\
\leq &(\lambda_t^h - \tau \alpha_h) D_{\psi^{\Delta}}(\boldsymbol{q}_h, \boldsymbol{q}_{t,h}) - \lambda_{t+1}^h D_{\psi^{\Delta}}(\boldsymbol{q}_h, \boldsymbol{q}_{t+1,h}) + (\lambda_{t+1}^h - \lambda_t^h) D_{\psi^{\Delta}}(\boldsymbol{q}_h, \boldsymbol{q}_{1,h}) \\
&+ \left\langle V^h(\boldsymbol{z}_{t+\frac{1}{2}}) - V^h(\boldsymbol{z}_{t-\frac{1}{2}}), \boldsymbol{q}_{t+\frac{1}{2}, h} - \boldsymbol{q}_{t+1,h} \right\rangle - \lambda_t^h D_{\psi^{\Delta}}(\boldsymbol{q}_{t+1,h}, \boldsymbol{q}_{t+\frac{1}{2}, h}) \\
&- (\lambda_t^h - \tau \alpha_h) D_{\psi^{\Delta}}(\boldsymbol{q}_{t+\frac{1}{2}, h}, \boldsymbol{q}_{t,h}).
\end{aligned}
\tag{F.32}
$$

*Proof.* Plug $\boldsymbol{u}_1 = \boldsymbol{q}_{t+\frac{1}{2},h}, \boldsymbol{u}_2 = \boldsymbol{q}_{t+1,h}, \boldsymbol{g} = V^h(\boldsymbol{z}_{t-\frac{1}{2}}), \psi^{\mathcal{C}} = \alpha_h \psi^{\Delta}$ into Lemma C.4,

$$\tau \alpha_h \psi^{\Delta}(\boldsymbol{q}_{t+\frac{1}{2},h}) - \tau \alpha_h \psi^{\Delta}(\boldsymbol{q}_{t+1,h}) + \left\langle V^h(\boldsymbol{z}_{t-\frac{1}{2}}), \boldsymbol{q}_{t+\frac{1}{2},h} - \boldsymbol{q}_{t+1,h} \right\rangle$$
$$\leq \lambda_t^h \left( (1 - \frac{\tau \alpha_h}{\lambda_t^h}) D_{\psi^{\Delta}}(\boldsymbol{q}_{t+1,h}, \boldsymbol{q}_{t,h}) - D_{\psi^{\Delta}}(\boldsymbol{q}_{t+1,h}, \boldsymbol{q}_{t+\frac{1}{2},h}) - (1 - \frac{\tau \alpha_h}{\lambda_t^h}) D_{\psi^{\Delta}}(\boldsymbol{q}_{t+\frac{1}{2},h}, \boldsymbol{q}_{t,h}) \right).$$
(F.33)

Plug $\boldsymbol{u}_1 = \boldsymbol{q}_{t+1,h}, \boldsymbol{u}_2 = \boldsymbol{q}_h, \boldsymbol{g} = V^h(\boldsymbol{z}_{t+\frac{1}{2}}), \lambda_1 = \lambda_t^h, \lambda_2 = \lambda_{t+1}^h, \psi^{\mathcal{C}} = \alpha_h \psi^{\Delta}$ into Lemma F.5,

$$\tau \alpha_h \psi^{\Delta}(\boldsymbol{q}_{t+1,h}) - \tau \alpha_h \psi^{\Delta}(\boldsymbol{q}_h) + \left\langle V^h(\boldsymbol{z}_{t+\frac{1}{2}}), \boldsymbol{q}_{t+1,h} - \boldsymbol{q}_h \right\rangle$$
$$\leq \lambda_t^h ((1 - \frac{\tau \alpha_h}{\lambda_t^h}) D_{\psi^{\Delta}}(\boldsymbol{q}_h, \boldsymbol{q}_{t,h}) - D_{\psi^{\Delta}}(\boldsymbol{q}_h, \boldsymbol{q}_{t+1,h}) - (1 - \frac{\tau \alpha_h}{\lambda_t^h}) D_{\psi^{\Delta}}(\boldsymbol{q}_{t+1,h}, \boldsymbol{q}_{t,h})) \quad \text{(F.34)}$$
$$+ (\lambda_{t+1}^h - \lambda_t^h)(D_{\psi^{\Delta}}(\boldsymbol{q}_h, \boldsymbol{q}_{1,h}) - D_{\psi^{\Delta}}(\boldsymbol{q}_h, \boldsymbol{q}_{t+1,h}) - D_{\psi^{\Delta}}(\boldsymbol{q}_{t+1,h}, \boldsymbol{q}_{1,h})).$$

By summing Eq (F.33) and Eq (F.34) up, then adding $\left\langle V^h(\boldsymbol{z}_{t+\frac{1}{2}}) - V^h(\boldsymbol{z}_{t-\frac{1}{2}}), \boldsymbol{q}_{t+\frac{1}{2},h} - \boldsymbol{q}_{t+1,h} \right\rangle$ on both sides,

$$\tau \alpha_h \psi^{\Delta}(\boldsymbol{q}_{t+\frac{1}{2},h}) - \tau \alpha_h \psi^{\Delta}(\boldsymbol{q}_h) + \left\langle V^h(\boldsymbol{z}_{t+\frac{1}{2}}), \boldsymbol{q}_{t+\frac{1}{2},h} - \boldsymbol{q}_h \right\rangle$$
$$\leq \left\langle V^h(\boldsymbol{z}_{t+\frac{1}{2}}) - V^h(\boldsymbol{z}_{t-\frac{1}{2}}), \boldsymbol{q}_{t+\frac{1}{2},h} - \boldsymbol{q}_{t+1,h} \right\rangle$$
$$+ \lambda_t^h \left( (1 - \frac{\tau \alpha_h}{\lambda_t^h}) D_{\psi^{\Delta}}(\boldsymbol{q}_{t+1,h}, \boldsymbol{q}_{t,h}) - D_{\psi^{\Delta}}(\boldsymbol{q}_{t+1,h}, \boldsymbol{q}_{t+\frac{1}{2},h}) - (1 - \frac{\tau \alpha_h}{\lambda_t^h}) D_{\psi^{\Delta}}(\boldsymbol{q}_{t+\frac{1}{2},h}, \boldsymbol{q}_{t,h})) \right)$$
$$+ \lambda_t^h ((1 - \frac{\tau \alpha_h}{\lambda_t^h}) D_{\psi^{\Delta}}(\boldsymbol{q}_h, \boldsymbol{q}_{t,h}) - D_{\psi^{\Delta}}(\boldsymbol{q}_h, \boldsymbol{q}_{t+1,h}) - (1 - \frac{\tau \alpha_h}{\lambda_t^h}) D_{\psi^{\Delta}}(\boldsymbol{q}_{t+1,h}, \boldsymbol{q}_{t,h}))$$
$$+ (\lambda_{t+1}^h - \lambda_t^h)(D_{\psi^{\Delta}}(\boldsymbol{q}_h, \boldsymbol{q}_{1,h}) - D_{\psi^{\Delta}}(\boldsymbol{q}_h, \boldsymbol{q}_{t+1,h}) - D_{\psi^{\Delta}}(\boldsymbol{q}_{t+1,h}, \boldsymbol{q}_{1,h}))$$
$$\leq \left\langle V^h(\boldsymbol{z}_{t+\frac{1}{2}}) - V^h(\boldsymbol{z}_{t-\frac{1}{2}}), \boldsymbol{q}_{t+\frac{1}{2},h} - \boldsymbol{q}_{t+1,h} \right\rangle$$
$$+ (\lambda_t^h - \tau \alpha_h) D_{\psi^{\Delta}}(\boldsymbol{q}_h, \boldsymbol{q}_{t,h}) - \lambda_{t+1}^h D_{\psi^{\Delta}}(\boldsymbol{q}_h, \boldsymbol{q}_{t+1,h}) + (\lambda_{t+1}^h - \lambda_t^h) D_{\psi^{\Delta}}(\boldsymbol{q}_h, \boldsymbol{q}_{1,h})$$
$$- \lambda_t^h D_{\psi^{\Delta}}(\boldsymbol{q}_{t+1,h}, \boldsymbol{q}_{t+\frac{1}{2},h}) - (\lambda_t^h - \tau \alpha_h) D_{\psi^{\Delta}}(\boldsymbol{q}_{t+\frac{1}{2},h}, \boldsymbol{q}_{t,h}).$$
$\square$

By the two lemmas above, we can prove that the update of Reg-DS-OptMD (4.2) is stable.

**Lemma F.7** (Stability of `Reg-DS-OptMD`). For any $t = 1, 2, ...$, when $\psi^{\Delta}$ is Euclidean norm, Reg-CFR satisfies that

$$\|\boldsymbol{q}_{t-\frac{1}{2},h} - \boldsymbol{q}_{t,h}\| \leq \frac{C_1}{\lambda_{t-1}^h}, \quad \|\boldsymbol{q}_{t+\frac{1}{2},h} - \boldsymbol{q}_{t,h}\| \leq \frac{C_1}{\lambda_t^h}, \quad \text{(F.35)}$$

for some constant $C_1$.

*Proof.* Consider the update rule Eq (4.2), by first-order optimality, for any $h \in \mathcal{H}^{\mathcal{Z}}$, we have

$$\left\langle V^h(\boldsymbol{z}_{t-\frac{1}{2}}) + \lambda_t^h \nabla \psi^{\Delta}(\boldsymbol{q}_{t,h}) - (\lambda_{t-1}^h - \tau) \nabla \psi^{\Delta}(\boldsymbol{q}_{t-1,h}) - (\lambda_t^h - \lambda_{t-1}^h) \nabla \psi^{\Delta}(\boldsymbol{q}_{1,h}), \right.$$
$$\left. \boldsymbol{q}_{t-\frac{1}{2},h} - \boldsymbol{q}_{t,h} \right\rangle \geq 0 \quad \text{(F.36)}$$
$$\left\langle V^h(\boldsymbol{z}_{t-\frac{3}{2}}) + \lambda_{t-1}^h \nabla \psi^{\Delta}(\boldsymbol{q}_{t-\frac{1}{2},h}) - (\lambda_{t-1}^h - \tau) \nabla \psi^{\Delta}(\boldsymbol{q}_{t-1,h}), \boldsymbol{q}_{t,h} - \boldsymbol{q}_{t-\frac{1}{2},h} \right\rangle \geq 0.$$

Add them up,

$$\left\langle \lambda_{t-1}^h \nabla \psi^{\Delta}(\boldsymbol{q}_{t-\frac{1}{2},h}) - \lambda_t^h \nabla \psi^{\Delta}(\boldsymbol{q}_{t,h}) + (\lambda_t^h - \lambda_{t-1}^h) \nabla \psi^{\Delta}(\boldsymbol{q}_{1,h}), \boldsymbol{q}_{t-\frac{1}{2},h} - \boldsymbol{q}_{t,h} \right\rangle$$
$$\leq \left\langle V^h(\boldsymbol{z}_{t-\frac{1}{2}}) - V^h(\boldsymbol{z}_{t-\frac{3}{2}}), \boldsymbol{q}_{t-\frac{1}{2},h} - \boldsymbol{q}_{t,h} \right\rangle. \quad \text{(F.37)}$$

Since $\psi^\Delta$ 1-strong convex with respect to 2-norm, we have

$$\psi^\Delta(\boldsymbol{q}_{t-\frac{1}{2},h}) - \psi^\Delta(\boldsymbol{q}_{t,h}) \geq \left\langle \nabla\psi^\Delta(\boldsymbol{q}_{t,h}), \boldsymbol{q}_{t-\frac{1}{2},h} - \boldsymbol{q}_{t,h} \right\rangle + \frac{1}{2}\|\boldsymbol{q}_{t-\frac{1}{2},h} - \boldsymbol{q}_{t,h}\|^2$$

$$\psi^\Delta(\boldsymbol{q}_{t,h}) - \psi^\Delta(\boldsymbol{q}_{t-\frac{1}{2},h}) \geq \left\langle \nabla\psi^\Delta(\boldsymbol{q}_{t-\frac{1}{2},h}), \boldsymbol{q}_{t,h} - \boldsymbol{q}_{t-\frac{1}{2},h} \right\rangle + \frac{1}{2}\|\boldsymbol{q}_{t-\frac{1}{2},h} - \boldsymbol{q}_{t,h}\|^2. \tag{F.38}$$

Add them up then we will get,

$$\left\langle \nabla\psi^\Delta(\boldsymbol{q}_{t-\frac{1}{2},h}) - \nabla\psi^\Delta(\boldsymbol{q}_{t,h}), \boldsymbol{q}_{t-\frac{1}{2},h} - \boldsymbol{q}_{t,h} \right\rangle \geq \|\boldsymbol{q}_{t-\frac{1}{2},h} - \boldsymbol{q}_{t,h}\|^2. \tag{F.39}$$

Therefore,

$$\lambda_{t-1}^h\|\boldsymbol{q}_{t-\frac{1}{2},h} - \boldsymbol{q}_{t,h}\|^2 + (\lambda_t^h - \lambda_{t-1}^h)\left\langle \nabla\psi^\Delta(\boldsymbol{q}_{1,h}) - \nabla\psi^\Delta(\boldsymbol{q}_{t,h}), \boldsymbol{q}_{t-\frac{1}{2},h} - \boldsymbol{q}_{t,h} \right\rangle$$

$$\leq \left\langle \lambda_{t-1}^h\nabla\psi^\Delta(\boldsymbol{q}_{t-\frac{1}{2},h}) - \lambda_t^h\nabla\psi^\Delta(\boldsymbol{q}_{t,h}) + (\lambda_t^h - \lambda_{t-1}^h)\nabla\psi^\Delta(\boldsymbol{q}_{1,h}), \boldsymbol{q}_{t-\frac{1}{2},h} - \boldsymbol{q}_{t,h} \right\rangle$$

$$\leq \left\langle V^h(\boldsymbol{z}_{t-\frac{1}{2}}) - V^h(\boldsymbol{z}_{t-\frac{3}{2}}), \boldsymbol{q}_{t-\frac{1}{2},h} - \boldsymbol{q}_{t,h} \right\rangle$$

$$\leq \|V^h(\boldsymbol{z}_{t-\frac{1}{2}}) - V^h(\boldsymbol{z}_{t-\frac{3}{2}})\| \cdot \|\boldsymbol{q}_{t-\frac{1}{2},h} - \boldsymbol{q}_{t,h}\|. \tag{F.40}$$

And by definition,

$$\lambda_{t-1}^h \leq \lambda_t^h = \sqrt{(\lambda_{t-1}^h)^2 + \|V^h(\boldsymbol{z}_{t-\frac{1}{2}}) - V^h(\boldsymbol{z}_{t-\frac{3}{2}})\|^2} \leq \lambda_{t-1}^h + \|V^h(\boldsymbol{z}_{t-\frac{1}{2}}) - V^h(\boldsymbol{z}_{t-\frac{3}{2}})\|, \tag{F.41}$$

so that

$$\lambda_{t-1}^h\|\boldsymbol{q}_{t-\frac{1}{2},h}-\boldsymbol{q}_{t,h}\|^2 \leq (\|\nabla\psi^\Delta(\boldsymbol{q}_{1,h})-\nabla\psi^\Delta(\boldsymbol{q}_{t,h})\|+1)\cdot\|V^h(\boldsymbol{z}_{t-\frac{1}{2}})-V^h(\boldsymbol{z}_{t-\frac{3}{2}})\|\cdot\|\boldsymbol{q}_{t-\frac{1}{2},h}-\boldsymbol{q}_{t,h}\| \tag{F.42}$$

which implies that

$$\|\boldsymbol{q}_{t-\frac{1}{2},h} - \boldsymbol{q}_{t,h}\| \leq \frac{O(1)}{\lambda_{t-1}^h}, \tag{F.43}$$

since $\nabla\psi^\Delta$ is bounded by constant when $\psi^\Delta$ is Euclidean norm. And $\|V^h(\boldsymbol{z}_{t-\frac{1}{2}}) - V^h(\boldsymbol{z}_{t-\frac{3}{2}})\|$ is also bounded by constant since both the regularizer and $\|F(\boldsymbol{z})\|_\infty$ are bounded.

At the same time, directly from update rule Eq (4.2),

$$\left\langle V^h(\boldsymbol{z}_{t-\frac{1}{2}}) + \tau\nabla\psi^\Delta(\boldsymbol{q}_{t,h}), \boldsymbol{q}_{t+\frac{1}{2},h} \right\rangle + \lambda_t^h D_{\psi^\Delta}(\boldsymbol{q}_{t+\frac{1}{2},h}, \boldsymbol{q}_{t,h}) \leq \left\langle V^h(\boldsymbol{z}_{t-\frac{1}{2}}) + \tau\nabla\psi^\Delta(\boldsymbol{q}_{t,h}), \boldsymbol{q}_{t,h} \right\rangle \tag{F.44}$$

which implies that

$$\frac{\lambda_t^h}{2}\|\boldsymbol{q}_{t+\frac{1}{2},h} - \boldsymbol{q}_{t,h}\|^2 \leq \lambda_t^h D_{\psi^\Delta}(\boldsymbol{q}_{t+\frac{1}{2},h}, \boldsymbol{q}_{t,h}) \leq \left\langle V^h(\boldsymbol{z}_{t-\frac{1}{2}}) + \tau\nabla\psi^\Delta(\boldsymbol{q}_{t,h}), \boldsymbol{q}_{t,h} - \boldsymbol{q}_{t+\frac{1}{2},h} \right\rangle$$

$$\leq \|V^h(\boldsymbol{z}_{t-\frac{1}{2}}) + \tau\nabla\psi^\Delta(\boldsymbol{q}_{t,h})\| \cdot \|\boldsymbol{q}_{t,h} - \boldsymbol{q}_{t+\frac{1}{2},h}\|$$

$$\leq O(1)\|\boldsymbol{q}_{t,h} - \boldsymbol{q}_{t+\frac{1}{2},h}\|. \tag{F.45}$$

Hence, we have

$$\|\boldsymbol{q}_{t+\frac{1}{2},h} - \boldsymbol{q}_{t,h}\| \leq \frac{O(1)}{\lambda_t^h}. \qquad \square$$

*Proof of Lemma F.2.* By Lemma F.10,

$$\|V^h(\boldsymbol{z}_{t+\frac{1}{2}}) - V^h(\boldsymbol{z}_{t-\frac{1}{2}})\|^2 \leq (L_2\sum_{h\in\mathcal{H}^z}\|\boldsymbol{q}_{t+\frac{1}{2},h} - \boldsymbol{q}_{t-\frac{1}{2},h}\|)^2 \leq P^2 L_2^2\frac{C_1^2}{(\lambda_{t-1}^h)^2}$$

where the last inequality is by and Lemma F.7 and

$$\|\boldsymbol{q}_{t+\frac{1}{2},h} - \boldsymbol{q}_{t-\frac{1}{2},h}\| \leq \|\boldsymbol{q}_{t+\frac{1}{2},h} - \boldsymbol{q}_{t,h}\| + \|\boldsymbol{q}_{t,h} - \boldsymbol{q}_{t-\frac{1}{2},h}\| \leq \frac{C_1}{\lambda_{t-1}^h}.$$

Then, by letting $\kappa = T^{\frac{1}{2}}$, we have

$$
\lambda_{T+1}^h M^h + \sum_{t=2}^T \frac{\|V^h(z_{t+\frac{1}{2}}) - V^h(z_{t-\frac{1}{2}})\|^2}{\lambda_t^h}
$$

$$
\leq \sqrt{T^{\frac{1}{2}} + \sum_{t=1}^T \frac{P^2 L_2^2 C_1^2}{(\lambda_{t-1}^h)^2}} M^h + \sum_{t=2}^T \frac{P^2 L_2^2 C_1^2}{\lambda_t^h (\lambda_{t-1}^h)^2}
$$

$$
\leq O(1) \cdot \sqrt{T^{\frac{1}{2}} + T \cdot T^{-\frac{1}{2}}} + O(1) \cdot T \cdot T^{-\frac{3}{4}}
$$

$$
\leq O(T^{\frac{1}{4}}),
$$

which completes the proof. □

### F.6 Auxiliary lemmas for Reg-CFR

In this section, we prove some auxiliary lemmas for Reg-CFR. We begin with the expanding form of the Bregman divergence generated by the dilated Euclidean norm.

**Lemma F.8.** When $\psi^\Delta(q) = \frac{1}{2} \sum_i q_i^2$, we have

$$
D_{\psi^z}(z_1, z_2) = \sum_{h \in \mathcal{H}^z} \frac{\alpha_h}{2} z_{1,\sigma(h)} \|\frac{z_{1,h}}{z_{1,\sigma(h)}} - \frac{z_{2,h}}{z_{2,\sigma(h)}}\|^2 = \sum_{h \in \mathcal{H}^z} \alpha_h z_{1,\sigma(h)} D_{\psi^\Delta}(q_{1,h}, q_{2,h}). \quad \text{(F.46)}
$$

*Proof.* Firstly, we can write $\psi^\mathcal{Z}$ in the form

$$
\psi^\mathcal{Z}(z) = \sum_i \frac{\alpha_{h(i)}}{2} \cdot \frac{z_i^2}{\sum_{j \in \Omega_{h(i)}} z_j}, \quad \text{(F.47)}
$$

then $\frac{\partial \psi^\mathcal{Z}(z)}{\partial z_i}$ will be

$$
\frac{\partial \psi^\mathcal{Z}(z)}{\partial z_i} = \frac{\alpha_{h(i)}}{2} \left[ \frac{2z_i}{\sum_{j \in \Omega_{h(i)}} z_j} - \sum_{k \in \Omega_{h(i)}} \frac{z_k^2}{\left(\sum_{j \in \Omega_{h(i)}} z_j\right)^2} \right] = \frac{\alpha_{h(i)}}{2} \left[ 2q_i - \sum_{k \in \Omega_{h(i)}} q_k^2 \right] \quad \text{(F.48)}
$$

where $q_i = \frac{z_i}{z_{\sigma(h(i))}}$.

And by the definition of Bregman divergence, we have

$$
\begin{aligned}
D_{\psi^z}(z_1, z_2) =& \psi^\mathcal{Z}(z_1) - \psi^\mathcal{Z}(z_2) - \langle \nabla \psi^\mathcal{Z}(z_2), z_1 - z_2 \rangle \\
=& \sum_i \frac{\alpha_{h(i)}}{2} z_{1,i}(q_{1,i} - 2q_{2,i} + \sum_{k \in \Omega_{h(i)}} q_{2,k}^2) \\
& - \sum_i \frac{\alpha_{h(i)}}{2} z_{2,i}(q_{2,i} - 2q_{2,i} + \sum_{k \in \Omega_{h(i)}} q_{2,k}^2)
\end{aligned} \quad \text{(F.49)}
$$

Notice that

$$
\begin{aligned}
\sum_{i \in \Omega_h} z_{1,i}(q_{1,i} - 2q_{2,i} + \sum_{k \in \Omega_{h(i)}} q_{2,k}^2) &= \sum_{i \in \Omega_h} z_{1,i}(q_{1,i} - 2q_{2,i}) + \sum_{i \in \Omega_h} z_{1,i} \sum_{k \in \Omega_{h(i)}} q_{2,k}^2 \\
&= \sum_{i \in \Omega_h} z_{1,\sigma(h(i))}(q_{1,i}^2 - 2q_{1,i}q_{2,i}) + z_{1,\sigma(h(i))} \sum_{k \in \Omega_{h(i)}} q_{2,k}^2 \\
&= \sum_{i \in \Omega_h} z_{1,\sigma(h(i))}(q_{1,i} - q_{2,i})^2.
\end{aligned} \quad \text{(F.50)}
$$

Similarly, we will get $\sum_{i \in \Omega_h} z_{2,i}(q_{2,i} - 2q_{2,i} + \sum_{k \in \Omega_{h(i)}} q_{2,k}^2) = 0$. Therefore, $D_{\psi^z}(z_1, z_2) = \sum_{h \in \mathcal{H}^z} \frac{\alpha_h}{2} z_{1,\sigma(h)} \|\frac{z_{1,h}}{z_{1,\sigma(h)}} - \frac{z_{2,h}}{z_{2,\sigma(h)}}\|^2$. □

**Lipschitz continuity of $V^h(z)$**    Here we will show that $V^h(z)$ is Lipschitz continuous with respect to $q$. We first show that $z$ is in a Lipschitz continuous manner with respect to $q$.

**Lemma F.9.** In $\gamma$−perturbed treeplex $\mathcal{Z}^\gamma$ with $\gamma \geq 0$ , for any $z, z' \in \mathcal{Z}^\gamma$, we have

$$\|z - z'\| \leq L_1 \sum_{h \in \mathcal{H}^\mathcal{Z}} \|q_h - q'_h\| \tag{F.51}$$

for some game-dependent constant $L_1$.

*Proof.* We first consider the base case when $\mathcal{Z}^\gamma$ is a $\gamma$-perturbed simplex, where Eq (F.51) is satisfied with $L_1 = 1$ since $q = z$.

We consider the two basic operator, Cartesian product and branching, in the definition of treeplex (see Definition 2.1 for details). We want to prove that both of them keep smoothness, that is, Eq (F.51) remains satisfied after applying the operation to multiple treeplexes where Eq (F.51) is satisfied.

Firstly, for Cartesian product , if Eq (F.51) is satisfied for $\mathcal{Z}^\gamma_1, \mathcal{Z}^\gamma_2, ..., \mathcal{Z}^\gamma_m$, then for any $z = (z_1, z_2, ..., z_m), z' = (z'_1, z'_2, ..., z'_m) \in \mathcal{Z}^\gamma = \mathcal{Z}^\gamma_1 \times \mathcal{Z}^\gamma_2 \times ... \times \mathcal{Z}^\gamma_m$, we have

$$\|z - z'\| \leq \sum_{i=1}^m \|z_i - z'_i\| \leq L_1 \sum_{i=1}^m \sum_{h \in \mathcal{H}^{\mathcal{Z}_i}} \|q_h - q'_h\| = L_1 \sum_{h \in \mathcal{H}^\mathcal{Z}} \|q_h - q'_h\|. \tag{F.52}$$

Notice that we abuse the notation $\mathcal{H}^{\mathcal{Z}_i}$ and $\mathcal{H}^\mathcal{Z}$ here to denote the set of all information sets in $\mathcal{Z}^\gamma_i$ and $\mathcal{Z}^\gamma$.

To be convenient, let define the branching of $m$ $\gamma$−perturbed treeplexes $\mathcal{Z}^\gamma_1, \mathcal{Z}^\gamma_2, ..., \mathcal{Z}^\gamma_m$ and a $\gamma$-perturbed simplex $\Delta^\gamma_m$ be $\mathcal{Z}^\gamma = \{(p, p_1z_1, p_2z_2, ..., p_mz_m) : p \in \Delta^\gamma_m, z_i \in \mathcal{Z}^\gamma_i\}$. It's easy to see that it is equivalent to using the original branching operator for $m$ times in a bottom-up manner.

Suppose for any $z_i, z'_i \in \mathcal{Z}^\gamma_i$, $\|z_i - z'_i\| \leq L_i \sum_{h \in \mathcal{H}^{\mathcal{Z}_i}} \|q_{i,h} - q'_{i,h}\|$. For $z = (p, p_1z_1, p_2z_2 + ..., p_mz_m), z' = (p', p'_1z'_1, p'_2z'_2, ..., p'_mz'_m) \in \mathcal{Z}^\gamma$,

$$\|z - z'\| \leq \sum_{i=1}^m \|p_iz_i - p'_iz'_i\| + \|p - p'\| \tag{F.53}$$

And we have

$$\begin{aligned} \|p_iz_i - p'_iz'_i\| &= \|(p_iz_i - p_iz'_i) + (p_iz'_i - p'_iz'_i)\| \\ &\leq \|p_i(z_i - z'_i)\| + \|(p_i - p'_i)z'_i\| \\ &= p_i\|z_i - z'_i\| + |p_i - p'_i| \cdot \|z'_i\| \\ &\leq \|z_i - z'_i\| + |\mathcal{Z}^\gamma_i| \cdot \|p - p'\| \end{aligned} \tag{F.54}$$

where the fourth inequality is by $\mathcal{Z}^\gamma_i \subset \mathbb{R}^{|\mathcal{Z}^\gamma_i|}$.

Therefore,

$$\begin{aligned} &\|(p_1z_1 + p_2z_2 + ... + p_mz_m) - (p'_1z'_1 + p'_2z'_2 + ... + p'_mz'_m)\| \\ &\leq \sum_{i=1}^m \|z_i - z'_i\| + (\sum_{i=1}^m |\mathcal{Z}^\gamma_i| + 1)\|p - p'\| \\ &\leq \sum_{i=1}^m L_i \sum_{h \in \mathcal{H}^{\mathcal{Z}_i}} \|q_{i,h} - q'_{i,h}\| + (P + 1)\|p - p'\| \\ &\leq \max\{L_1, L_2, ..., L_m, P + 1\} \sum_{h \in \mathcal{H}^\mathcal{Z}} \|q_h - q'_h\| \end{aligned} \tag{F.55}$$

where the third line is by the inductive assumption and the fourth line is by definition of $\mathcal{H}^\mathcal{Z}$ and $q$.

Therefore, recursively applying Eq (F.52) and Eq (F.55), we will have for any $z, z' \in \mathcal{Z}^\gamma$

$$\|z - z'\| \leq L_1 \sum_{h \in \mathcal{H}^\mathcal{Z}} \|q_h - q'_h\| \tag{F.56}$$

where we take $L_1 = P + 1$.

Finally, since both operator keeps the smoothness, by induction, we know that for any treeplex Eq (F.51) is satisfied. $\qquad\square$

Now we can prove that $V^h(\boldsymbol{z})$ is Lipschitz continuous with respect to $\boldsymbol{q}$.

**Lemma F.10.** When $\psi^\Delta$ is $L_p$-Lipschitz continuous, that is, $|\psi^\Delta(\boldsymbol{x}) - \psi^\Delta(\boldsymbol{x}')| \le L_p \|\boldsymbol{x} - \boldsymbol{x}'\|$ for any $\boldsymbol{x}, \boldsymbol{x}' \in \Delta^\gamma$, and $|\psi^\Delta(\boldsymbol{x})|$ is upper-bounded by a constant $C_B^\Delta$ for any $\boldsymbol{x} \in \Delta^\gamma$, then for any $\boldsymbol{z}, \boldsymbol{z}' \in \mathcal{Z}^\gamma$ and $h \in \mathcal{H}$, we have

$$\|V^h(\boldsymbol{z}) - V^h(\boldsymbol{z}')\| \le L_2 \sum_{h \in \mathcal{H}^{\mathcal{Z}}} \|\boldsymbol{q}_h - \boldsymbol{q}'_h\| \tag{F.57}$$

where $L_2$ is a game-dependent constant.

*Proof.* Here we consider $h \in \mathcal{H}^{\mathcal{X}}$, and $h \in \mathcal{H}^{\mathcal{Y}}$ can be addressed similarly.

$$\begin{aligned}
\|V^h(\boldsymbol{z}) - V^h(\boldsymbol{z}')\| &\le \sum_{i \in \Omega_h} \left( |(\boldsymbol{A}(\boldsymbol{y} - \boldsymbol{y}'))_i| + \sum_{h' \in \mathcal{H}_i} |W^{h'}(\boldsymbol{z}) - W^{h'}(\boldsymbol{z}')| \right) \\
&\le \sum_{i \in \Omega_h} \left( \|\boldsymbol{y} - \boldsymbol{y}'\|_1 + \sum_{h' \in \mathcal{H}_i} |W^{h'}(\boldsymbol{z}) - W^{h'}(\boldsymbol{z}')| \right) \\
&\le \sum_{i \in \Omega_h} \left( P\|\boldsymbol{y} - \boldsymbol{y}'\| + \sum_{h' \in \mathcal{H}_i} |W^{h'}(\boldsymbol{z}) - W^{h'}(\boldsymbol{z}')| \right)
\end{aligned} \tag{F.58}$$

where the second inequality is because each entry of $\boldsymbol{A}$ is in $[-1, 1]$ and the last inequality is by $\|\boldsymbol{y} - \boldsymbol{y}'\|_1 \le \sqrt{P}\|\boldsymbol{y} - \boldsymbol{y}'\|$.

$$\begin{aligned}
&|W^h(\boldsymbol{z}) - W^h(\boldsymbol{z}')| \\
&= \left| (\langle \boldsymbol{q}_h, V^h(\boldsymbol{z}) \rangle + \tau \alpha_h \psi^\Delta(\boldsymbol{q}_h)) - (\langle \boldsymbol{q}'_h, V^h(\boldsymbol{z}') \rangle + \tau \alpha_h \psi^\Delta(\boldsymbol{q}'_h)) \right| \\
&\le \left| \langle \boldsymbol{q}_h, V^h(\boldsymbol{z}) \rangle - \langle \boldsymbol{q}'_h, V^h(\boldsymbol{z}') \rangle \right| + \tau \alpha_h \left| \psi^\Delta(\boldsymbol{q}_h) - \psi^\Delta(\boldsymbol{q}'_h) \right| \\
&\le |\langle \boldsymbol{q}_h, V^h(\boldsymbol{z}) - V^h(\boldsymbol{z}') \rangle| + |\langle \boldsymbol{q}_h - \boldsymbol{q}'_h, V^h(\boldsymbol{z}') \rangle| + \tau \|\boldsymbol{\alpha}\|_\infty L_p \|\boldsymbol{q}_h - \boldsymbol{q}'_h\| \\
&\overset{(i)}{\le} \|\boldsymbol{q}_h\|_1 \cdot \|V^h(\boldsymbol{z}) - V^h(\boldsymbol{z}')\|_\infty + \|\boldsymbol{q}_h - \boldsymbol{q}'_h\| \cdot \|V^h(\boldsymbol{z}')\| + \tau \|\boldsymbol{\alpha}\|_\infty L_p \|\boldsymbol{q}_h - \boldsymbol{q}'_h\| \\
&\overset{(ii)}{\le} \sum_{i \in \Omega_h} |(\boldsymbol{A}(\boldsymbol{y} - \boldsymbol{y}'))_i| + \sum_{i \in \Omega_h} \sum_{h' \in \mathcal{H}_i} |W^{h'}(\boldsymbol{z}) - W^{h'}(\boldsymbol{z}')| \\
&\quad + P(1 + \tau \|\boldsymbol{\alpha}\|_\infty C_B^\Delta)\|\boldsymbol{q}_h - \boldsymbol{q}'_h\| + \tau \|\boldsymbol{\alpha}\|_\infty L_p \|\boldsymbol{q}_h - \boldsymbol{q}'_h\|.
\end{aligned} \tag{F.59}$$

Here $(i)$ is by Hölder's inequality. $(ii)$ is by $\|\boldsymbol{q}_h\|_1 = 1$ and $\|V^h(\boldsymbol{z})\| \le \|\boldsymbol{A}\boldsymbol{y}\|_1 + P\tau\|\boldsymbol{\alpha}\|_\infty C_B^\Delta \le P(1 + \tau\|\boldsymbol{\alpha}\|_\infty C_B^\Delta)$. By recursively applying this inequality, we have

$$\begin{aligned}
&|W^h(\boldsymbol{z}) - W^h(\boldsymbol{z}')| \\
&\le \|\boldsymbol{A}(\boldsymbol{y} - \boldsymbol{y}')\|_1 + P(1 + \tau\|\boldsymbol{\alpha}\|_\infty C_B^\Delta) \sum_{h \in \mathcal{H}^{\mathcal{Z}}} \|\boldsymbol{q}_h - \boldsymbol{q}'_h\| + \tau\|\boldsymbol{\alpha}\|_\infty L_p \sum_{h \in \mathcal{H}^{\mathcal{Z}}} \|\boldsymbol{q}_h - \boldsymbol{q}'_h\|
\end{aligned} \tag{F.60}$$

Notice that

$$\|\boldsymbol{A}(\boldsymbol{y} - \boldsymbol{y}')\|_1 \le P\|\boldsymbol{y} - \boldsymbol{y}'\|_1 \le P^2\|\boldsymbol{y} - \boldsymbol{y}'\| \le L_1 P^2 \sum_{h \in \mathcal{H}^{\mathcal{Z}}} \|\boldsymbol{q}_h - \boldsymbol{q}'_h\| \tag{F.61}$$

where the first inequality is because $\boldsymbol{A} \in [-1, 1]^{M \times N}$ and the third inequality is by Lemma F.9. Therefore,

$$|W^h(\boldsymbol{z}) - W^h(\boldsymbol{z}')|$$

$$\leq P^2 L_1 \sum_{h \in \mathcal{H}^{\mathcal{Z}}} \|\boldsymbol{q}_h - \boldsymbol{q}'_h\| + P(1 + \tau\|\boldsymbol{\alpha}\|_\infty C_B^\Delta) \sum_{h \in \mathcal{H}^{\mathcal{Z}}} \|\boldsymbol{q}_h - \boldsymbol{q}'_h\| + \tau\|\boldsymbol{\alpha}\|_\infty L_p \sum_{h \in \mathcal{H}^{\mathcal{Z}}} \|\boldsymbol{q}_h - \boldsymbol{q}'_h\|$$

$$= L_3 \sum_{h \in \mathcal{H}^{\mathcal{Z}}} \|\boldsymbol{q}_h - \boldsymbol{q}'_h\| \tag{F.62}$$

where $L_3 = P^2 L_1 + P(1 + \tau\|\boldsymbol{\alpha}\|_\infty C_B^\Delta) + \tau\|\boldsymbol{\alpha}\|_\infty L_p$.

And back to Eq (F.58), we have

$$\|V^h(\boldsymbol{z}) - V^h(\boldsymbol{z}')\| \leq C_\Omega \cdot P\|\boldsymbol{z} - \boldsymbol{z}'\| + P \cdot L_3 \sum_{h \in \mathcal{H}^{\mathcal{Z}}} \|\boldsymbol{q}_h - \boldsymbol{q}'_h\|$$

$$\leq P(L_1 C_\Omega + L_3) \sum_{h \in \mathcal{H}^{\mathcal{Z}}} \|\boldsymbol{q}_h - \boldsymbol{q}'_h\| \tag{F.63}$$

$$= L_2 \sum_{h \in \mathcal{H}^{\mathcal{Z}}} \|\boldsymbol{q}_h - \boldsymbol{q}'_h\|,$$

where the second inequality comes from Lemma F.9. $\qquad\square$

## G   CONCLUSIONS AND FUTURE WORK

In this paper, we investigate the regularization technique, a widely used one in reinforcement learning and optimization, in solving EFGs. Firstly, we prove that `Reg-DOMD` can achieve the first result of last-iterate convergence rate to the NE without the unique NE assumption, for dilated OMD-type algorithms with constant stepsizes, in terms of both duality gap and the distance to the set of NE. We further prove that by solving the regularized problem, CFR with `Reg-DS-OptMD` as regret minimizer, which we called `Reg-CFR`, can achieve best-iterate convergence result in finding NEs and asymptotic last-iterate convergence in finding approximate extensive-form perfect equilibria. These results constitute the first last-iterate convergence results for CFR-type algorithms. Furthermore, we have shown empirically that for CFR and CFR+, solving the regularized problem can achieve better last-iterate performance, further demonstrating the power of regularization in solving EFGs. We leave it for future work to study its explicit convergence rate.

