# OpenReview forum: "The Power of Regularization in Solving Extensive-Form Games"
_ICLR.cc/2023/Conference — ICLR 2023 poster_

### Official Review · Reviewer_r8KK · 2022-10-25

**Confidence:** 4
**Correctness:** 3
**Technical Novelty And Significance:** 3
**Empirical Novelty And Significance:** 2
**Recommendation:** 5

**Clarity, Quality, Novelty And Reproducibility:**

CLARITY: The paper is well written, even though is quite hard to follow.

QUALITY: As far as I am concerned, most of the results are correct, I only have some concerns on the convergence to EFPE.

NOVELTY: Most of the results are novel, but the adopted techniques are not.

REPRODUCIBILITY: Proofs and experiments are fairly reproducible.

**Strength And Weaknesses:**

STRENGTHS

1) The problem of designing last-iterate convergent no-regret algorithms is interesting and it is receiving a lot of attention from the literature.

2) The paper is well written.

WEAKNESSES

1) The paper is hard to follow as it comes as a list of results, which are indeed related, but require the adoption of different perspectives and techniques, making the reader uncomfortable.

2) Theorem 3.1 is rather straightforward to derive, since it directly follows from the results in [Cen et al. 2021b]. Am I right? The other results require a straightforward application of some well-known techniques, such as the laminar-regret decomposition by Farina et al. 2019.

3) I have some concerns about the claim that the Reg-CFR algorithm converges to an EFPE. In particular, as far as I am concerned, the key technical result to prove this is Lemma F.3, but this only predicates on the duality gap. This should be related to the distance to a Nash equilibrium of the game, but it does not say anything about convergence to EFPE (since in zero-sum games all the Nash equilibria, including EFPEs, have the same value that is the value of the game). Am I right or am I missing something?

**Summary Of The Paper:**

This paper investigates the problem of learning a NE in two-players zero-sum games via no-regret algorithms. While most of the state-of-the-art algorithms guarantee convergence in terms of average strategies, existent methods that guarantee last-iterate convergence require strong assumptions like the uniqueness of the NE. In the present work, the authors provide regularization methods that allow to achieve last-iterate convergence without such assumptions. The first algorithm they propose is a regularized version of Online Mirror Descent that optimizes a strongly convex objective directly on the whole strategy polytope of the player. In the second part of the paper, the authors rely on the laminar regret decomposition to decompose the regret in additive terms at each information set to be minimized separately, claiming convergence of such method to a perfect equilibrium. The authors provide proofs of convergence with associated convergence rates. Finally, an experimental evaluation to support the theoretical claims is presented.

**Summary Of The Review:**

Overall, the paper is well written, even though is quite hard to follow. The technical results are interesting, but not exceptional. I also have some concerns on a claim in the paper, namely that related to the coverage to EFPE.

---

> ### Author Response · Authors · 2022-11-09
> **Response to Reviewer r8KK**
>
> **Q**: The paper is hard to follow as it comes as a list of results, which are indeed related, but require the adoption of different perspectives and techniques, making the reader uncomfortable.
>
>   **A**: Sorry for making it a bit hard to follow.  The paper indeed contains rich results that are all induced by the use of the regularization techniques in EFG (as our title accurately highlighted). Using **regularization** as the keyword to unify the results was the best tradeoff we were able to have, in order to include all the interesting results. Other concrete suggestions about organization and presentation would be much appreciated, and we are more than happy to revise the paper accordingly.
>
>
> **Q**: Theorem 3.1 is only an easy derivation from Cen et al.
>
> **A**: It is non-trivial to extend [Cen et al. (2021)] in the normal-form game setting, to the EFG setting, since some key lemmas in [Cen et al. (2021)] utilized the properties of matrix games which do not hold in EFGs. For example, a core observation in [Cen et al. (2021)] is that the $\tau$-NE satisfies that $x^*_{\tau,i}\propto \exp([\textbf{Ay}^*_\tau]_i/\tau)$ which does not hold in EFGs. Additionally, the base of the proof Lemma 1 and Lemma 2 in [Cen et al. (2021)] also fails to hold in EFGs, since  they both used the property of the $\tau$-NE mentioned above. Therefore, we are actually fundamentally different from Cen et al.
>
> Moreover, the key point of the first part is **Theorem 3.2**, which provides convergence guarantee on the **distance to the NE set of the original problem**. This result is new even for NFGs, which we believe resolves the problem of achieving iterate-convergence for policy extragradient algorithms with multiplicative updates without the unique NE assumption.
>
> **Q**: The second part is a straightforward adoption of laminar regret decomposition.
>
> **A**: In fact, laminar regret decomposition is only a generalization of the well-known CFR framework, and we use it to make the presentation of our result in the second part cleaner and more intuitive. We want to emphasize that laminar regret decomposition **does not** guarantee any last-iterate convergence, but the regularization we added enables it. Also, we **did not** claim it as any of our contributions.
>
> We would like to prove that **under the CFR framework (laminar regret decomposition)**, we can also get last-iterate convergence guarantee by adding regularization, which was not established in the literature, including the original laminar regret decomposition paper.
>
> Now we would like to re-emphasize the significance of getting **last-iterate** convergence in the CFR framework.
> In the past, when using deep neural network to do function approximation under CFR framework, the average-iterate convergence is actually a big problem. The seminal work Deep-CFR [Brown et al. (2019)] trained an additional network to output the average policy, which incurs an additional approximation error. [Steinberger (2019), Steinberger et al. (2020)] stores neural network at **every** iteration on disk, which would be impractical for large games (typically it takes more than $10^5$ iterations to converge for CFR). To get the average-iterate convergence, they will sample one neural network and totally follow the policy of that network. This is why we believe that the second part would be a great improvement on those Deep CFR work.
>
> **Q**: Why do we only give duality gap guarantee in Lemma F.3 on convergence to EFPE.
>
> **A**: Sorry for the confusion. In the submitted version, we do not claim we can converge to EFPE. For instance, in the abstract, we claim that "finding the NE of perturbed EFGs, which is useful for finding approximate extensive-form perfect equilibria (EFPE)". We noticed the potential confusion caused by the name approximate EFPE and we modified all *approximate EFPE* to *NE of the perturbed EFG* in our latest version (highlighted in blue).
>
> Lastly, to the best of the authors' knowledge, Lemma F.3 (Now it is Lemma F.4 in the latest version) is the first relationship between (the approximate) NE in a perturbed EFG and EFPE, which has been justified empirically in existing works, where people use NE in perturbed EFG to approximate EFPE, see e.g., [Kroer et al. (2017) Farina et al. (2017)].

---

> > ### Author Response · Authors · 2022-11-09
> > **Response to Reviewer r8KK (Reference)**
> >
> > [1] Brown N, Lerer A, Gross S, et al. Deep counterfactual regret minimization[C]//International conference on machine learning. PMLR, 2019: 793-802.
> >
> > [2] Steinberger E. Single deep counterfactual regret minimization[J]. arXiv preprint arXiv:1901.07621, 2019.
> >
> > [3] Steinberger E, Lerer A, Brown N. DREAM: Deep regret minimization with advantage baselines and model-free learning[J]. arXiv preprint arXiv:2006.10410, 2020.
> >
> > [4] Kroer C, Farina G, Sandholm T. Smoothing method for approximate extensive-form perfect equilibrium[J]. arXiv preprint arXiv:1705.09326, 2017.
> >
> > [5] Farina G, Kroer C, Sandholm T. Regret minimization in behaviorally-constrained zero-sum games[C]//International Conference on Machine Learning. PMLR, 2017: 1107-1116.

---

> ### Author Response · Authors · 2022-11-17
> **Any Further Questions?**
>
> Dear Reviewer, Thank you very much again for the comments. We were wondering if there are any remaining changes you would like us to make or questions you would like us to answer? We are more than happy to address them (before the rebuttal deadline ends on Friday). Otherwise, could you take a chance to reevaluate our paper, and update your score accordingly? Thanks!

---

> ### Author Response · Authors · 2022-12-07
> **Any Further Questions?**
>
> Dear Reviewer,
>
> Thank you very much again for the comments.
>
> We were wondering if there are any remaining changes you would like us to make or questions you would like us to answer? We are more than happy to address them (before the rebuttal deadline ends on Dec 12). Otherwise, could you take a chance to reevaluate our paper, and update your score accordingly?
>
> Thanks!
>
> The Authors

---

### Official Review · Reviewer_D4YS · 2022-10-26

**Confidence:** 2
**Clarity, Quality, Novelty And Reproducibility:** Please see my comments above for the …
**Correctness:** 4
**Technical Novelty And Significance:** 3
**Empirical Novelty And Significance:** Not applicable
**Recommendation:** 5

**Strength And Weaknesses:**

Strength:
  The paper shows the impact of adding specific regularization terms to different algorithms in solving extensive-form games, which either improves the convergence rate or has weaker assumptions than before.

Weakness:
   There has been some previous work like Cen et al. (Reg-OMWU) in exploring the usage of regularization term. Since it's in the same line of work (trying to get benefits by adding regularization), the paper did not explain if there is any connection between the results in this paper and the previous work results. More specifically, the most importance step to achieve all the results presented in the paper is to connect the regularized update back to the original un-regularized problem with iterate convergence. Although the work in Cen et al. is for NFGs, I am not quite sure if the extension to EFGs is incremental or challenging.

**Summary Of The Paper:**

The paper investigates the impact of regularization in solving extensive-form games (EFGs). It shows that after adding the regularization terms to different pay-off functions, several convergence  results could be achieved with either improved convergence rate or weaker assumptions. More specifically, for the dilated optimistic mirror descent, the regularized version can achieve a fast O(1/T ) last-iterate convergence in terms of duality gap without the uniqueness assumption of the Nash equilibrium (NE). For the counterfactual regret minimization, adding regularization term helps get the first last-iterate convergence results for CFR type algorithms.

**Summary Of The Review:**

Overall I think the paper result is interesting and meaningful in further improving the state of the art works by add specific regularization terms. But it would be better to clearly build the connection between this work and the previous work in using the regularization term to really show that the effort is not trivial or incremental.

---

> ### Author Response · Authors · 2022-11-09
> **Response to Reviewer D4YS**
>
> **Q**: Comparison with Cen et al.
>
> **A**: Most importantly, when connecting back to the original problem, Cen et al. only gave the convergence guarantee in terms of **duality gap**, while we give both duality gap and **distance** to the NE set for the original problem. To our best knowledge, this is the first result even in NFGs for policy extragradient algorithms with multiplicative updates to get a convergence guarantee of distance to NE set without unique NE assumption (for the original problem, Cen et al. only achieved duality gap convergence guarantee).
>
>
> Moreover, it is even non-trivial to extend the result in the regularized game of Cen et al. to the EFG setting, since some key lemmas in [Cen et al. (2021)] utilized the properties of matrix games which do not hold in EFGs. For example, a core observation in [Cen et al. (2021)] is that the $\tau$-NE satisfies that $x^*_{\tau,i}\propto \exp([\textbf{Ay}^*_\tau]_i/\tau)$ which does not hold in EFGs. Additionally, the base of the proof Lemma 1 and Lemma 2 in [Cen et al. (2021)] also fails to hold in EFGs, since  they both used the property of the $\tau$-NE mentioned above. Therefore, we are actually fundamentally different from Cen et al.
>
> Lastly, the second part of our paper, about the last-iterate convergence for CFR framework, is **totally different from/irrelevant to** the setting of Cen et al. In Cen et al., they considered the last-iterate convergence of multiplicative update rules (OMWU) in NFGs, while our second part provided the first result about last-iterate convergence when we use **regret decomposition** under the CFR framework for EFGs.

---

> > ### Author Response · Authors · 2022-11-11
> > **Response to some public reviewer**
> >
> > We noticed that some public reviewer had some comments about the second part of our paper. We copy-paste the comments here from our email notification, as they were deleted by him/herself soon while we checked them on OpenReview website. We still post them here since we find the comments valuable, and hope we can clarify them further.
> >
> > "To my knowledge, Perolat et al. (2021) have proposed the last-iterate convergence guarantee for the regret decomposition under the CFR framework for EFGs. In addition, Perolat et al. (2022) have leveraged this technique to build superhuman Stratego AI. Moreover, they simply run the vanilla OMD (FTRL) algorithm on each information set, whereas you need to run the Optimistic FTRL algorithm. More importantly, they seem to not need to compute the exploitability to dynamically update $\tau$ during the learning process."
> >
> > [1] Perolat, Julien, et al. “From Poincaré recurrence to convergence in imperfect information games: Finding equilibrium via regularization.” International Conference on Machine Learning. PMLR, 2021.
> >
> > [2] Perolat, Julien, et al. “Mastering the Game of Stratego with Model-Free Multiagent Reinforcement Learning.” arXiv preprint arXiv:2206.15378 (2022).

---

> > > ### Author Response · Authors · 2022-11-11
> > > **Response to some public reviewer (Cont'd)**
> > >
> > > **Q**: Perolat et al. (2021) has proposed the first last-iterate CFR framework.
> > >
> > > **A**: Thank you for your comment. Firstly, we'd like to emphasize that although Perolat et al. (2021) used the update rule that looks similar to CFR, it is not the **CFR framework** we usually referred to. The update rule in Perolat et al. (2021) is not derived from a **regret decomposition framework**, and in fact, the paper did not even mention the notion of regret, nor claimed that their algorithm is under the CFR framework. In comparison, we provide the result about regret (see Appendix F.3).
> > >
> > > Moreover, the update rule is **continuous-time** while CFR is usually about discrete-time updates. To our best knowledge, the seminal work CFR [1] is about discrete-time and all the follow-up works like MCCFR [2], CFR+ [3], Deep-CFR [4], D-CFR [5] are all talking about discrete-time cases. In contrast, the second part of our paper was about the authentic discrete-time CFR framework. Hence, we are a bit unsure if it is fair to claim the algorithm in Perolat et al. (2021) is really CFR, nor it dilutes/weakens our contribution.
> > >
> > >
> > > Finally, due to the differences in continuous-time and discrete-time algorithms, the analytical techniques in the two papers are **fundamentally different**, as we discussed in our related work section. In Perolat, Julien, et al., a continuous-time Lyapunov argument was used. In our paper, we get the last-iterate convergence through the discrete-time regret analyses of the regularized CFR. Moreover, for average-iterate, we also have **convergence rate** guarantees.
> > >
> > > **Q**: Perolat et al. (2021) used MD (FTRL) while we use the optimistic version.
> > >
> > > **A**:  We resort to the optimistic framework at the very beginning of our paper in order to obtain **faster rates** in general. It was also usually **more compatible** with the last-iterate guarantees. We used it also in order to be **consistent** with our first part. We believe it is also possible to prove results for CFR using the non-optimistic version, under our current proof techniques. We leave it as an important future work.
> > >
> > > **Q**: Perolat et al. (2021) do not need to calculate duality gap to dynamically update $\tau$.
> > >
> > > **A**: Thank you for pointing out so that we can explain a bit about that. Actually, we do not need to calculate the duality gap during execution either (see Corollary 4.4.). Algorithm 1 is only an empirical algorithm for the **first part** (DOMD part) of our paper. We provide the description of a theoretically sound algorithm without the requirement of calculating the duality gap above Theorem 3.2 (highlighted in blue in our latest version).
> > >
> > > Thank you very much again for the valuable comments. Please feel free to let us know if there are any other questions, and we are more than happy to address them.
> > >
> > >
> > > [1] Zinkevich M, Johanson M, Bowling M, et al. Regret minimization in games with incomplete information[J]. Advances in neural information processing systems, 2007, 20.
> > >
> > > [2] Lanctot M, Waugh K, Zinkevich M, et al. Monte Carlo sampling for regret minimization in extensive games[J]. Advances in neural information processing systems, 2009, 22.
> > >
> > > [3] Tammelin O, Burch N, Johanson M, et al. Solving heads-up limit texas hold'em[C]//Twenty-fourth international joint conference on artificial intelligence. 2015.
> > >
> > > [4] Brown N, Lerer A, Gross S, et al. Deep counterfactual regret minimization[C]//International conference on machine learning. PMLR, 2019: 793-802.
> > >
> > > [5] Brown N, Sandholm T. Solving imperfect-information games via discounted regret minimization[C]//Proceedings of the AAAI Conference on Artificial Intelligence. 2019, 33(01): 1829-1836.

---

> ### Author Response · Authors · 2022-11-17
> **Any Further Questions?**
>
> Dear Reviewer, Thank you very much again for the comments. We were wondering if there are any remaining changes you would like us to make or questions you would like us to answer? We are more than happy to address them (before the rebuttal deadline ends on Friday). Otherwise, could you take a chance to reevaluate our paper, and update your score accordingly? Thanks!

---

> ### Author Response · Authors · 2022-12-07
> **Any Further Questions?**
>
> Dear Reviewer,
>
> Thank you very much again for the comments.
>
> We were wondering if there are any remaining changes you would like us to make or questions you would like us to answer? We are more than happy to address them (before the rebuttal deadline ends on Dec 12). Otherwise, could you take a chance to reevaluate our paper, and update your score accordingly?
>
> Thanks!
>
> The Authors

---

### Official Review · Reviewer_M1He · 2022-10-28

**Confidence:** 2
**Correctness:** 3
**Technical Novelty And Significance:** 2
**Empirical Novelty And Significance:** 2
**Recommendation:** 3

**Clarity, Quality, Novelty And Reproducibility:**

Clarity: Paper is clear, well written and mathematically founded.

Quality: In terms of expressing the ideas in writing, the presentation seems as a sequence of theorems without a discussion at the end. This is mainly associated with the scope and goal of the paper, which here seems to be more about the limiting behavior of a game setting, and less about a converging Learning method. In general, there should be a connection/instantiation to a learning procedure.

Novelty: From a first read, it does not overlap with previous papers.

Reproducibility: Seems ok, because the focus here is the modelling, not the experiment. The experiments are there to 'validate' the regularizer effect. Although there is not any code link provided, this is not an issue, because someone can use the pseudocode (ADAPTIVE WEIGHT-SHRINKING ALGORITHM) if necessary, for the reproduction of the last iterate performance.

**Strength And Weaknesses:**

S: The problem is well motivated, the paper is well written and the proofs seem ok from a first read.

W: The impact of the paper as a 'Learning Representations' result needs to be more investigated. The impact on EFG seems to be there, but it is not sufficiently and explicitly linked to a Learning task. There are some experiments in the supplement, but they remain in the setting of a game, rather than Learning/Inference/Prediction.
Therefore the results seemingly lie mainly in the area of Algorithmic GT, and the insertion of the regularizer is not shifting the study necessarily to Learning Theory (given the scope of ICLR).

W2: A 'discussion' section would be more illuminating at the end of the main paper.
The ending of the paper seems a little 'abrupt' in terms of summing up the achieved complexity.

W3: In general, the goal of the main submission document is to discuss the theorems, like the authors are doing in page 7, and not emphasize the proofs if space is limited. For example Lemma 1 and Corollary 4.4. could be placed in the Supplement allowing for more discussion in its place.



**Summary Of The Paper:**

The paper analyzes EFG when a regularization term is inserted in the game setting and it comes up with the first last-iterate convergence results for CFR type algorithms, in contrast to existing results on the average iterate convergence. It provides new algorithms with considerably low complexity and proves rigorously their guarantees.

**Summary Of The Review:**

Overall it seems a rigorous paper, but the focus is on Algorithmic Game Theory, without complete instantiation to Learning Theory (to my opinion) in order to make a clear impact there. A better fit might be SAGT or EC conference.

---

> ### Author Response · Authors · 2022-11-09
> **Response to Reviewer M1He**
>
> **Q**: The paper didn't mention much about learning representations.
>
> **A**: The main concern of our paper is the last-iterate convergence of DOMD and CFR. Such last-iterate convergence would be a huge advantage when comes to using deep neural network to learn the representations in large games like Texas Hold'em. We believe that we are actually establishing some fundamental theoretical result for achieving function approximation in large EFGs.
>
>
> Lastly, learning in games has already been a recognized sub-area of learning and efficient computation of NE is at the heart of the area. For example, Wei et al. (ICLR 2021) which is about last-iterate convergence of OMD in NFGs, was also published on ICLR. Thus, we believe that ICLR is actually suitable for such theoretical work which paves the way for future work on learning representations in games.
>
> **Q** Lack of discussion section.
>
> **A**: Thank you for your comment! Due to the limitation of pages, we move the discussion section to Appendix G. We have added a pointer to it at the end of our main text in the latest version.
>
> **Q**: Should not discuss the proof.
>
> **A**: Thank you for your comment! We have replaced the proof sketch in the second part by a discussion about the impact of last-iterate convergence of CFR framework (please refer to the blue part on Page 9).
>
> **Q**: Reproducibility.
>
> **A**: In supplementary materials uploaded to Openreview, we have provided the code.

---

> ### Author Response · Authors · 2022-11-17
> **Any Further Questions?**
>
> Dear Reviewer, Thank you very much again for the comments. We were wondering if there are any remaining changes you would like us to make or questions you would like us to answer? We are more than happy to address them (before the rebuttal deadline ends on Friday). Otherwise, could you take a chance to reevaluate our paper, and update your score accordingly? Thanks!

---

> ### Author Response · Authors · 2022-12-07
> **Any Further Questions?**
>
> Dear Reviewer M1He,
>
> Thank you very much again for the comments.
>
> We were wondering if there are any remaining changes you would like us to make or questions you would like us to answer? We are more than happy to address them (before the rebuttal deadline ends on Dec 12).
>
> We just noticed that you have just lowered the score without any reasoning or justification. We were wondering what else should we pay attention to, in order to further improve the paper?
>
> Thanks!
>
> The Authors

---

### Official Review · Reviewer_KrJP · 2022-11-03

**Confidence:** 5
**Clarity, Quality, Novelty And Reproducibility:** Clarity is ok.
**Correctness:** 4
**Technical Novelty And Significance:** 3
**Empirical Novelty And Significance:** Not applicable
**Recommendation:** 8

**Strength And Weaknesses:**

Strengths:

* The authors provide a useful set of results around the performance of CFR/DOMWU/DOGDA on strongly-convex strongly-concave saddle EFGs. This is indeed useful work, and I think it deserves publishing.
* The results on CFR seem particularly nice, and less similar to existing results by e.g. Cen et al.
* The $1/t$ iterate convergence in Theorem 3.2 was surprising to me. It seems like a nice result. Unfortunately I was brought in as a last-minute reviewer and did not have time to verify this claim.

Weaknesses:

* The authors mischaracterize the existing literature and open problem on last-iterate convergence guarantees. It is not that surprising that one can get last-iterate guarantees without uniqueness when solving a regularized EFG. This is in some sense already known since the same thing occurs in NFGs, and strongly-convex strongly-concave saddle-point problems more generally.
* The related work discussion is extremely limited in the body of the paper. Many ideas existed already, but those papers are either not cited or only discussed vaguely in the appendix. Nesterov smoothing and the excessive gap technique in particular stand out in this regard. Also, Cen et al. also showed quite similar results for NFGs, which somewhat makes the present paper overselling. I also feel strongly that there should be existing literature on last-iterate convergence in strongly-convex strongly-concave saddle-point problems more broadly, in the first-order methods literature.

**Summary Of The Paper:**

The authors study regularized extensive-form games (EFGs). They show that various algorithms achieve last-iterate convergence guarantees on such games. Then, the authors use the fact that for small amounts of regularization the solution to the regularized problem is also an approximate solution to the original problem.

**Summary Of The Review:**

Fundamentally, I think that this paper mistakes what the research community was trying to achieve by looking for last-iterate convergence guarantees. All the previous literature that the authors cite tries to show last-iterate convergence of OMWU, DOGD, DOMWU, etc, for various cases of bilinear saddle-point problems (matrix games, EFGs, etc). Indeed, there, it is an open problem how to deal with the issue of requiring uniqueness in order to get convergence when using entropy regularization. The authors do not provide a resolution to this problem though: the authors provide a *different* algorithm which yields last-iterate convergence by solving a regularized problem. This approach was already known to yield last-iterate guarantees (e.g. in Cen et al. 2021). OMWU and Reg-OMWU are not the same algorithm, yet the authors repeated conflate these algorithms in the abstract and introduction, thereby giving the misleading impression that they resolve an open problem.

Here is an example of an incorrect claim:

> we first show that dilated optimistic mirror
descent (DOMD), an efficient variant of OMD for solving EFGs, with adaptive
regularization can achieve a fast Oe(1/T) last-iterate convergence in terms of duality gap without the uniqueness assumption of the Nash equilibrium (NE).

The authors show a quite different result: they show that last-iterate convergence occurs in a strong-convex-strongly-concave game, due to the fixed amount of regularization. It was known that such last-iterate guarantees are possible; the uniqueness assumption specifically is trying to deal with last-iterate convergence of the *standard* DOMD algorithm. We already knew from Nesterov smoothing (which should be cited! The authors cite the excessive gap technique paper, but not the original Nesterov smoothing paper) that you can add strong-convex regularization to a problem and get better guarantees.

A minor note here, by the way, is that Wei et al already show last-iterate linear-rate convergence of OGDA in EFGs: their result is for any bilinear saddle-point problem with polyhedral decision sets. It's just that their result requires $\ell_2$ as the regularizer, and thus does not allow dilated regularizers.

Given the present way that the paper is presented, I am leaning towards rejecting the paper, especially because the authors are already aware of these issues and chose not to address them in their revised paper (see ethics concern below). I will say that I actually think the results are pretty nice when viewed simply as results about a different algorithm, and they could arguably be accepted from that perspective. In that case, I would expect the discussion of the obtained results to be more in line with how Cen et al. present their results: they readily acknowledge that their results are on regularized problems, and thus are not directly comparable to the goal of showing last-iterate results for DOMWU/OMWU/etc.

I do have a few questions for the authors:

* Am I correct that in Theorem 3.2, the algorithm completely resets for every $\epsilon$? In particular, if I wanted to get a particular level of precision, I would not actually run this restarting algorithm at all, but instead pick my desired $\epsilon$ and simply start the algorithm at that $\epsilon$?
* For the second statement in Theorem 3.2, is this a property of regularizing with the entropy regularizer, or is it a specific property of Reg-DOMWU? In particular, suppose I solve the entropy-regularized problem with some other algorithm and obtain an iterate guarantee. Can your result be used to directly bound the iterate distance in the original problem? I don't really see why such a property should be specific to running DOMWU on the regularized problem.

---

> ### Author Response · Authors · 2022-11-09
> **Response to Reviewer KrJP**
>
> **Q**: Last-iterate convergence guarantee without uniqueness assumption is not surprising in a regularized game due to the strongly-convex strongly-concave nature of regularization?
>
> **A**: In Theorem 3.1, we prove the convergence to the NE in the **regularized** game. But in Theorem 3.2, we first prove that Reg-DOMD has an $\tilde O(1/T)$ convergence rate in terms of **duality gap** for the **original** problem, which is an extension of Cen et al. to the EFG case. Then, we prove a brand new **slope** result for Reg-DOMWU, which can guarantee that when the duality gap to the NE of the **original** problem is small, then the distance to the NE set of the **original** problem is small.
>
> Therefore, we actually get the **iterate-convergence to the original problem** without uniqueness assumption, which is new even for policy extragradient algorithms with multiplicative updates in NFGs (for the **original problem**, Cen et al. only achieved **duality gap** convergence result).
>
> **Q**: Wei et al. already addressed the issue of OGDA in EFGs.
>
> **A**: Yes, we were fully aware of that. However, by using L2 regularization instead of dilated regularization, the algorithm becomes computationally inefficient for EFGs. That is, the projection to the treeplex takes $O(P^2\log P)$, where $P$ is the number of nodes in the treeplex. On the other hand, by using dilated regularization, it takes linear time for each update. This difference would be significant when comes to large EFGs like Texas Hold'em. Wei et al.'s general results cannot address the case with dilated regularization for EFGs.
>
> **Q**: For any desired precision $\epsilon$, do we need to reset the algorithm in Theorem 3.2?
>
> **A**: No. Thank you for your comment for pointing the confusing description out. In fact, we will initialize $\tau=\tau_0$ for some hyper-parameter $\tau_0$ at the beginning and run Reg-DOMD in episodes. In each episode, we update the parameters $\mathbf z_t$ and $\hat{\mathbf {z}}_{t+1}$ for $\tilde \Theta(1/\tau)$ iterations so that the duality gap of $\hat {\mathbf {z}}_t$ will be lower than $O(\tau)$ according to Lemma D.1 and Theorem 3.1. Then, we will shrink $\tau$ by one half and start the next episode from scratch. Thus, we can guarantee that we will finally reach the desired precision $\epsilon$ in $\tilde O(T)$ iterations. We have updated it in our latest version (highlighted in blue).
>
> **Q**: Why the iterate convergence result only apply on DOMWU.
>
> **A**: To prove the iterate convergence, we need to guarantee that for any $t$ and $i\in {\rm supp}(\mathcal Z^*)$ ($\mathcal Z^*$ is the NE set of the original problem, ${\rm supp}(\mathcal Z^*)=\bigcup_{\mathbf z\in\mathcal Z^*}{\rm supp}(\mathbf z)$), the iterate $z_{t,i}, \hat z_{t,i}\geq \epsilon_{\rm dil}$, where $\epsilon_{\rm dil}$ is a game-dependent constant. Such property will be only guaranteed by the entropy regularization (see Lemma D.6 for details).
>
> **Q**: Ethics review about NeurIPS.
>
> **A**:  In fact, the main concern (final decision) of NeurIPS reviewers is about the missing references. If the reviewer recalls correctly, one reviewer kept silent for the whole rebuttal time, and lowered the score in the **last several hours before the rebuttal ends** to ask us to add references, and said he/she would raise the score back if we did so.  Unfortunately, he/she did not raise the score back, even though we have added the references properly (and did not get to know if it is  proper enough or not, as the rebuttal period ends).
>
> Also, as mentioned before, we **did not** "change the question by instead considering a regularized problem", but instead using regularized problem as a surrogate, and managed to achieve **what we want for the original problem**, with new techniques.
>
> Admittedly, this is not exactly the same as the open problem, which focuses on the authentic OMWU update rule (we previously viewed them as the same class as our regularized-OMD algorithm, as the related reference Cen et al. did, since they both used "Bregman distance associated with entropy"). In the revised paper, we have thus modified the statement about the open problem, just saying that we "make a step towards solving it" (highlighted in blue). Hope it is more accurate and addresses the comment from the reviewer. But still, the results stand, in solving the original problem in terms of **distance-to-the-NE-set**, and are novel in the literature.

---

> > ### Comment · Reviewer_KrJP · 2022-11-17
> > **Response to response**
> >
> > Dear authors, thank you for the response. Below I give my thoughts on the response.
> >
> > # regarding OMWU claims
> > > we actually get the iterate-convergence to the original problem without uniqueness assumption
> >
> > Yes, as I already stated in my review, the slope result is interesting. However, this does not change the fact that the paper is written as if you resolve the question of *last iterate convergence of OMWU without uniqueness* in EFGs. Quoting from the updated version of your paper:
> >
> > > In this paper, we remove the uniqueness condition, while establishing the last-iterate convergence for OMWU in EFGs
> >
> > This is not a correct claim in my opinion. Moreover, it claims to resolve an important problem in the field when it does not, which is unfair towards other authors working on this topic. Again, as I stated in my review, what you show is a related but fundamentally different result: you show that strongly-convex strongly-concave regularized EFGs can be solved and with last-iterate convergence via OMWU *on those regularized games* but *not by running OMWU on the original game*. This is completely analogous to Cen et al. Again, it is an interesting result, but it is also something that seemed basically guaranteed to hold if one knows about Nesterov smoothing and the performance of algorithms on strongly-convex strongly-concave games, and certainly based on Cen et al. The main difference is your slope results, and the generalization to EFGs. Both of these are interesting, but they do not change the fact that you are not resolving the open problem of last-iterate convergence of OMWU without uniqueness on the unregularized game.
> >
> > Let me try to explain one sense in which your result is different: Part of the question regarding last-iterate convergence has to do with an online learning question: what if all the agents are independently running OMWU. Could we hope that they will eventually converge in decentralized fashion to a Nash equilibrium in last iterates? The answer is yes for unique Nash eq (with the caveat that they may need to use constant stepsizes that are compatible with each other). Your paper does not answer this question, because in your result the last-iterate guarantee results from agreeing to solve a strongly-convex-strongly-concave surrogate problem, and moreover it requires agreeing to have a decreasing regularization schedule, and to agree on joint doubling trick iterations. Thus, your paper gives an algorithm with last-iterate guarantees, which is useful if we are using online learning as a tool for performing offline equilibrium computation. However, your result does not really answer whether we can hope for a decentralized group of agents to converge in last iterate to a Nash equilibrium.
> >
> > I'll reiterate that I do think that this paper contains useful results, and I like the approach taken. I just disagree with the way that the results are currently presented, because I think the claims about which questions are being answered are incorrect.
> >
> > # regarding Section 3.2
> > > Notice that although τ is changing, the stepsize η keeps fixed/constant, which differs from
> > Hsieh et al. (2021), where the stepsize is adaptive
> >
> > This seems like an unfair comparison to me. I think what is happening here is that you are avoiding the adaptive stepsize because the adaptive smoothing is playing a similar role.
> >
> > > Therefore, we will select an initial $\tau$ (not relevant to the desired precision ) and adaptively shrink it without knowing the desired precision
> >
> > I get that, my question was about whether this algorithm actually carries forward any information between different choices for $\tau$? In particular, is it simply a doubling trick with complete restarts? I guess so? And that's why you need $\tilde O$ to hide the $\log T$ factor resulting from the doubling?

---

> > > ### Author Response · Authors · 2022-11-19
> > > **Re: Response to response**
> > >
> > > Thanks for the additional feedback. We address your comments as follows.
> > >
> > > Firstly, we would like to kindly remind the reviewer that, we have already revised our statement about solving the "open" problem in the previously updated version. We appreciate the comment from the reviewer regarding the open problem statement, and have changed our claim to **make a step toward** solving it.
> > >
> > > Secondly, thanks for the good catch on the sentence, and sorry for the confusion. That was indeed one place we missed. What we meant was "OMWU-type" algorithms, or, as we now explained more clearly, and as Cen et al. defined, "policy extragradient algorithms with multiplicative updates", which **include** the regularized one (as Cen et al. did). We have searched thoroughly and tried to make the edits consistently. Thanks again for pointing it out.
> > >
> > > **Q**: Reg-OMWU needs to shrink the $\tau$, which means the players cannot reach equilibrium by running the algorithm independently.
> > >
> > >
> > > We appreciate this new piece of comment but respectfully disagree with it. First, our focus throughout is "equilibrium computation" (exactly the same as Cen et al. (2021) and many other references we cited that used "regularization" in solving games), but never claimed to "justify the equilibrium as the emerging behavior of decentralized learning algorithms" (which we are aware that is another important perspective in the learning in games literature).
> > >
> > > On the other hand, even "justifying the equilibrium" can be viewed as "part of the question regarding last-iterate convergence",  existing results mostly also require some "agreement" among agents, if one has to argue: the agents need to agree on using the same Bregman divergence function in the OMD update rule (Daskalakis et al. 2019, Cen et al. 2021).
> > > Moreover, even for the same Bregman divergence (KL divergence), Cen et al. (2021) proposed two update rules, PU & OMWU, and the agents need to agree on using the same update rule to converge (please refer to Theorem 1 in Cen et al. (2021) for details).
> > > In some more advanced and recent results, agents even need to run OMWU with the **same learning rate** (Wei et al. 2021, Lee et al. 2021).
> > > It seems a bit unfair to devalue the contribution of our work from this stand of point.
> > >
> > > We thank the reviewer again for the valuable comment.
> > >
> > > **Q**: Not fair compared to Hsieh et al. (2021).
> > >
> > > **A**: Here we only want to clarify that we are using a **constant** stepsize, instead of an **adaptive** one. Also, note that we did not mean to "compare" which one is "better", but just stated some facts and differences regarding the stepsize being used. Finally, if the reviewer kindly remembered, we added this sentence following the suggestion from one of our NeurIPS reviewers, in order to emphasize that we are using constant-stepsizes (which is usually viewed as more favorable), unlike Hsieh et al. (2021).
> > >
> > > **Q**: Is it a doubling trick with a complete restart?
> > >
> > > **A**: Yes it is, and that was why we have $\tilde O(1/T)$ instead of $O(1/T)$.
> > >
> > >
> > > We would like to thank the reviewer again for the very helpful and valuable comments, especially those on different understandings of certain concepts. They have greatly helped improve our paper. We hope our response and our revision of the paper address your concerns. We are happy to discuss and exchange more thoughts if needed. Thank you very much.
> > >
> > >
> > > [1] Constantinos Daskalakis and Ioannis Panageas. Last-iterate convergence: Zero-sum games and
> > > constrained min-max optimization. In Avrim Blum, editor, 10th Innovations in Theoretical Computer Science Conference, ITCS 2019, January 10-12, 2019, San Diego, California, USA, volume 124 of LIPIcs, pages 27:1–27:18.
> > >
> > > [2] Shicong Cen, Yuting Wei, and Yuejie Chi. Fast policy extragradient methods for competitive games with entropy regularization. NeurIPS 2021.
> > >
> > > [3] Chen-Yu Wei, Chung-Wei Lee, Mengxiao Zhang, Haipeng Luo: Linear Last-iterate Convergence in Constrained Saddle-point Optimization. ICLR 2021
> > >
> > > [4] Lee C W, Kroer C, Luo H. Last-iterate convergence in extensive-form games[J]. Advances in Neural Information Processing Systems, 2021, 34: 14293-14305.

---

> > > > ### Comment · Reviewer_KrJP · 2022-11-20
> > > > **Thanks for the updated response**
> > > >
> > > > I think this discussion was productive, and I am overall happy with the changes made to the paper. As I stated initially, my primary claim was that, to me, the paper was presented as if it resolved the question of OMWU last-iterate convergence in the unregularized game, and I now think that issue has been resolved. As a result, I have updated my score.
> > > >
> > > > Below I have a few nitpicks regarding the last response, though they are not super important.
> > > >
> > > > > Q: Reg-OMWU needs to shrink the $\tau$, which means the players cannot reach equilibrium by running the algorithm independently.
> > > >
> > > > I agree that last-iterate convergence in the context of solving EFGs in an offline sense is useful in the sense that the authors describe. My comment was there because the paper previously read as if it resolved the open question of ``authentic OMWU'' last-iterate convergence, which is motivated by the online setting as well.
> > > >
> > > > I agree that in the online setting, it is slightly unfortunate that agents must agree on a Bregman divergence and stepsize. It would be nice to remove those requirements for sure. That said, I think those are *way* milder assumptions than agreeing to synchronize on a decreasing regularization schedule. But in any case, since the paper now more explicitly explain that it is for the offline setting and Reg-OMWU, this discussion is mostly not important anymore.
> > > >
> > > > > we are using a constant stepsize, instead of an adaptive one
> > > >
> > > > While this is true in a certain sense, there is also a sense in which you are indirectly using an adaptive stepsize. Because the regularizer plays an almost identical role to the Bregman divergence in OMWU, I believe your decreasing regularization schedule is more-or-less equivalent to a decreasing stepsize schedule. In fact, I would not be surprised if it were possible to reinterpret Reg-OMWU as OMWU with some sort of synchronized decreasing stepsize schedule. If you read the paper of Hsieh et al. closely, I also think it may be possible to reinterpret their algorithm as a Reg-OMWU-style algorithm with a fixed stepsize (since they have two different Bregman terms in their setup as well).
> > > >
> > > > # Experiments
> > > > I did have one last comment about the experiments:
> > > >
> > > > For CFR+ and CFR, can you please clarify in the paper whether you are using linear averaging or uniform averaging, and whether you are using alternation? CFR+ is typically defined to mean the CFR decomposition with RM+, *and* linear averaging + alternation.

---

> > > > > ### Author Response · Authors · 2022-11-21
> > > > > **Re: Re: Updated response**
> > > > >
> > > > > We are grateful that the reviewer appreciated our responses and revision, and raised the score accordingly. We are also very grateful for the productive conversations, which helped us improve our paper. Thank you! We appreciate your final comments and address them as follows.
> > > > >
> > > > > **Q**: Decrease $\tau$ is a stronger agreement.
> > > > >
> > > > > **A**: Yes, we agree that decreasing $\tau$ is a stronger agreement than agreeing on the update rule or Bregman divergence. We will add a bit more discussion on this in our next  version of the paper (pdf is not allowed to update currently).
> > > > >
> > > > > **Q**: There is some relationship between adaptive regularization and adaptive stepsize.
> > > > >
> > > > > **A**: Thanks for the insightful comment. Indeed, we acknowledge that there might be some relationship between an adaptive learning rate and an adaptive regularization term. We will discuss it and make the corresponding modifications in the updated paper.
> > > > >
> > > > > **Q**: The details about CFR & CFR+ in the experiments.
> > > > >
> > > > > **A**: The CFR in our paper is decomposition + uniform averaging with RM to minimize the regret. The CFR+ in our paper aligns with that of Tammelin et al. (2015), the original paper that proposed CFR+, which included linear averaging, alternation, and RM+.
> > > > >
> > > > > Lastly, we would like to emphasize that even when we are plotting **Figure 5**, the last-iterate convergence of CFR & CFR+ with regularization, we still keep **all the features** of the original one (RM+, alternation...). Instead, we only add a regularization term to the original objective function and we observed last-iterate convergence for original CFR & CFR+ after that!
> > > > >
> > > > >
> > > > > We would like to thank the reviewer again for the detailed and valuable reviews.
> > > > >
> > > > > [1] Oskari Tammelin, Neil Burch, Michael Johanson, and Michael Bowling. Solving Heads-Up Limit Texas Hold’em. Proceedings of the TwentyFourth International Joint Conference on Artificial Intelligence, IJCAI 2015.

---

> > ### Comment · Reviewer_KrJP · 2022-12-07
> > **Regarding the slope result and related literature**
> >
> > Dear authors,
> >
> > I took a look at the appendix, and specifically Lemma D.5 stood out to me. Could you comment on how this relates to e.g. the saddle-point metric subregularity condition from Wei et al.
> >
> > Also, since this is an important part of the paper, I would suggest cleaning up the proof. I couldn't find a definition of $C_\Omega$, and you probably shouldn't base the proof on a construct $\epsilon_{dil}$ that isn't defined until a later point in the paper.

---

> > > ### Author Response · Authors · 2022-12-09
> > > **Response to KrJP**
> > >
> > > Thanks for the great suggestion, we will make a more detailed comparison in the future version.
> > >
> > > Comparing to saddle-point metric subregularity, the result is different.
> > >
> > > $$
> > > \forall \textbf{z}\in\mathcal Z\setminus \mathcal Z^*,~~~~\sup_{\textbf{z}'\in\mathcal Z} F(\textbf{z})\frac{(\textbf{z}-\textbf{z}')}{\|\textbf{z}-\textbf{z}'\|}\geq C_1\|\textbf{z}-\prod_{\mathcal{Z}^*} \textbf{z}\|\tag{Subregularity}
> > > $$
> > >
> > > $$
> > > \forall \textbf{z}\in \mathcal F\setminus \mathcal Z^*,~~~~\max_{\textbf z'\in \mathcal V^*(\prod_{\mathcal Z^*}(\textbf z))}F(\textbf{z})(\textbf{z}-\textbf{z}')\geq C_2\|\textbf{z}-\prod_{\mathcal{Z}^*} \textbf{z}\|\tag{Lemma D.5.}
> > > $$
> > >
> > > - $\mathcal Z^*$ is the NE set
> > > - $\mathcal F$ is a subset of the treeplex $\mathcal Z$ where $\textbf z\in\mathcal F$ satisfies that $z_i\geq \epsilon_{\rm dil}$ in $supp(\mathcal Z^*)$ and we proved that all iterates of Reg-DOMWU lies in $\mathcal F$ (Lemma D.6.).
> > > - $\forall \textbf z\in V^*(\textbf z^* )$, we have $F(\textbf z^*)^\top \textbf z=F(\textbf z^*)^\top \textbf z^*=\rho$, where $\rho$ is the game value.
> > >
> > > Actually, Lemma D.5. is an extension of Lemma 14 in Wei et al (2021) or Lemma 14 in Lee et al. (2021), which removes the unique NE assumption. The challenges are basically two points.
> > >
> > > - A critical constant is $\xi=\min\lbrace \xi_x,\xi_y\rbrace$ where $\rho$ is the game value, $\xi_x=\min_{i\not\in supp(\textbf x^*)} (\textbf A\textbf y^*)_i-\rho$ and $\xi_y$ defined similarly. When NE is unique, we have $\xi>0$ due to the strict complementary slackness. But when we do not have uniqueness assumption, we only have $\xi\geq 0$.
> > >
> > >   - We change the space we are considering. Instead of using Cartesian coordinate to represent the strategy, we use the pure strategy of the treeplex to represent the strategy (any mixed strategy is a linear combination of pure strategies). In this way, we can still define some $\xi>0$ in Eq (D.25).
> > >
> > > - Wei et al. construct a set $\mathcal X'=\lbrace\textbf x:\textbf x\in\mathcal X,||\textbf x-\textbf x^*||\geq x^*_{\min}\rbrace$. The constant in the proof $x_{\min}^*=\min_{i\in supp(\textbf x^*)} x_i>0$ does not exist when we do not have uniqueness assumption. Because now we have $x_{\min}^*=\min_{i}\inf_{i\in supp(\textbf x^*)} x_i$. It is defined over an **open set** $\lbrace \textbf x^*:i\in supp(\textbf x^*)\rbrace$,  which means that $x_{\min}^*$ may be equal to 0.
> > >
> > >   - We use the feature of (dilated) KL divergence. It will guarantee that for iterates at any time $t$, we have $\hat z_{t,i}\geq \epsilon_{\rm dil}$ for $i\in supp(\mathcal Z^*)$ (Lemma D.6.). Then, we could define $\mathcal X'=\lbrace\textbf x:\textbf x\in\mathcal X,||\textbf x-\prod_{\mathcal X^*}(\textbf x)||\geq \epsilon_{\rm dil}\rbrace$ and we can still prove that for any  $\textbf x\in \mathcal F_x\setminus \mathcal X^*$, there exists $\textbf x'\in\mathcal X'$
> > >
> > >   - $$
> > >     \forall \textbf y,\frac{(\textbf x-\prod_{\mathcal X^*}(\textbf x))^\top \textbf A\textbf y}{\|\textbf x-\prod_{\mathcal X^*}(\textbf x)\|}=\frac{(\textbf x'-\prod_{\mathcal X^*}(\textbf x'))^\top \textbf A\textbf y}{\|\textbf x'-\prod_{\mathcal X^*}(\textbf x')\|}.
> > >     $$
> > >
> > > $C_{\Omega}$ is defined in the Section Preliminaries and $ C_{\Omega} :=\max_{h\in\mathcal H^{\mathcal Z}} |\Omega_h|$ denote the maximum number of indices in each individual information set. We will re-emphasize it in the proof. Also, in the future revision, we will put the definition of $\epsilon_{\rm dil}$ before the proof of Lemma D.5.
> > >
> > > Thank you for your careful reading of our paper and useful suggestions!

---

> ### Author Response · Authors · 2022-11-17
> **Any Further Questions?**
>
> Dear Reviewer, Thank you very much again for the comments. We were wondering if there are any remaining changes you would like us to make or questions you would like us to answer? We are more than happy to address them (before the rebuttal deadline ends on Friday). Otherwise, could you take a chance to reevaluate our paper, and update your score accordingly? Thanks!

---

### Author Response · Authors · 2022-11-09
**General Response**

We thank all the reviewers for the valuable comments. We first would like to re-emphasize several high-level points regarding our contributions that seem to be misunderstood/missed by the reviewers:

- **(Novelty and significance of the CFR part.)** Our paper constitutes of **TWO** main parts, the regularized OMD, and the regularized CFR. They are based on fundamentally different techniques and require solving  technical challenges that are unique for EFGs (instead of NFG as in Cen et. al). We believe our contribution from the CFR-type algorithm and the corresponding analysis is also important.

- **(Technical contributions of the OMD part.)** Even within the first part, we made new  contributions, that is showing last-iterate convergence in terms of **distance to the NE set** of the **original unregularized problem** for multiplicative update rules (OMWU) without the uniqueness assumption, which requires novel techniques that did not exist in the literature ( e.g., Cen et. al ). Note that we did not "change the question by instead considering a regularized problem", but use the regularized one as a **surrogate**, and obtained the **distance-to-NE-set** result as desired, for the **original problem**. The generality of EFG (compared to NFG) also breaks some of the techniques in Cen et. al, and we have addressed them. Note that in our submitted draft, we have pointed out the "connection" and "difference" from Cen et al, clearly, which seems to be missed by some reviewers.

- **(Missing references.)** Another key comment, including that from NeurIPS as one reviewer pointed out, was missing references. We have thus included the discussions of the additional references in our new version. Indeed, except the fact that the **idea** of using regularization somewhere draws some connection, we did not see a strong relevance to weakening the technical contributions of our paper, see e.g., Nesterov smoothing technique (which we have already cited in the form of his book), Hofbauer and Hopkins (2005), Perolat et al. (2021), Leonardos et al. (2021), etc. as we discussed clearly in Sec. A.1. The discussion about the most related references, Cen et al. and Wei et al, have already been provided in the paper. We will be more than happy to include other references if the reviewers see fit.

---

### Author Response · Authors · 2022-11-21
**Acknowledging Review KrJP + Any additional questions from other reviewers?**

Dear reviewers,

We thank reviewer KrJP for the detailed and valuable conversations with us, acknowledging our responses and revisions, and adjusting their score accordingly. We would like to check with other reviewers if they have any further questions and are satisfied with our responses. We are more than happy to address any additional questions.

Thank you all!

Sincerely,

The authors

---

### Decision · Program_Chairs · 2023-01-20

**Decision:**

Accept: poster

**Justification For Why Not Higher Score:**

While the technical results are good and of interest to the ICLR community, this is not an obvious paper that would go in the higher pile. The paper further requires improvements to the presentation and comparison to related work (some of it was addressed during the discussion period, but there is still room for improvement).

**Justification For Why Not Lower Score:**

The main objections to the paper's acceptance were made by a non-expert and do not appear serious enough to be grounds for rejection. The expert opinion is that the paper has interesting and new technical results and as such deserves to be published.

**Metareview: Summary, Strengths And Weaknesses:**

The paper studies last iterate convergence in extensive form games for two classes of algorithms that are typically used in solving extensive form games: (dilated) online mirror descent and (DOMD) and counterfactual regret minimization (CFR). The main idea is to adaptively regularize the original problem and apply the studied algorithms to such adaptively regularized problems to obtain convergence results for the original problem. While similar adaptive regularization techniques exist in optimization and have also been used in context of normal form games, their application to the extensive form games is by no means trivial and required surpassing multiple technical obstacles. Further, for DOMD-type regularized algorithms, the convergence guarantee in terms of the distance to Nash equilibria is novel even for normal form games, while the paper's result about last iterate convergence for regularized CFR is completely new. The paper further provides preliminary numerical results that show that the proposed regularization techniques are promising for standard examples of extensive form games.

There were initial concerns regarding the paper's connection to the work of Cen et al. and more broadly the comparison to related literature. While some of these concerns were cleared during the discussion period, the authors are advised to carefully revise their paper and provide a better overview of existing results. The authors should particularly carefully address the comments made by Reviewer KrJP. Further, the presentation of the paper should be improved to make it more accessible to non-expert readers and it would also be good to add a high level discussion of the results either at the end of the paper or following the theorem statements.

**Note From Pc:**

if the above contains the word "oral" or "spotlight" please see: "oral" presentation means -> notable-top-5% and "spotlight" means -> notable-top-25%. As stated in our emails, we are disassociating presentation type from AC recommendations

**Summary Of Ac-Reviewer Meeting:**

I had scheduled a virtual meeting, however, none of the reviewers was able to show up. Reviewers D4YS and r8KK never responded to my OpenReview message or my email. Reviewers M1He and KrJP followed up with me over email.

Reviewer M1He was not supportive of acceptance. While I can agree with the reviewer's criticism and consider it something the authors should address in the final version, the provided reasons for rejection are not something I would normally use as grounds for rejection (e.g., the paper not having a discussion session or the authors pleading with the reviewers to engage in the discussion and adjust their scores if they had addressed their concerns). The reviewer additionally indicated a low confidence score. The primary objection to the paper is its topic, which I consider well aligned with ICLR and within one of its target areas (online learning, optimization, algorithmic game theory).

Reviewer KrJP is an expert who has worked on closely related questions and whom I had invited as an expert reviewer. This reviewer initially had reservations about the paper (primarily due to the way in which contributions were presented + regarding related work), which were addressed by the authors during the discussion period. The reviewer meaningfully engaged in a discussion with the authors and increased the score as a result.